# Vintix II: Decision Pre-Trained Transformer is a Scalable In-Context Reinforcement Learner

**Andrei Polubarov**
AXXX, Applied AI Institute

**Nikita Lyubaykin**
AXXX, Innopolis University

**Alexander Derevyagin**
AXXX, HSE

**Artyom Grishin**
Innopolis University

**Igor Saprygin**
HSE

**Aleksandr Serkov**
HSE

**Mark Averchenko**
ITMO University

**Daniil Tikhonov**
HSE

**Maksim Zhdanov**
AXXX, NUST MISIS

**Alexander Nikulin**
AXXX, MSU

**Ilya Zisman**
Humanoid

**Albina Klepach**
AXXX

**Alexey Zemtsov**
AXXX, NUST MISIS

**Vladislav Kurenkov**
AXXX, Innopolis University

## Abstract

Recent progress in in-context reinforcement learning (ICRL) has demonstrated its potential for training generalist agents that can acquire new tasks directly at inference. Algorithm Distillation (AD) pioneered this paradigm and was subsequently scaled to multi-domain settings, although its ability to generalize to unseen tasks remained limited. The Decision Pre-Trained Transformer (DPT) was introduced as an alternative, showing stronger in-context reinforcement learning abilities in simplified domains, but its scalability had not been established. In this work, we extend DPT to diverse multi-domain environments, applying Flow Matching as a natural training choice that preserves its interpretation as Bayesian posterior sampling. As a result, we obtain an agent trained across hundreds of diverse tasks that achieves clear gains in generalization to the held-out test set. This agent improves upon prior AD scaling and demonstrates stronger performance in both online and offline inference, reinforcing ICRL as a viable alternative to expert distillation for training generalist agents.

## 1 Introduction

Building general-purpose agents remains a central goal for the AI community. Significant progress has been made in NLP and CV (OpenAI, 2024; Team, 2024; Bai et al., 2025; Wang et al., 2024a), with recent models capable of solving a wide range of tasks. In contrast, reinforcement learning lags behind in developing Large Action Models that can handle broad task diversity. The prevailing approach involves training Transformer-based architectures (Vaswani et al., 2023) on offline datasets in a manner analogous to LLM pretraining (Reed et al., 2022; Gallouédec et al., 2024), and enhancing inference through retrieval-augmented generation (Sridhar et al., 2025). Despite these advances, a key limitation persists: such systems typically do not leverage rewards as an external signal for real-time policy correction and adaptation.

In-context learning (ICL) has made adaptation appear routine for large language models: a short prompt can alter model's behavior without parameter updates (Brown et al., 2020). Transferring in-context learning capabilities to reinforcement learning aims to enable agents to achieve reward-optimizing behavior on new environments without task-specific finetuning. Algorithm Distillation (AD) (Laskin et al., 2022) and the Decision Pretrained Transformer (DPT) (Lee et al., 2023) are flagship approaches in this area, with several recent extensions (Zisman et al., 2024; 2025; Sinii et al., 2024; Nikulin et al., 2025; Dong et al., 2025). While these methods demonstrate in-context

behavior when deployed in single-domain and grid-like environments, their scalability to diverse, continuous-control settings remains underexplored.

Polubarov et al. (2025) scaled in-context RL to the multi-domain setting by training AD on a large cross-domain dataset. Despite strong in-context capabilities on the training tasks, performance on unseen tasks remained limited, leaving room for improvement. At the same time, the ability of DPT to approximate posterior sampling over actions positions it as a strong backbone alternative for scaling in-context RL.

As the task diversity of the data increases, more expressive policy classes (Mandlekar et al., 2021) are required to distill increasingly multi-modal behaviors. Vanilla DPT and its extensions primarily target discrete-action environments (Lee et al., 2023; Dong et al., 2025), where inference-time sampling is straightforward. To fully unlock the potential of DPT in continuous, multi-modal action spaces, such expressive policy classes with native inference-time sampling must be adopted.

In this work, we scale DPT to the cross-domain setting and address the challenges posed by complex, multi-modal continuous action distributions by leveraging a flow-based policy head, thereby enabling in-context adaptation to unseen tasks and parametric variations across diverse environments.

Our key contributions are:

1. Scaling DPT to the cross-domain setting with a wide range of tasks, including robotic locomotion and manipulation, HVAC control, PDE optimization, autonomous driving, and other applications (see Section 3).

2. Substantially improving test-time performance on unseen tasks relative to prior Large Action Models (see Figure 3).

3. Building on Polubarov et al. (2025), we collect and open-source a large cross-domain dataset containing over 700M transitions (a 3.2x increase) across 209 training tasks spanning 10 domains, with 46 additional tasks (compared to 15 previously) reserved for evaluation to support future research in the field (see Table 1).

## 2 RELATED WORK

**In-Context Reinforcement Learning.** The term in-context learning refers to the ability of large language models to adapt to new tasks when prompted with a few demonstrations at inference time (Brown et al., 2020; Liu et al., 2021). Transferring this adaptability to the meta-reinforcement learning (Meta-RL) setting has led to the development of a wide range of approaches (Beck et al., 2025; Moeini et al., 2025). The first family of methods is derived from $RL^2$ (Duan et al., 2016), where task-related information is encoded with sequence models updated through joint online policy iteration. Notable successors include AMAGO-1,2 (Grigsby et al., 2024a;b) and RELIC (Elawady et al., 2024). The second line of work focuses on model-based approaches that either infer a task-related belief state (Dorfman et al., 2021; Wang et al., 2023a) or construct world models for planning (Rimon et al., 2024; Son et al., 2025). The final category frames Meta-RL from the perspective of context-based imitation learning, including Algorithm Distillation (AD) (Laskin et al., 2022) and its extensions (Zisman et al., 2024; Sinii et al., 2024; Tarasov et al., 2025), which aim to distill policy improvement from collected learning trajectories, as well as the Decision Pretrained Transformer (DPT) (Lee et al., 2023) and its variants (Dong et al., 2025; Chen et al., 2025), which perform posterior sampling by training on demonstrations relabeled with optimal actions. Our model builds upon the DPT framework and extends it to the multi-domain continuous control setting.

**Generalist Agents and Large Action Models.** Generalist agent systems are designed to operate across diverse task domains with varying MDP structures. The first category of approaches develops such agents through multi-domain training, either from scratch (Reed et al., 2022; Gallouédec et al., 2024), with potential expansion via retrieval-augmented generation at inference time (Sridhar et al., 2025), or by augmenting vision-language models (VLMs) with action experts. The latter strategies include channel-wise action discretization (Brohan et al., 2023; Team et al., 2024; Kim et al., 2024), compression-based tokenization schemes (Pertsch et al., 2025), and flow-based generative controllers that capture high-dimensional, multi-modal action distributions (Wen et al.,

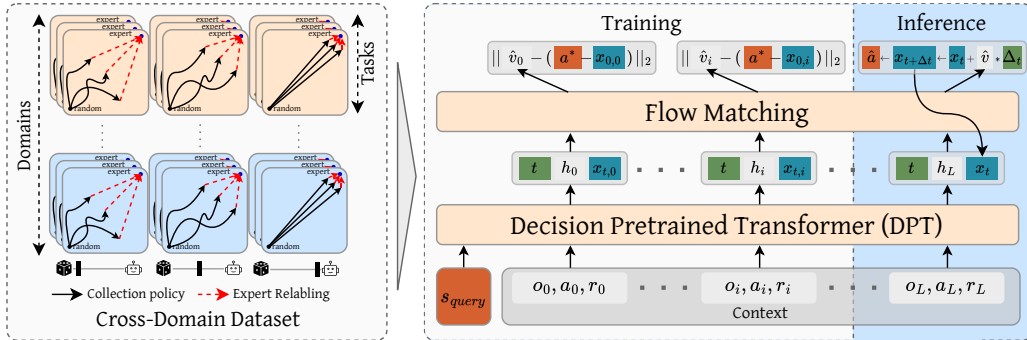

Figure 1: **Approach Overview. Stage 1:** A dataset collected using the noise-distillation technique (Zisman et al., 2024), which covers suboptimal regions of the state space, is relabeled with the demonstrator's optimal actions. **Stage 2:** During training, the flow-matching (OT) head is conditioned on hidden representations from the Decision Pretrained Transformer (Lee et al., 2023). **Stage 3:** At inference, actions are decoded using Heun's method.

2025; Black et al., 2024; Shukor et al., 2025; Intelligence et al., 2025). The second line of research focuses on scaling offline, memory-based meta-reinforcement learning, most notably Algorithm Distillation (AD) (Laskin et al., 2022) and the Decision Pretrained Transformer (DPT) (Lee et al., 2023), to cross-domain settings (Polubarov et al., 2025). Our work follows the latter direction, enriching cross-domain datasets and leveraging inference-time self-correction through in-context reinforcement learning and prompting with expert demonstrations, rather than conditioning on vision-language model representations.

**Flow Based Methods in Reinforcement Learning.** Following the success of denoising diffusion models (Sohl-Dickstein et al., 2015; Ho et al., 2020; Dhariwal & Nichol, 2021) and flow matching approaches (Lipman et al., 2023; Esser et al., 2024) in modeling complex distributions, researchers have sought to leverage their expressive power and mode-preserving properties for continuous control applications (Zhu et al., 2024). The main research directions include: augmenting Tabula Rasa reinforcement learning algorithms with diffusion and flow matching methods (Wang et al., 2023b; Chi et al., 2024; Wang et al., 2024b; Zhang et al., 2025; Park et al., 2025); applying generative flow-based models as planners and world models (Janner et al., 2022; He et al., 2023; Farebrother et al., 2025); using flow matching as an action expert for generalist agents (Ni et al., 2023) and vision-language-action (VLA) models (Black et al., 2024; Shukor et al., 2025; Intelligence et al., 2025). Our work is most closely aligned with the last direction, focusing on the application of flow-based policies in offline, memory-based Meta-RL setting.

## 3 APPROACH

Our approach extends DPT (Lee et al., 2023) to the multi-domain continuous control setting with complex multi-modal action distributions. Previous DPT variants that operated on continuous actions (Dong et al., 2025) yield only moderate results, presumably because Gaussian heads fail to capture multi-modality effectively, creating a likelihood mismatch for complex action posteriors. To address these limitations and enable native sampling from action distribution during online inference, our approach integrates a rectified-flow objective with in-context conditioning, allowing a single model to represent richer action distributions. To scale up training, a diverse dataset covering 10 domains with distinct MDP structures was collected. The following sections describe the dataset, the architecture of the flow-based head for DPT, and our training and inference procedures.

### 3.1 DATASET

The training dataset is collected using the Continuous Noise Distillation (CND) method described in the `Vintix` paper (Polubarov et al., 2025). Progressive action noising alters demonstrator policy quality, thereby increasing the coverage of the visited $\{s, a, r\}$ tuples distribution (see Appendix G

for task-level reward progression graphs), which typically improves test-time performance (Brown et al., 2019). The collected state-action transitions are then relabeled with demonstrator actions as required by the DPT pipeline (Lee et al., 2023). Our training dataset spans 10 diverse domains, which we describe in detail in Appendix C.1. Higher-level dataset statistics are listed in Table 1, and a task-level dataset summary is available in Appendix I. To test the model's adaptation to unseen environments, we reserved 46 tasks across all domains for validation and excluded them from training. A description of the train–test split and its motivation is provided in Appendix C.2.

| Domain | Tasks | Episodes | Timesteps | Sample Weight |
|---|---|---|---|---|
| Industrial-Benchmark | 16 | 288k | 72M | 10.1% |
| Bi-DexHands | 15 | 216.2k | 31.7M | 4.5% |
| Meta-World | 45 | 670k | 67M | 9.4% |
| Kinetix | 42 | 1.1M | 62.8M | 8.9% |
| CityLearn | 20 | 146.4k | 106.7M | 15.0% |
| ControlGym | 9 | 230k | 100M | 14.1% |
| HumEnv | 12 | 120k | 36M | 5.1% |
| MuJoCo | 11 | 665.1k | 100M | 14.1% |
| SinerGym | 22 | 42.3k | 30.9M | 4.4% |
| Meta-Drive | 17 | 271.9k | 102.6M | 14.4% |
| **Overall** | **209** | **3.8M** | **709.7M** | **100%** |

Table 1: **Training Dataset Summary.** Training dataset covers 209 distinct tasks across 10 domains ranging from robotic manipulation to HVAC control and autonomous driving

## 3.2 MODEL ARCHITECTURE

The model consists of three blocks: an encoder that maps input actions, observations, and rewards into fixed-size embedding vectors; a transformer backbone responsible for the main representational capacity; and a flow-based decoder head conditioned on the transformer's hidden states, which outputs actions in the original action space (see Figure 1).

To address variability in action and observation spaces, we partition all tasks into non-overlapping groups sharing the same action-observation structure. Each group is processed by its own encoder and decoder (both represented as MLPs), making the model task-agnostic within each group and enforcing reliance on contextual information.

As the backbone, we used the TinyLLaMA (Zhang et al., 2024a) implementation of the transformer. All positional encodings were removed, as they are not required in DPT (Lee et al., 2023). The final model consists of 16 layers, 24 heads, an embedding size of 1536, and a post-attention feed-forward hidden size of 6144, resulting in a total of 928 million parameters. Training was performed on 8 H100 GPUs with a batch size of 64 and input sequences of length 4096. A full description of the training setup is provided in Appendix E.

Given the transformer output $h \in \mathbb{R}^d$, we parametrize the context-dependent vector field $u(t, h, x_t)$ : $[0, 1] \times \mathbb{R}^d \times \mathbb{R}^a \to \mathbb{R}^a$ using a time encoder $\gamma : [0, 1] \to \mathbb{R}^{d_\gamma}$ and an MLP $v_\eta(\gamma(t), h, x_t)$ : $\mathbb{R}^{d_\gamma + d + a} \to \mathbb{R}^a$.

The vector field defines a context-dependent flow $\psi(t, h, x_0) : [0, 1] \times \mathbb{R}^d \times \mathbb{R}^a \to \mathbb{R}^a$ as the solution to $\dot{x}_t = v(t, h, x_t)$ with initial condition $x_0$. The flow at terminal time $t = 1$ is defined as the policy

$$\pi(\cdot \mid h) = \psi(1, h, \cdot),$$

so that sampling $a \sim \pi(\cdot \mid h)$ corresponds to drawing $x_0 \sim p_0$ and integrating the ODE to $t = 1$.

## 3.3 TRAINING

The data-loading pipeline follows the vanilla DPT setup (Lee et al., 2023) and samples $(o_q, C, a^\star)^\tau$ tuples from the multi-task cross-domain dataset $\mathcal{D} = \bigcup_\tau \mathcal{D}\tau$. Here, $o_q$ is the query observation, $a^\star \sim \pi^\star(\cdot \mid o_q)$ is the corresponding demonstrator action, and $C = \{(o_i, a_i, r_i)\}_{i=1}^L$ denotes a task-specific context of length $L$. Each context element consists of the observation, the applied action,

and the resulting reward. Unlike the vanilla DPT implementation, which also includes the next observation $o'$, we omit $o'$ as our experiments indicate that it does not affect model performance.

Each input sequence consists of a BOS token, one query token, and $L$ context tokens that are randomly permuted. A causal Transformer encodes the sequence and produces hidden states $\{h_j\}_{j=0}^{L+1}$. A context-conditioned vector-field head supervises all positions $j \in \{1, \ldots, L+1\}$ to predict $a^\star$ by minimizing the rectified-flow matching objective (Liu et al., 2022):

$$\mathcal{L}_{\mathrm{RF}} \;=\; \mathbb{E}_{t_j \sim \mathcal{U}(0,1),\, x_{0,j} \sim \mathcal{N}(\mathbf{0}, I_a)} \big\| v_\eta\big(h_j, x_{t,j}, \gamma(t_j)\big) - (a^\star - x_{0,j}) \big\|_2^2, \quad x_{t,j} = (1-t_j)\, x_{0,j} + t_j\, a^\star$$

We encode $t_j \in [0,1]$ using a sinusoidal time embedding $\gamma(t_j) = [\sin(t_j f); \cos(t_j f)] \in \mathbb{R}^{d_\gamma}$. The frequency vector $f \in \mathbb{R}^{d_\gamma/2}$ is learnable and initialized on a logarithmic scale over $[f_{\min}, f_{\max}]$:

$$f_k \;=\; f_{\min} \left( \frac{f_{\max}}{f_{\min}} \right)^{\frac{k}{d_\gamma/2-1}}, \quad k = 0, \ldots, \tfrac{d_\gamma}{2} - 1.$$

Pseudo-code formalizing the training step of our model is provided in Appendix B.1.

## 3.4 Inference

Policy evaluation is conducted in two regimes: online and offline, following the DPT paper. In the online setting, the model begins with an empty context $C = \{.\}$ and incrementally appends observed $(o_q, a, r)$ interactions during deployment. When $|C|$ exceeds the maximum length $L$, the oldest transition is removed. In the offline setting, the context $C = \{(o_i, a_i, r_i)\}_{i=1}^{L_C}$ with $L_C \leq L$ is fixed and remains unchanged throughout inference.

Given a selected task-group $g$ with action dimension $g_a$, $x_0 \sim \mathcal{N}(\mathbf{0}, I_{g_a})$ is sampled from the base distribution and integrated through the learned vector field $v_\eta$ from $t = 0$ to $t = 1$, yielding $x_1$, which is then taken as the output action $a$. Heun's method (second-order Runge–Kutta) with $M$ uniform steps and step size $\Delta t = 1/M$ is applied for numerical ODE integration. The vector field is conditioned on the hidden state of the last Transformer token $h_L$. Appendix B.2 provides the full inference procedure.

## 4 Experimental Evaluation

### 4.1 Evaluation Details

▶ **Metrics** The main evaluation metric for all experiments in this section is the total episode return, normalized with respect to random and expert policy scores: $score^{normalized} = \frac{score^{raw} - score^{random}}{score^{demonstrator} - score^{random}}$, following Gallouédec et al. (2024); Sridhar et al. (2025); Polubarov et al. (2025). Although demonstrators had to be retrained, the normalization scores align with those reported in Polubarov et al. (2025) within one standard deviation for all overlapping domains. Task-level normalization scores for all domains considered in the study are provided in Appendix H. To compute aggregated scores per domain, we used the inter-quartile mean (IQM) implementation of Agarwal et al. (2021), as it yields more robust performance estimates.

▶ **Baselines** We compare `Vintix II` against two prior action models: `Vintix` (Polubarov et al., 2025) and `REGENT` (Sridhar et al., 2025). We leave out the comparison with `JAT` (Gallouédec et al., 2024) as it was shown (Polubarov et al., 2025) to under-perform `Vintix` by a substantial margin on all of the domains considered. Comparisons with `Vintix` are conducted in both online and offline settings, as both modes are supported, whereas comparisons with `REGENT` focus on prompted (offline) evaluation, since it was designed specifically for this deployment scenario. Scaled returns for `REGENT` are taken from the original paper. It is important to note that, due to the use of improved demonstrators for Meta-World in both our work and `Vintix`, their absolute returns are equal to or higher than those reported in `JAT` and `REGENT` (see appendix D.1). Although this lowers our normalized scores relative to those reported in `REGENT`, we retain the comparison, as our focus is on measuring the ability to match demonstrator performance rather than on absolute score differences.

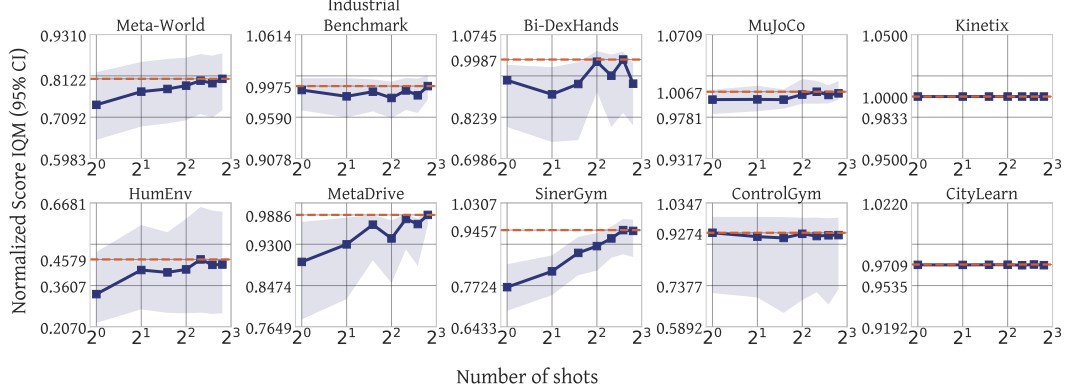

Figure 2: **Online inference on training tasks** Although the model starts with no context, it infers relevant task-specific information for self-correction. All runs are conducted with four different random seeds, and the IQM of normalized scores is reported.

We do not compare our model with $RL^2$ (Duan et al., 2016), Amago-$\{1, 2\}$ (Grigsby et al., 2024a;b), or RELIC (Elawady et al., 2024), as these Meta-RL algorithms operate in an online setting, whereas our work focuses on training from static datasets.

▶ **Evaluation Protocols** We conducted all experiments in both online and offline setups as described in Section 3.4. Online inference uses FIFO-style memory filling, effectively implementing a sliding attention window over past interaction history, while offline inference is performed with a fixed set of demonstrator episodes serving as the task-specific context.

It should be noted that the evaluation setup for Meta-World was modified. In both `Vintix` and `REGENT`, due to inheritance from `JAT` (Gallouédec et al., 2024), goal positions are not fixed during dataset collection and evaluation, thereby violating the ML1 protocol (Yu et al., 2021) for measuring goal adaptation within a single task. To address this, we fixed the goal position across environment resets for all dataset collection and evaluation runs. Consequently, our approach was exposed to orders of magnitude fewer goal positions during pre-training and was therefore evaluated under a more challenging setup.

## 4.2 Performance on Training Tasks

▶ **Online Deployment** Firstly, we evaluate the ability of our model to perform iterative self-correction on training tasks with no context provided. To verify this, the model was deployed on 209 training tasks using the online inference procedure.

Figure 2 shows normalized returns per episode during online inference. Most domains exhibit consistent improvement over the course of deployment, highlighting the model's test-time adaptive behavior. On Kinetix, ControlGym, MuJoCo, and MetaDrive, near-demonstrator performance is reached from the very first episode, indicating that DPT infers a strong prior for training tasks and requires only a few episodes for correction. Individual task-level scores for training environments are provided in Appendix H.1.

▶ **Offline Deployment** To examine how shifting to the offline evaluation scenario affects performance on training tasks, we re-ran the model with 2500 task-specific transitions provided.

As shown in Figure 3, additional task-specific prompts improve performance across all 10 domains, yielding an average gain of +4.1%. This represents a considerable improvement given that online performance on training tasks is already near-demonstrator. Scaled performance for offline runs is reported in Appendix H.2.

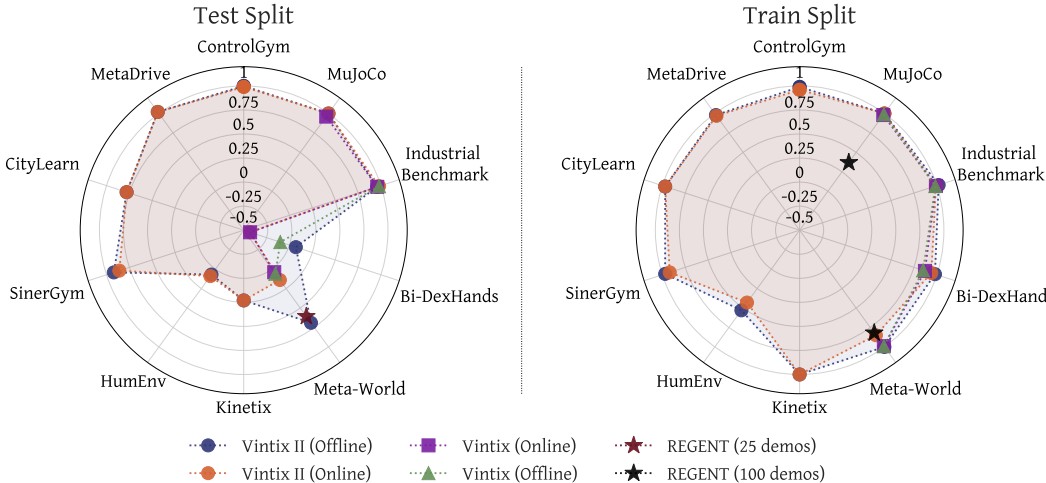

Figure 3: **Domain-level normalized scores** (Left) Evaluation results for 46 tasks *unseen* during training. (Right) Evaluation results for 209 *training* tasks. Offline runs of our model were conducted with a prompt size of 2500 transitions, compared to 5000 transitions for `Vintix`, and 25 and 100 episodes for `REGENT` on testing and training tasks, respectively. Domain-level aggregation is performed using IQM.

## 4.3 PERFORMANCE ON UNSEEN TASKS

▶ **Offline Deployment**  Next, we benchmark the model's ability to infer demonstrator-level policies on entirely unseen tasks from a limited set of expert demonstrations (offline scenario). For this purpose, we evaluated our model on all 46 held-out tasks across 10 domains. The prompt size was capped at 2500 transitions to ensure a fair comparison with `REGENT` (25 demonstrations) on the Meta-World ML45 split, as our model's context length is restricted to 4096 tokens (see Section 3.2) and thus cannot accommodate more than 40 episodes of demonstrations on Meta-World. `Vintix` was evaluated on four overlapping domains (Industrial-Benchmark, Bi-DexHands, MuJoCo, and Meta-World) with demonstrator prompts provided.

The results are summarized in Figure 3. In the offline setting, our model achieves over 75% of demonstrator performance on entirely unseen tasks in MetaDrive, CityLearn, SinerGym, and ControlGym domains (102%, 78%, 92%, and 100% normalized scores, respectively). When compared with the prompted version of `Vintix`, it yields improvements in scaled return of +17% on Bi-DexHands, +4% on MuJoCo with parametric variations, and +63% on the Meta-World ML45 split. Furthermore, comparison with `REGENT` reveals that our approach reaches an 8.2% higher normalized return on 5 unseen tasks in the Meta-World ML45 benchmark. More detailed, environment-level scaled returns are reported in Appendix H.2.

These results suggest that the flow-based DPT architecture provides a strong inductive bias, supporting the emergence of fully parametric in-context imitation, which in turn leads to enhanced performance compared to prior action models.

▶ **Online Deployment**  Further experiments aim to evaluate the ability of our model to exhibit adaptive behavior in entirely unseen environments and their task variations, without any prior information provided. To test this, we rerun our model on 46 unseen tasks, starting with an empty initial context, which is referred to as online evaluation in Section 3.2.

Figure 3 shows that cold-start evaluation of our model performs on par with the prompted version for 8 out of 10 domains, while it lags behind on the Meta-World ML45 and Bi-DexHands ML20 benchmarks. However, several studies (Anand et al., 2021; Grigsby et al., 2024b) argue that task adaptation on the Meta-World ML45 split is unrealistic without additional information, which may explain moderate results in this domain. Bi-DexHands ML20 is even more challenging due to its

higher control dimensionality, variable observation and action space structures, and a smaller number of training tasks (Chen et al., 2022). Task-level granularity is available in Appendix H.1.

Since our data collection and evaluation protocols for Meta-World fix the end-effector goal state between resets and ensure that training and evaluation goal sets are non-overlapping, the training tasks in Meta-World effectively implement the ML1 benchmark for held-out goal variations, making them suitable for assessing in-context capabilities. Accordingly, Figure 3 (right radar plot) reports online evaluation results for the Meta-World ML1 split. The results show that demonstration-less version of our model achieves an 85% normalized score, outperforming REGENT, which was provided with 100 expert-level episodes covering richer end-effector goal distribution, by 3%.

The observed advantage may be attributed to deployment-time self-corrective behavior, which is presumably induced by DPT's potential ability to effectively implement context-based Bayesian posterior sampling (Lee et al., 2023).

## 4.4 ANALYSIS AND ABLATIONS

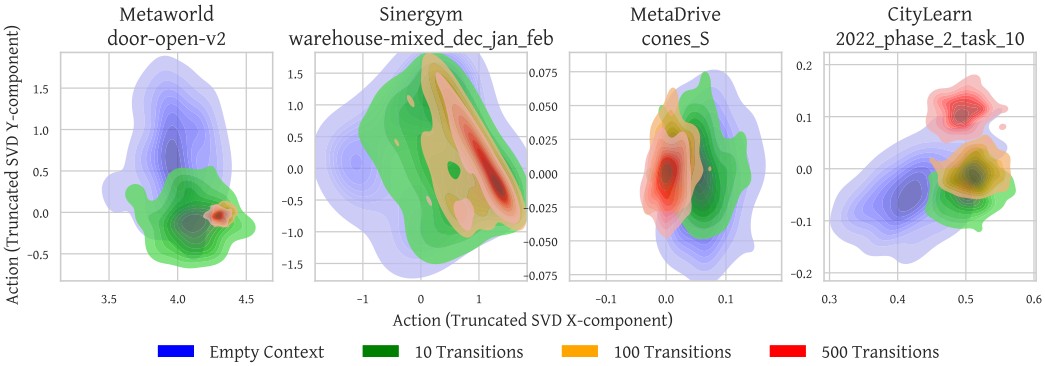

Figure 4: **Action beliefs over context sizes** Action distributions are shown for different prompt sizes during offline evaluation, ranging from no context to 10, 100, and 500 transitions of task-specific demonstrator data. Projections into 2D space are obtained using Truncated SVD. The gradual decrease in distributions' entropy indicates that our model exhibits a behavior consistent with posterior sampling.

▶ **Progressive Concentration of Action Beliefs** One of the main theoretical results of Lee et al. (2023) was to show that DPT implements in-context posterior sampling (PS), a generalization of Thompson sampling for RL in MDPs. To test whether our trained agent exhibits this behavior, we analyze its action distributions as a function of context length $L$. For each task in Figure 4, we fix a query observation $o_q$ from evaluation rollouts and vary the context from empty to 500 $(o_q, a, r)$ demonstrator tuples. At each $L$, we draw 100 action samples for every $o_q$. The KDEs in Figure 4 consistently show a posterior-like contraction: with short contexts the distributions are wide and uncertain, while longer contexts sharpen into narrow peaks, accompanied by a monotone decrease in entropy, evidence consistent with in-context PS. More action belief graphs can be found in Appendix F.

▶ **Effect of Number of Demonstrations** To conduct an ablation study assessing how the size of the demonstration prompt influences performance on unseen tasks during offline evaluation, we redeployed our model with task-specific prompts ranging from 500 to 4000 $(o_q, a, r)$ tuples. Context size is reported in transitions rather than episodes, as DPT operates on a permuted, episode-agnostic context dataset. Results for REGENT on Meta-World ML45 are compared on a like-for-like basis, since both models cap episode length at 100 timesteps in this domain. Vintix was evaluated with a 5000-transition prompt across all shared domains.

Figure 5 aggregates the results of the experiment, showing that performance improves with prompt size for Meta-World, Industrial-Benchmark, and SinerGym, while remaining stable for all other

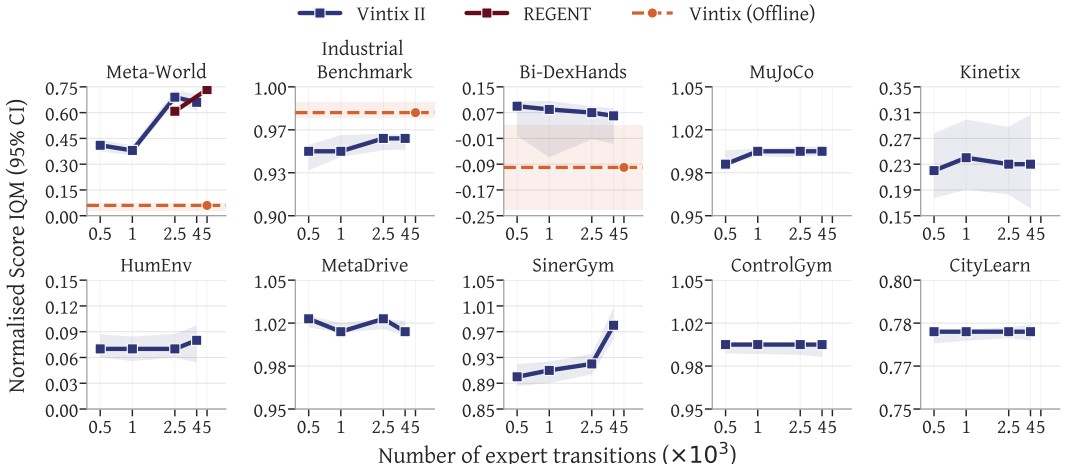

Figure 5: **Normalized returns vs. number of demonstrations** Offline evaluation is conducted on 46 held-out tasks with task-specific prompts of varying size. Results for `Vintix` and `REGENT` are reported on the corresponding domains. IQM aggregation is applied across 4 random seeds.

domains. On Meta-World, our model scales slightly faster than `REGENT`; however, it is constrained by a context length of 4096, which limits comparisons under larger demonstration budgets.

Overall, observed results suggest that augmenting the context with additional data boosts model's performance, or at least does not degrade it in cases where only a few demonstrations are sufficient for successful task completion.

## 5  CONCLUSION

In this work, we demonstrated that the Decision Pretrained Transformer paired with a flow-based generative policy head provides a strong framework for scaling In-Context Reinforcement Learning across multiple diverse domains. Our model exhibits substantial capabilities both in zero-shot evaluation and when provided with a few demonstrations, and to our knowledge, it is the first Large Action Model that successfully operates in both regimes simultaneously. Furthermore, in contrast to Sridhar et al. (2025), which relies on a semi-parametric approach with additional retrieval modules, our agent is deployed in a fully parametric setting with fewer design choices at the inference stage. The 3.2x expansion of the dataset collected by Polubarov et al. (2025) enhances large-scale cross-domain training while keeping it compatible with all major ICRL approaches, bringing the research community closer to creating generalist systems that support cross-domain knowledge transfer (Beck et al., 2025; Moeini et al., 2025).

While current results are promising, it should be noted that even with the expanded dataset, our model is trained with a token-to-parameter ratio of less than one. Recent work on scaling laws for large foundation models reports an optimal ratio of around 20 tokens per parameter (Hoffmann et al., 2022; Besiroglu et al., 2024). This underscores the importance of further scaling training datasets and highlights the need for comprehensive studies of scaling laws for large action models. Another limitation lies in demonstration-less evaluation, which still lags behind prompted runs, suggesting that although ICRL models are strong in exploitation, their test-time exploration capabilities remain limited. In addition, the challenge of developing action models that are agnostic to the input-output dimension remains open, restricting current models from transferring to entirely unseen domains and limiting their applicability in practical scenarios.

## REPRODUCIBILITY AND LLM STATEMENT

To ensure reproducibility of our results, we provide implementation details in Appendix B and Appendix E. Additionally, source code and datasets will be made available to reviewers during the

rebuttal process, as they require polishing for improved accessibility. LLMs (specifically ChatGPT) were used solely to refine the manuscript.

## ACKNOWLEDGMENT

This work was supported by the Ministry of Economic Development of the Russian Federation (Agreement No. 139-10-2025-034 dated June 19, 2025, unique identifier IGK 000000C313925P4D0002).

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

## A  BACKGROUND

**Reinforcement Learning**  Reinforcement learning (RL) (Sutton & Barto (1998)) formalizes sequential decision making as a Markov Decision Process (MDP) $(\mathcal{S}, \mathcal{A}, p, r, p_0, \gamma)$. Here $\mathcal{S}$ and $\mathcal{A}$ denote the state and action spaces, $p(s' \mid s, a)$ is the transition kernel, $r(s, a)$ is the reward function, $p_0$ is the initial-state distribution, and $\gamma \in [0, 1)$ is a discount factor. A policy $\pi(a \mid s)$ induces trajectories $(s_0, a_0, r_1, s_1, \dots)$ with $s_0 \sim p_0$, actions sampled from $\pi(\cdot \mid s_t)$, and next states drawn from $p(\cdot \mid s_t, a_t)$. The standard objective is to find a policy that maximizes the expected discounted return

$$J(\pi) = \mathbb{E}\Big[ \sum_{t=0}^{\infty} \gamma^t r_{t+1}\Big].$$

Classical RL methods estimate value functions and improve policies through repeated interaction with a fixed environment. In this setting a separate policy is typically optimized for a single MDP, so even moderate changes in the reward function or transition dynamics tend to degrade performance and often require retraining, rather than yielding an agent that robustly operates across different dynamics.

**In-Context Reinforcement Learning**  Unlike standard RL, ICLR refers to agents that are capable of learning during inference. Such agents are conditioned not only on the current state $s_t$, but also on some context $C$, which helps the model understand the dynamics of the environment and maximize reward through a trial-and-error paradigm. Prominent representatives of such models are AD and DPT (Laskin et al. (2022); Lee et al. (2023)). Both models are capable not only of solving training tasks, but also of adapting to unseen dynamics.

DPT's main advantage is that its behaviour during inference can be interpreted as Bayesian posterior sampling. To achieve this, the authors pretrain a transformer $M_\theta(\cdot \mid s_{\text{query}}, D_j)$ on an offline dataset of trajectories using a simple supervised objective: given a query state $s_{\text{query}}$ and an in-context dataset of past transitions $D_j = \{(s_1, a_1, s_1', r_1), \dots, (s_j, a_j, s_j', r_j)\}$,

the model is trained to predict the optimal action $a^\star$:

$$\mathcal{L}(\theta) = \mathbb{E}_D \left[ \sum_{j \in [n]} l(M_\theta(\cdot \mid s_{\text{query}}, D_j), a^\star) \right],$$

where $l(M_\theta(\cdot \mid s_{\text{query}}, D_j), a^\star) = -\log M_\theta(a^\star \mid s_{\text{query}}, D_j)$. Parameters $\theta$ are optimized to minimize $\mathcal{L}(\theta)$.

The authors evaluate the model in two regimes: offline and online. In the offline regime the context $D_j$ is provided in advance, while in the online setting $D_j$ is extended with the interaction history. Sampling actions from the output distribution $M_\theta(\cdot \mid s_{\text{query}}, D_j)$ leads to behaviour that combines efficient online exploration with conservative offline performance. This is why we focus on these algorithms in our work. A sparse training dataset that consists of many different tasks from different domains may further improve performance and help to build a strong large action model within the ICRL framework.

**Flow Matching**  Flow matching (Lipman et al. (2023); Liu et al. (2022)) is a widely used framework for building generative models. It offers simple training and fast inference, while being capable of modeling complex distributions. These properties have led to the popularity of flow matching in various continuous-control applications (Park et al. (2025); Zhang et al. (2025); Black et al. (2024)).

Suppose a target distribution $p(x)$ on the $d$-dimensional space $\mathbb{R}^d$ is given. Consider an ODE $\frac{dx}{dt} = v(x, t)$ with initial condition $x = x_0$, where the velocity field is $v : \mathbb{R}^d \times [0, 1] \to \mathbb{R}^d$. The flow $\psi : \mathbb{R}^d \times [0, 1] \to \mathbb{R}^d$ is the collection of solutions of this ODE such that, for any $x_0$, we have $\psi(x_0, 0) = x_0$ and $\frac{d}{dt} \psi(x_0, t) = v(\psi(x_0, t), t)$. The goal of flow matching is to fit a velocity field $v_\theta(x, t)$ with the corresponding flow $\psi_\theta(x, t)$ that transports an initial simple Gaussian distribution $\mathcal{N}(0, I_d)$ at $t = 0$ to the target distribution $p(x)$ at $t = 1$.

In this work we consider rectified flow matching (Liu et al. (2022)), as it provides a particularly simple training objective. Given a pair $(x_0, x_1)$ with $x_0 \sim \mathcal{N}(0, I_d)$ and $x_1 \sim p(x)$, and a time $t \sim \mathrm{Unif}(0, 1)$, the linear interpolation is defined by $x_t = (1 - t)\, x_0 + t\, x_1$. The loss is

$$\mathcal{L}(\theta) = \mathbb{E}_{x_0 \sim \mathcal{N}(0, I_d),\, x_1 \sim p(x),\, t \sim \mathrm{Unif}(0,1)} \big\| v_\theta(x_t, t) - (x_1 - x_0) \big\|_2^2.$$

Once $v_\theta$ is trained, samples $x_1$ from $p(x)$ can be obtained by first sampling $x_0$ from $\mathcal{N}(0, I_d)$ and then numerically solving the ODE up to $t = 1$.

## B  ALGORITHMS

### B.1  TRAIN STEP

---

**Algorithm 1** Training step

---

**Require:** causal transformer $T_\theta$; encoders $\phi_o, \phi_a, \phi_r$; flow head $v_\eta$; learnable BOS embedding $\beta$; learnable frequency vector $f$

1: Sample a minibatch $\{(o_q^{(b)}, C^{(b)}, a^{\star(b)})\}_{b=1}^B$
2: **for** $b = 1..B$ **do**
3:     $x_{0:L+1}^{(b)} \leftarrow [\beta,\ \mathrm{cat}(\phi_o(o_q^{(b)}), \mathbf{0}, \mathbf{0}),\ \{\mathrm{cat}(\phi_o(o_i), \phi_a(a_i), \phi_r(r_i))\}_{i=1}^L]$
4: **end for**
5: $h_{0:L+1} \leftarrow T_\theta(x_{0:L+1})$
6: **for** $b = 1..B$ **do**
7:     Sample $x_{0,j}^{(b)} \sim \mathcal{N}(\mathbf{0}, I_a), t_j^{(b)} \sim \mathcal{U}(0, 1)$ for $j = 1..L+1$
8:     $x_{t,j}^{(b)} \leftarrow (1 - t_j^{(b)}) x_{0,j}^{(b)} + t_j^{(b)} a^{\star(b)}$
9:     $\hat{v}_j^{(b)}(t, c, x^t) \leftarrow v_\eta(\gamma(t_j^{(b)}), h_j^{(b)}, x_{t,j}^{(b)})$
10: **end for**
11: $\mathcal{L}_{\mathrm{RF}} \leftarrow \frac{1}{B} \sum_{b=1}^B \| \hat{v}^{(b)} - (a^{\star(b)} - x_0^{(b)}) \|_2^2$

---

### B.2  INFERENCE STEP

---

**Algorithm 2** Inference step

---

**Require:** causal transformer $T_\theta$; encoders $\phi_o, \phi_a, \phi_r$; flow head $v_\eta$; BOS embedding $b$
**Require:** $C$ (might be empty), query observation $o_q$

1: $x_{0:L} \leftarrow [b,\ \mathrm{cat}(\phi_o(o_q, \mathbf{0}, \mathbf{0}),\ \{\mathrm{cat}(\phi_o(o_i), \phi_a(a_i), \phi_r(r_i))\}_{i=1}^L]$
2: $h_{0:L} \leftarrow T_\theta(x_{0:L})$
3: Sample $x_0 \sim \mathcal{N}(\mathbf{0}, I_a)$
4: $x \leftarrow x_0$
5: $\Delta t \leftarrow 1/M$
6: **for** $m = 0..M - 1$ **do**
7:     $t \leftarrow m \Delta t$
8:     $k1 \leftarrow v_\eta(\gamma(t/M), h_L, x)$
9:     $k2 \leftarrow v_\eta(\gamma(t/M + \Delta t), h_L,\ x + k1 * \Delta t,)$
10:     $x \leftarrow x + \frac{(k1 + k2)}{2} * \Delta t$
11: **end for**
12: **return** $a \leftarrow x$

---

# C  DATASET DETAILS

## C.1  GENERAL INFORMATION

**MuJoCo.** MuJoCo (Todorov et al., 2012) is a physics engine developed for multi-joint continuous control tasks. For this study, we selected 11 standard continuous control environments from OpenAI Gym (Brockman et al., 2016) and Gymnasium (Towers et al., 2024).

**Meta-World.** Meta-World (Yu et al., 2021) is an open-source robotic manipulation benchmark designed for meta-reinforcement learning and multitask learning, containing 50 distinct tasks. Unlike AMAGO-2 (Grigsby et al., 2024b), which uses full 500-transition rollouts, we limited the maximum episode length to 100 timesteps, as in JAT (Gallouédec et al., 2024). This adjustment increases the context size, measured in terms of the number of episodes, without compromising task completion, as the shorter horizon remains sufficient to solve all tasks in the benchmark. Additionally, we fixed the goal state between environment resets—changing it only after re-initialization therefore ensuring compatibility with traditional memory-based Meta-RL formulations.

**Bi-DexHands.** Bi-DexHands (Chen et al., 2022) provides a set of bimanual manipulation tasks specifically designed for experiments in RL, MARL, offline RL, multi-task RL, and Meta-RL. The benchmark includes 20 complex high-dimensional continuous control tasks. Some subsets of tasks share the same action and observation space structures, making the environment suite compatible with the Meta-RL framework.

**Industrial-Benchmark.** Industrial-Benchmark (Hein et al., 2017) is a synthetic RL benchmark designed to simulate real-world industrial applications, such as controlling gas and wind turbines. The environment's transition dynamics and stochasticity can be adjusted using the setpoint parameter $p$. By increasing this parameter from 0 to 100 in steps of 5, we obtained 21 tasks sharing a unified state and action structure. We selected the standard reward, which was subsequently scaled down by a factor of 100.

**Kinetix.** Kinetix (Matthews et al., 2025) is an open-ended 2D physics-based control environment built on the hardware-accelerated Jax2D engine. It integrates a wide range of procedurally generated tasks—including locomotion, manipulation, and video-game-like scenarios—within a single framework. Tasks are defined by variations in objects, goals, and dynamics, and the authors distinguish three task scales—$s$, $m$, and $l$—to categorize different levels of complexity. The reward structure is standardized: agents are incentivized to bring a "green shape" into contact with a "blue shape" and penalized for collisions with "red shapes," with an auxiliary dense reward based on inter-shape distance. To ensure that the model infers the reward function implicitly from the provided context, color information was omitted from the observation.

**CityLearn.** CityLearn (Nweye et al., 2024) is an open-source reinforcement learning framework that provides multiple environments for long-term management of building energy resources and demand response in urban settings. It includes 24 tasks, each defined by continuous action and observation spaces with dimensionalities that vary according to the simulation configuration and the number of buildings. In this study, however, only the Phase 1 and Phase 2 tasks from the CityLearn 2022 Challenge (Khattar & Jin, 2022) were considered. Both tasks simulate one year of operational electricity demand and photovoltaic (PV) generation data from five single-family buildings in the Sierra Crest housing development in Fontana. To expand the set of environments and enable the model to process multiple episodes within its context, these tasks were further divided into 24 monthly tasks.

**HumEnv.** HumEnv (Tirinzoni et al., 2025) is a humanoid locomotion environment built on MuJoCo (Todorov et al., 2012). It was originally developed for training and evaluating Meta's Meta-Motivo model. The benchmark includes three categories of tasks: Tracking, Goal, and Reward. Tracking tasks require following a sequence of target observations, Goal tasks involve reaching a designated state (e.g., standing with hands raised), and Reward tasks consist of performing a specific activity (e.g., running, lying down). Although the state–action space remains identical across

all tasks, the reward functions differ. The environment offers a broad set of predefined tasks in a high-dimensional setting, making it well-suited for our study.

**Sinergym.** Sinergym (Campoy-Nieves et al., 2025) is a flexible benchmark designed for training reinforcement learning (RL) algorithms to reduce energy costs while maintaining indoor temperatures within a fixed range by managing energy-consuming devices such as fans, coolers, and heaters. By default, the framework provides an interface to simulate energy consumption across eight model buildings, each evaluated under three different weather profiles. The observation and action spaces are defined within a stochastic continuous domain. To shorten episode length, the default annual timeline is partitioned into months, which are grouped such that neighboring months are included within the same task. In some cases, however, months were separated because energy consumption depended more strongly on temporal factors than on model actions. Furthermore, certain combinations of building models, weather profiles, and months generated excessive noise and were therefore excluded.

**MetaDrive.** MetaDrive (Li et al., 2022) is a reinforcement learning (RL) suite of environments designed to simulate autonomous driving. In this setting, roads, obstacles, and other vehicles are generated randomly. MetaDrive provides a wide range of tasks, as its road segments can be rearranged in numerous configurations, substantially altering both the road layout and the difficulty level for the agent. However, the state and action spaces remain fixed.

**ControlGym.** ControlGym (Zhang et al., 2024b) is an open-source gym providing environments that range from linear systems to chaotic, large-scale systems governed by partial differential equations (PDEs). It offers extensive parameterization to facilitate evaluation under conditions that approximate real-world applications. The benchmark includes 46 continuous-control tasks, each with continuous action spaces and latent (hidden) states and observations of varying dimensionality. In this study, we focus on a subset of tasks: two linear tasks—Aircraft ($ac\{i\}$) and Cable ($cm\{i\}$)—and two linear PDE tasks—*convection_diffusion_reaction* and *schrodinger*.

## C.2 Train vs. Test Tasks Split

To evaluate our model's capabilities for inference-time optimization, we divided the full set of 255 tasks into two disjoint subsets. The validation subset contained 46 tasks, which were separated from the training dataset of 209 tasks. Below, we provide the details of the split for each domain.

### C.2.1 MuJoCo

Performance in the MuJoCo domain was evaluated on locomotion tasks with modified physical parameters altered through the provided XML API. For each task, we adjusted viscosity from 0 to 0.05 and 0.1, while gravity was varied by ±10 percent.

### C.2.2 Meta-World

We adopted the commonly used ML45 train–test split for Meta-RL setting, which includes 45 training tasks and 5 tasks reserved for validation: *bin-picking*, *box-close*, *door-lock*, *door-unlock*, and *hand-insert*.

### C.2.3 Bi-DexHands

The ML20 benchmark, proposed in the original paper (Chen et al., 2022), was selected as the train–test split. It assigns 15 tasks to the training set, with the remaining 5 reserved for validation: *door-close-outward*, *door-open-inward*, *door-open-outward*, *hand-kettle*, and *hand-over*. However, the *hand-over* task has a unique state–action space dimensionality that is not represented in the training set, making it incompatible with the multi-head encoder architecture. To ensure correct calculation of quality metrics for this domain, we conservatively report a performance of 0 (random) for this task.

### C.2.4 INDUSTRIAL-BENCHMARK

Training and testing tasks for Industrial-Benchmark were obtained by dividing the environments into two non-overlapping subsets based on the setpoint parameter: values from 0 to 75 defined the training set, while setpoints from 80 to 100 were assigned to the validation set.

### C.2.5 KINETIX

Since we obtained generalist agents only for tasks from the Kinetix test split (whereas the Kinetix training set consists of a large collection of randomly generated tasks), we could not adopt the train–test split described in the original paper (Matthews et al., 2025). Instead, we designated six tasks for evaluation: *h8_unicycle_balance* (S), *arm_hard* and *h14_thrustblock* (M), and *mjc_walker*, *hard_lunar_lander*, and *pinball_hard* (L). The remaining predefined tasks were assigned to the training set. The test split was selected to enable meaningful evaluation across scales: S includes a non-trivial balancing task, M involves environments requiring precise control and adaptability without color information, and L consists of tasks demanding advanced movement strategies and obstacle navigation. Overall, this selection emphasizes tasks that are challenging, non-trivial, and well-suited for assessing generalization and transfer.

### C.2.6 CITYLEARN

Since no predefined split exists for individual CityLearn tasks (Nweye et al., 2024), the last two monthly tasks from both Phase 1 and Phase 2 were assigned to the validation set, yielding training and validation sets of 20 and 4 tasks, respectively. This splitting strategy is appropriate for time-series data, as it enables effective evaluation of the ability of in-context reinforcement learning algorithms to generalize to varying dynamics.

### C.2.7 HUMENV

In the original study (Tirinzoni et al., 2025), only Reward and Goal tasks were used for evaluation, while training was conducted exclusively on Tracking tasks, in line with MetaMotivo's architecture, which was designed to learn state representations. Based on task diversity and expert convergence, we selected 10 Reward tasks and 5 Goal tasks for training and reserved 3 tasks for testing: *t-pose* (Goal), *rotate-x-5-0.8* (Reward), and *split-0.5* (Reward).

### C.2.8 SINERGYM

Following the approach proposed by (Manjavacas et al., 2024), training and testing tasks were defined using different weather conditions for the same building types. Specifically, tasks with hot or cool weather profiles were assigned to the training set, while mixed-weather tasks were allocated to the test set. The final split consisted of 22 training tasks and 11 testing tasks.

### C.2.9 METADRIVE

The original paper (Li et al., 2022) did not specify a train/test split; therefore, we defined one. In total, we selected 21 tasks: 16 for training and 5 for testing. The MetaDrive environment generates maps using various types of road segments, including straight sections, curves, circles, and sharp turns. Since the exact configuration of each segment (e.g., curvature or placement) changes with every reset, we applied seeding to ensure that each task remained deterministic. However, the random selection of the lane in which the agent spawns was preserved.

### C.2.10 CONTROLGYM

ControlGym (Zhang et al., 2024b) does not provide a predefined train–test split by default. We therefore propose a split based on (1) state complexity (dimensionality) and (2) the robustness of demonstrator performance in terms of total reward. Specifically, three of the 12 environments were selected for validation: *ac1*, *cm5*, and *schrodinger*. When constructing the test set, we prioritized environments that have counterparts with the same action–observation space remaining in the training set, enabling meaningful assessments of transfer.

# D    DEMONSTRATORS

## D.1    TRAINING

**MuJoCo.**    For MuJoCo (Todorov et al., 2012), demonstrators were trained using behavioral cloning on the dataset provided by JAT (Gallouédec et al., 2024). The resulting demonstrators achieved performance levels consistent with those reported in the JAT paper.

**Meta-World.**    We used trained agents open-sourced by Gallouédec et al. (2024) as demonstrators for the Meta-World (Yu et al., 2021) benchmark. However, further analysis revealed that some agents achieved unsatisfactory success rates. To address this issue, we retrained demonstrators for the following tasks: *disassemble*, *coffee-pull*, *coffee-push*, *soccer*, *push-back*, *peg-insert-side*, and *pick-out-of-hole*. For efficient fine-tuning and hyperparameter search, we used the training scripts provided by JAT (Gallouédec et al., 2024). Re-training improved performance on the aforementioned tasks, although some demonstrators continued to exhibit unstable results.

**Bi-DexHands.**    PPO (Schulman et al., 2017) implementation provided by the authors (Chen et al., 2022) was used to obtain demonstrators. We increased the number of parallel IsaacGym (Liang et al., 2018) environments from 128 to 2048 to improve convergence, following recent empirical evidence (Mayor et al., 2025). For certain tasks, namely *Re-Orientation* and *Swing-Cup*, we were able to substantially exceed the performance reported in the original work. However, for some environments, the RL agents continued to exhibit stochastic performance even after training for over 1.5 billion timesteps.

**Industrial-Benchmark.**    The implementation of PPO from Stable-Baselines3 (Raffin et al., 2021) was used to train demonstrators. For all runs, we applied advantage normalization, set a KL-divergence limit of 0.2, used a discount factor of 0.97, 2500 environment steps, a batch size of 50, and a training duration of 1 million timesteps. To increase episodic context size and facilitate score comparability, we limited the episode length to 250 transitions. Additionally, we used delta-rewards for training our RL agents, as this reward formulation demonstrated superior performance compared to agents trained with scaled classic rewards, based on validation results for both types of returns.

**Kinetix.**    For each task scale ($s$, $m$, and $l$), we first trained a generalist agent using PPO (Schulman et al., 2017) with entity-type observations. The agents for scales S, M, and L were trained for 1B, 2B, and 3B timesteps, respectively, with a batch size of 12288 and 512 rollout steps. Hyperparameters were further tuned to increase success rates on procedurally generated random morphologies during training. Each generalist agent was then fine-tuned to solve individual tasks from the Kinetix test split. This process yielded reliable demonstrators for 48 test tasks across all scales. During data collection, observations were stored as real-valued vectors, resulting in three groups corresponding to scales S, M, and L.

**CityLearn.**    Demonstrators were trained using the Linear Programming (LP) reduction method proposed by the winning team of the CityLearn 2022 Challenge, Team Together (Nweye et al., 2022). The CPLEX solver from IBM ILOG (Manual, 1987) was used to compute the optimal monthly dispatch, which converged rapidly for each task. For both the Phase 1 and Phase 2 task sets, the average performance of the demonstrators matched or exceeded the best scores reported on the challenge leaderboard. When evaluated against the demonstrators on a monthly basis, the top three challenge solutions achieved comparable or lower scores.

**HumEnv.**    All demonstrators were trained using TD3 (Fujimoto et al., 2018), with MLPs serving as the Actor and Critic in accordance with the original configuration (with one fewer layer for certain tasks). Each network was trained for 30 million steps, consistent with the original paper. The resulting rewards were generally close to those reported in (Tirinzoni et al., 2025) across different tasks. The humanoid's initial state was determined by a set of motions and a fall probability, with fall positions sampled as defined in the authors' repository. Consequently, the initial state was either a fall or a random frame taken from a randomly selected motion. In all models, the original test motion set was used as the initial one, with a fall probability of 0.2.

**Sinergym.** As baseline algorithms, SAC (Haarnoja et al., 2018), TD3 (Fujimoto et al., 2018), and PPO (Schulman et al., 2017) were taken from the Stable-Baselines3 framework (Raffin et al., 2021) and trained on each task using the global parameters reported by (Campoy-Nieves et al., 2025). Each model was trained for approximately 0.5 million time steps per task. Overall, the three algorithms demonstrated comparable performance, but PPO achieved higher rewards on a greater number of tasks. Owing to its stability and consistent results, PPO was selected as the primary demonstrator for the Sinergym tasks.

**MetaDrive.** For our demonstrators, we used the expert policy provided by the authors (Li et al., 2022), which was trained with PPO (Schulman et al., 2017) on the generalization environment that procedurally generates road layouts in MetaDrive. This policy was subsequently fine-tuned on circle maps to improve performance. Since the PPO policy could not reliably avoid static obstacles, we adopted the authors' rule-based algorithm for maps containing such obstacles. Accordingly, three demonstrators were used: (1) PPO policy fine-tuned for 10 million steps on circle maps, achieving a 95% success rate; (2) PPO policy fine-tuned for 10 million steps on maps without circles or obstacles, achieving a 98% success rate; and (3) the rule-based algorithm, which achieved a 98% success rate on maps with obstacles.

**ControlGym.** For each task, we selected a demonstrator from three built-in heuristic controllers. These controllers access the environment's unobservable internal state to select actions, and the choice for each task was based on empirical performance. For most tasks, we used the $H_2/H_\infty$ controller derived from the block generalized algebraic Riccati equation (GARE) (Zhang et al., 2024b). Exceptions include the linear PDEs, for which we used LQR because $H_2/H_\infty$ is incompatible with these settings; *ac1* and *ac6*, for which we used LQR and LQG, respectively, both based on discrete-time state-space models (Zhang et al., 2024b). We modified $H_2/H_\infty$ to use Moore–Penrose pseudoinverses when matrix inverses were singular or ill-conditioned. Nevertheless, none of the controllers achieved stable performance on *ac3*, *ac7*, *ac8*, *ac9*, and *ac10*; these environments were therefore excluded from the target task set. We also trained PPO (Schulman et al., 2017) and SAC (Haarnoja et al., 2018) on these tasks, but neither achieved stable performance comparable to LQR or LQG.

## E HYPERPARAMETERS

| Hyperparameter | Value |
|---|---|
| Action Decoder Steps | 32 |
| Learning Rate | 0.00005 |
| Gradient Clipping Norm | 2.5 |
| Optimizer | Adam |
| Beta 1 | 0.9 |
| Beta 2 | 0.99 |
| Batch Size | 64 |
| Gradient Accumulation Steps | 1 |
| Transformer Layers | 16 |
| Transformer Heads | 24 |
| Context Length | 4096 |
| Transformer Hidden Dim | 1536 |
| FF Hidden Size | 6144 |
| MLP Type | GptNeoxMLP |
| Normalization Type | LayerNorm |
| Training Precision | bf16 |
| Parameters | 928053513 |

Table 2: Hyperparameter configuration

# F    ACTION BELIEFS

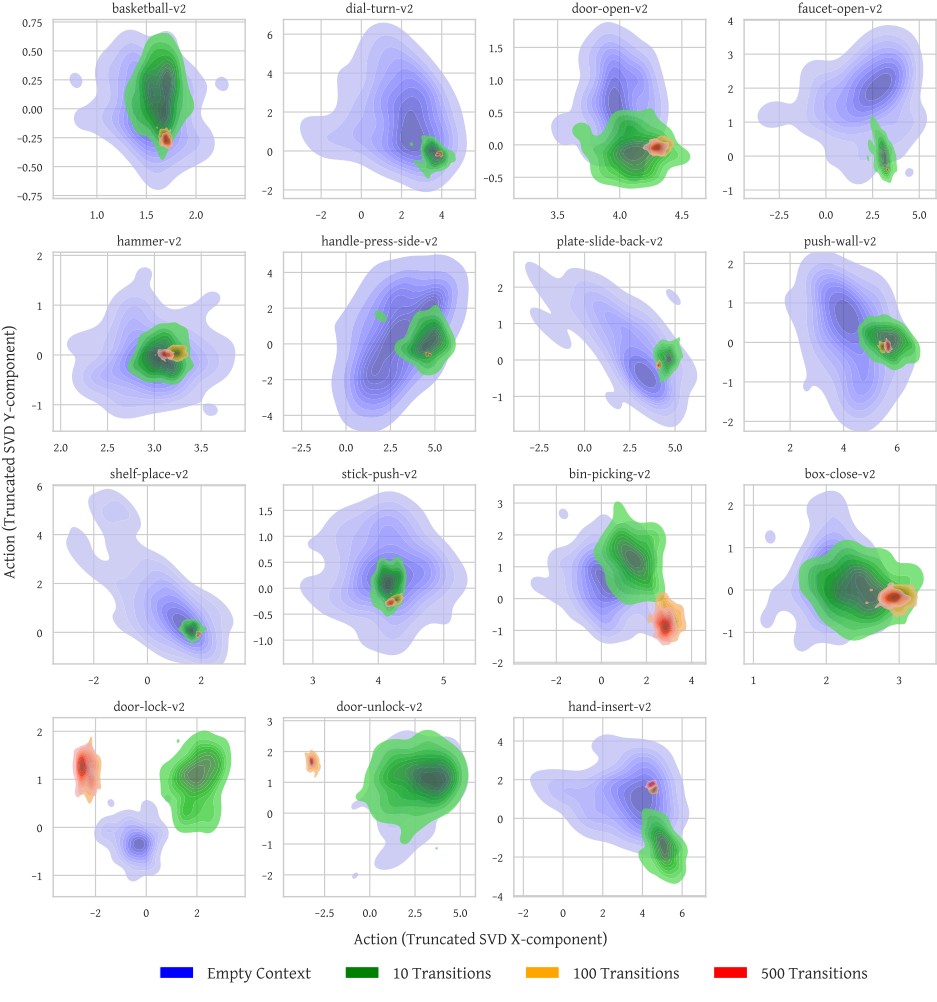

Figure 6: Action beliefs over context sizes for other tasks in Meta-World

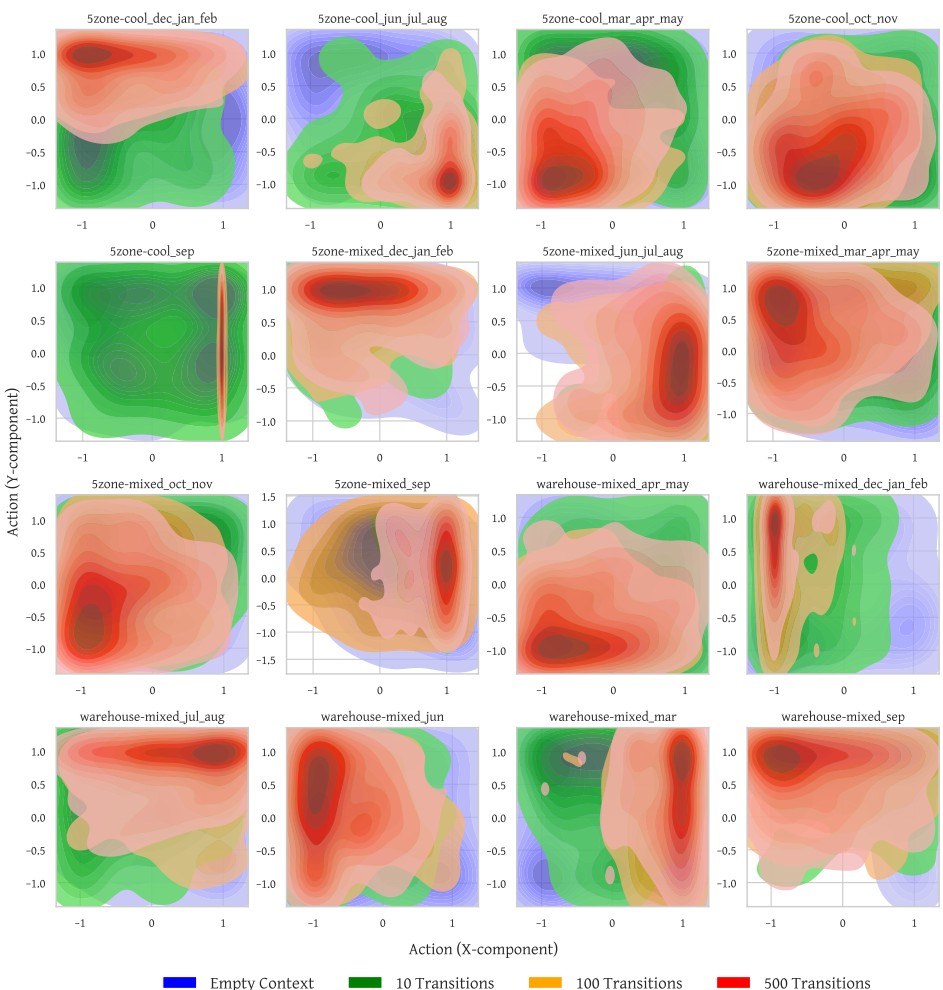

Figure 7: Action beliefs over context sizes for other tasks in SinerGym

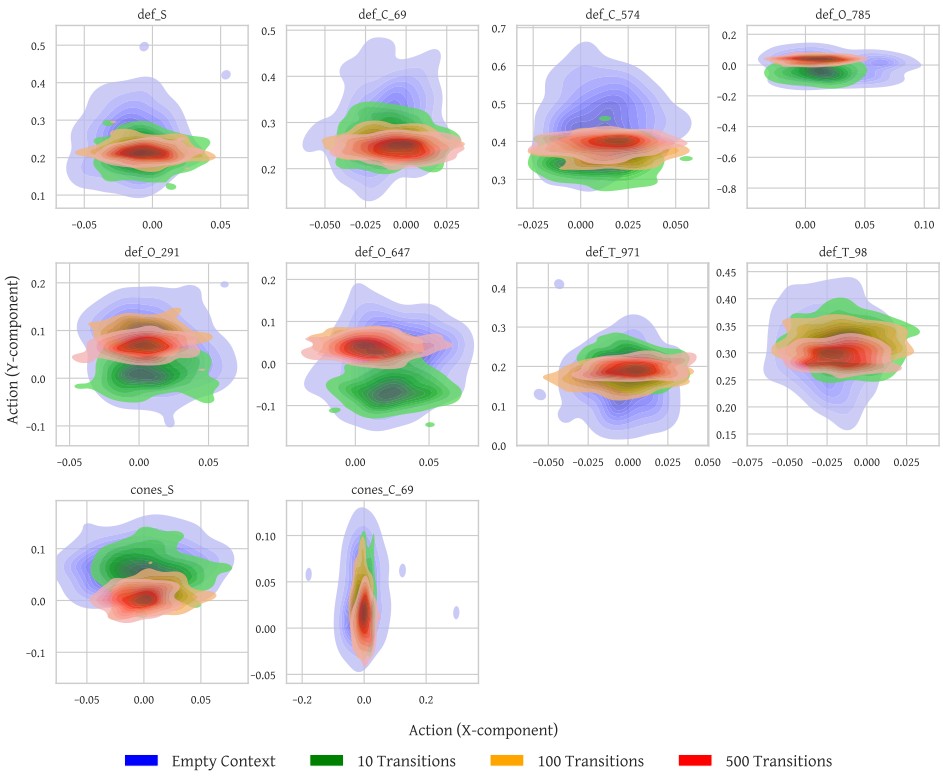

Figure 8: Action beliefs over context sizes for other tasks in MetaDrive

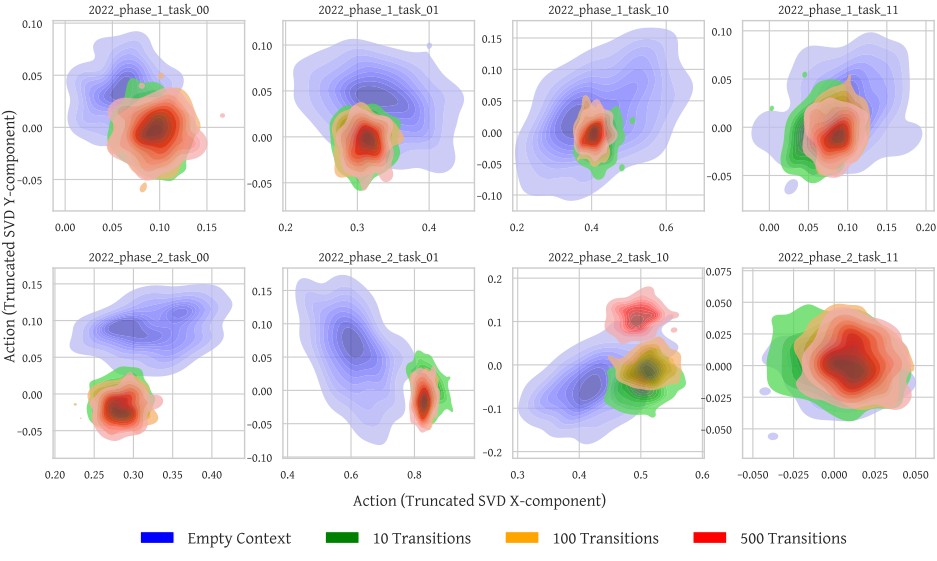

Figure 9: Action beliefs over context sizes for other tasks in CityLearn

# G    TASK-LEVEL DATASET VISUALIZATION

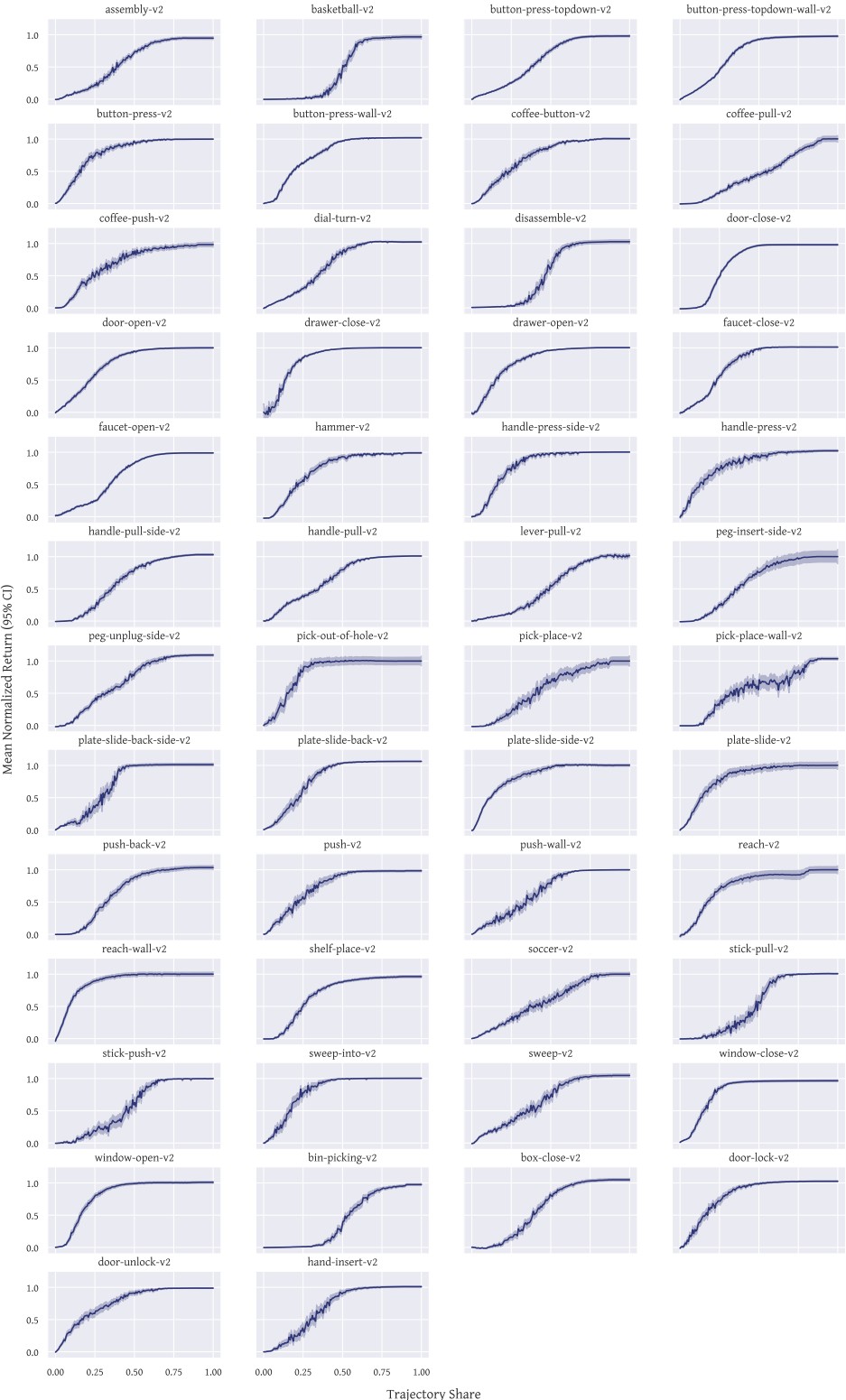

Figure 10: Mean normalized noise-distilled trajectories for Meta-World domain

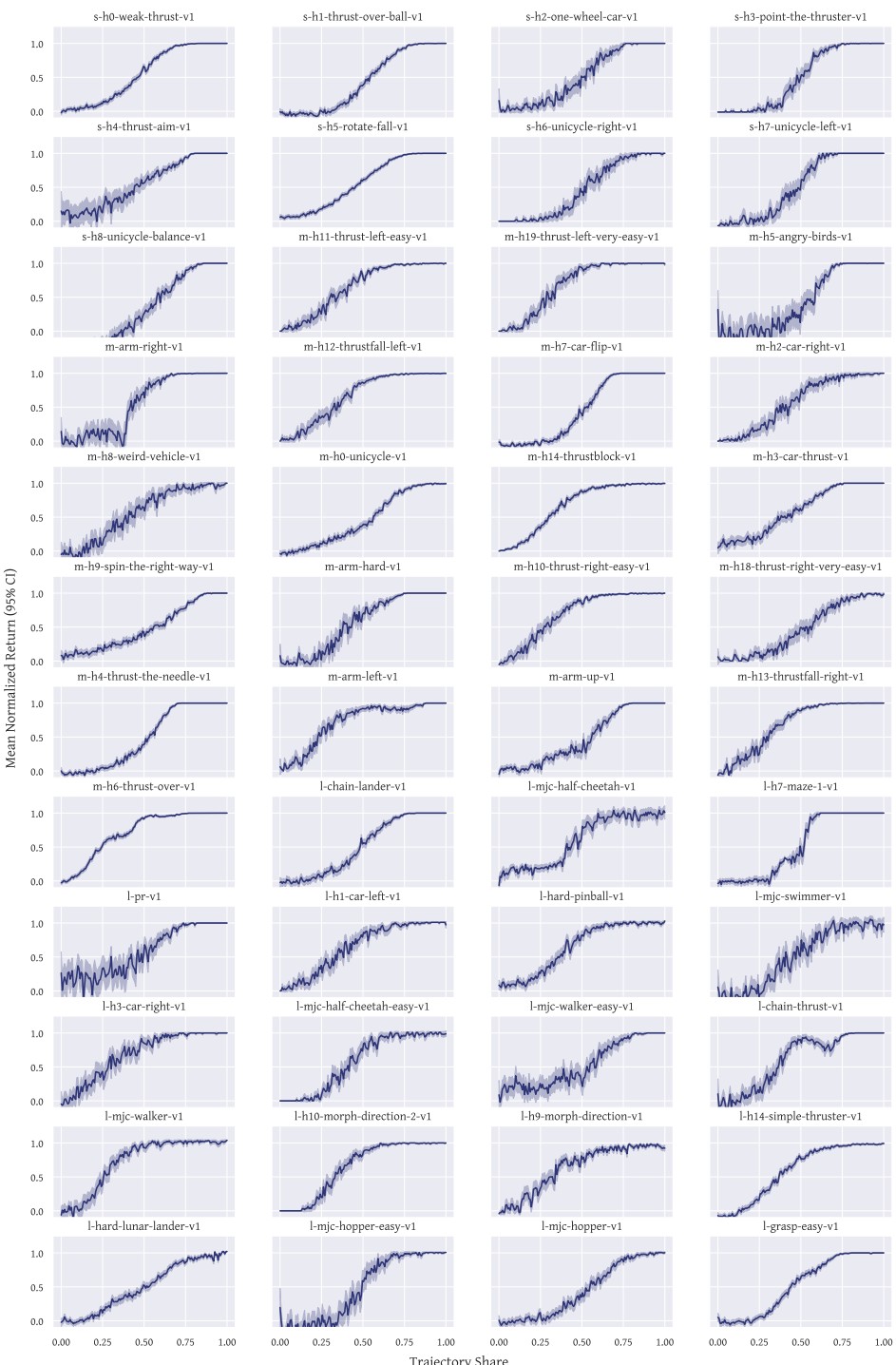

Figure 11: Mean normalized noise-distilled trajectories for Kinetix domain

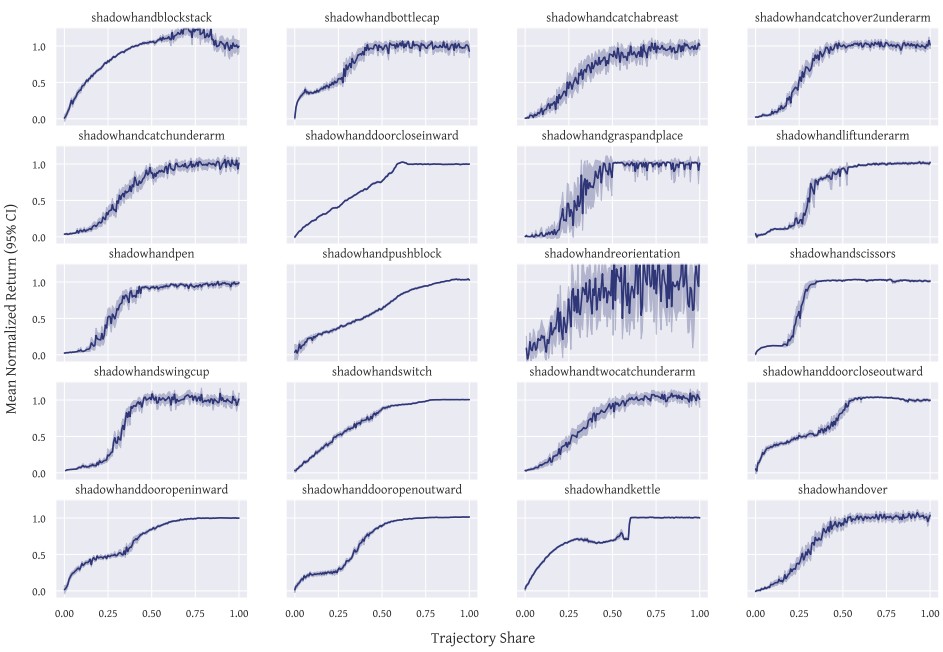

Figure 12: Mean normalized noise-distilled trajectories for Bi-DexHands domain

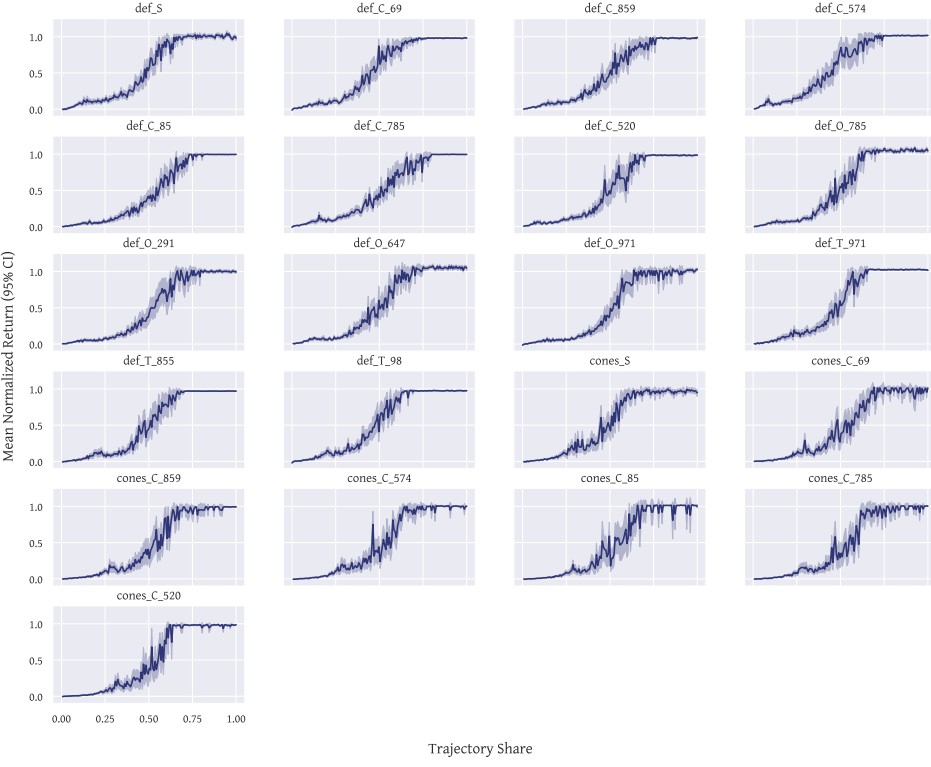

Figure 13: Mean normalized noise-distilled trajectories for MetaDrive domain

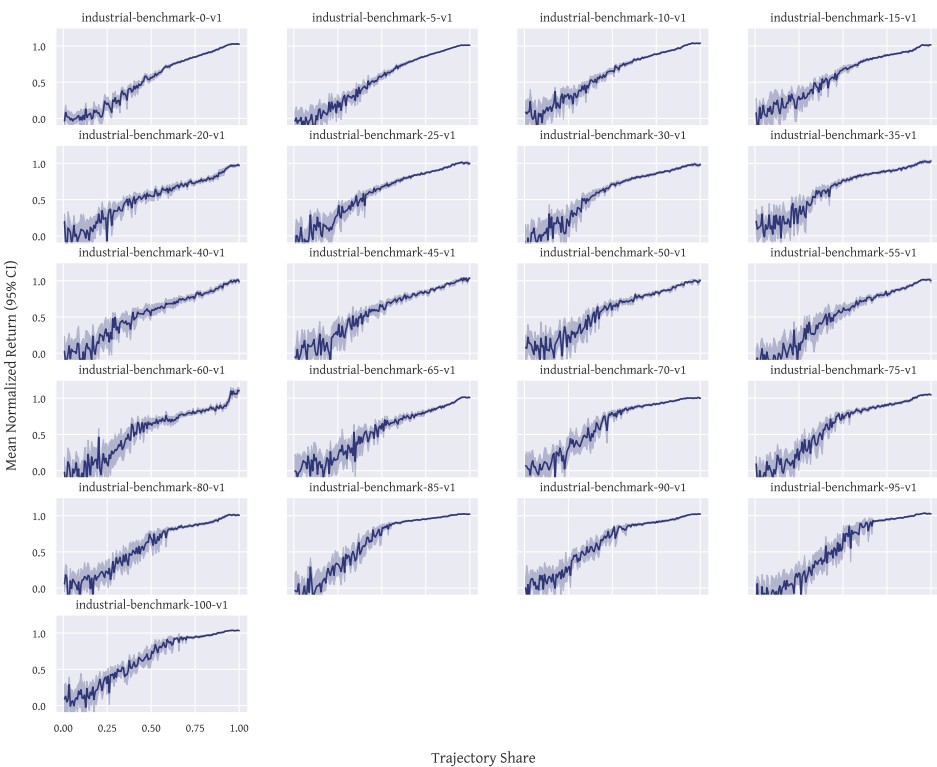

Figure 14: Mean normalized noise-distilled trajectories for Industrial-Benchmark domain

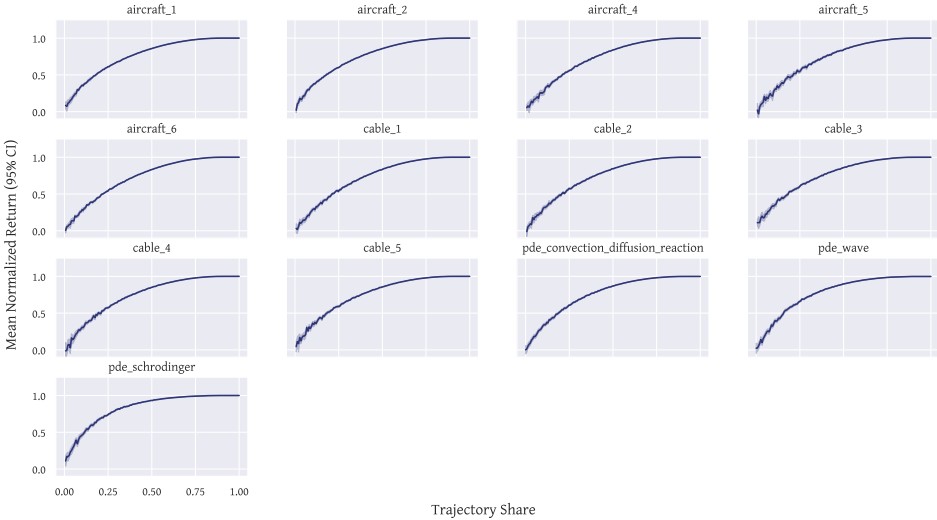

Figure 15: Mean normalized noise-distilled trajectories for ControlGym domain

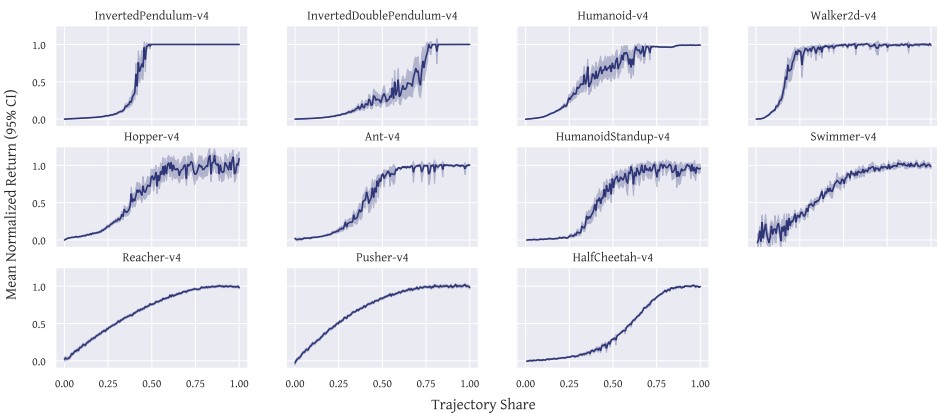

Figure 16: Mean normalized noise-distilled trajectories for MuJoCo domain

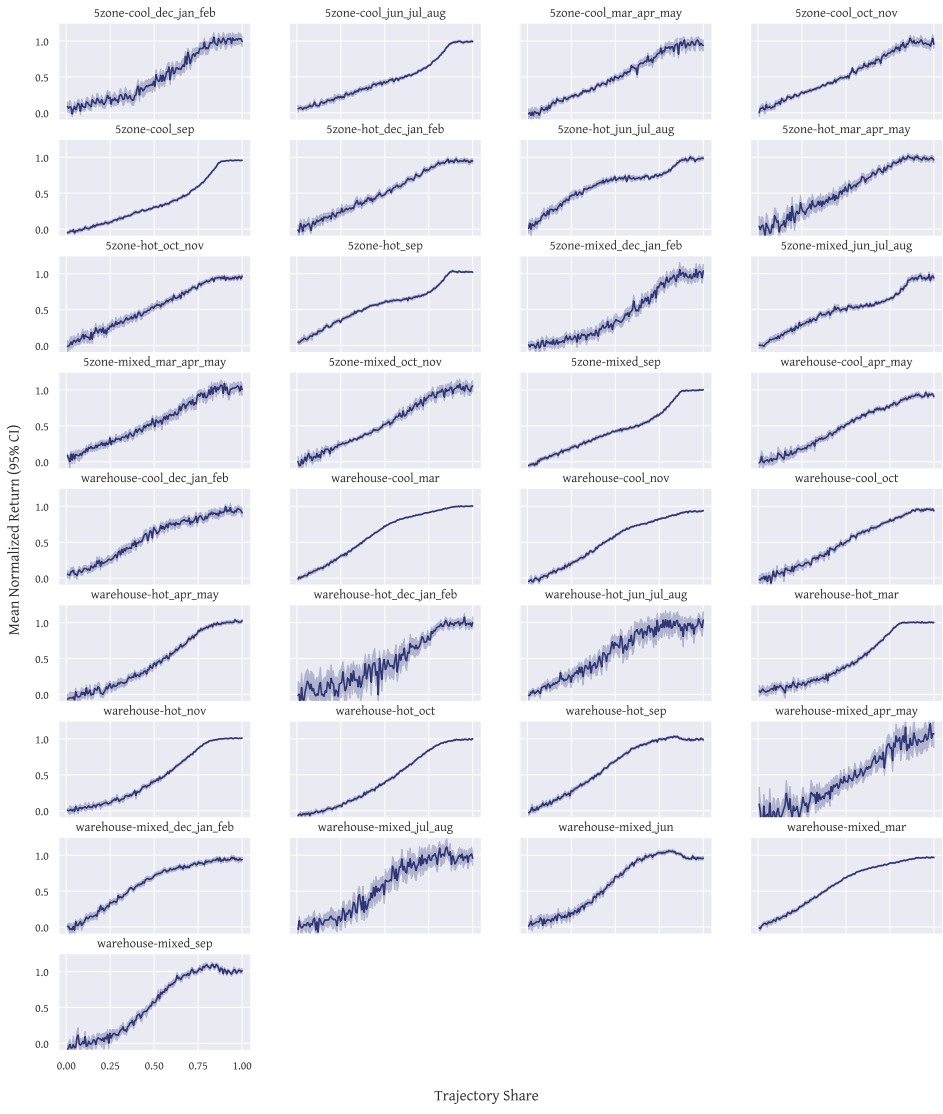

Figure 17: Mean normalized noise-distilled trajectories for SinerGym domain

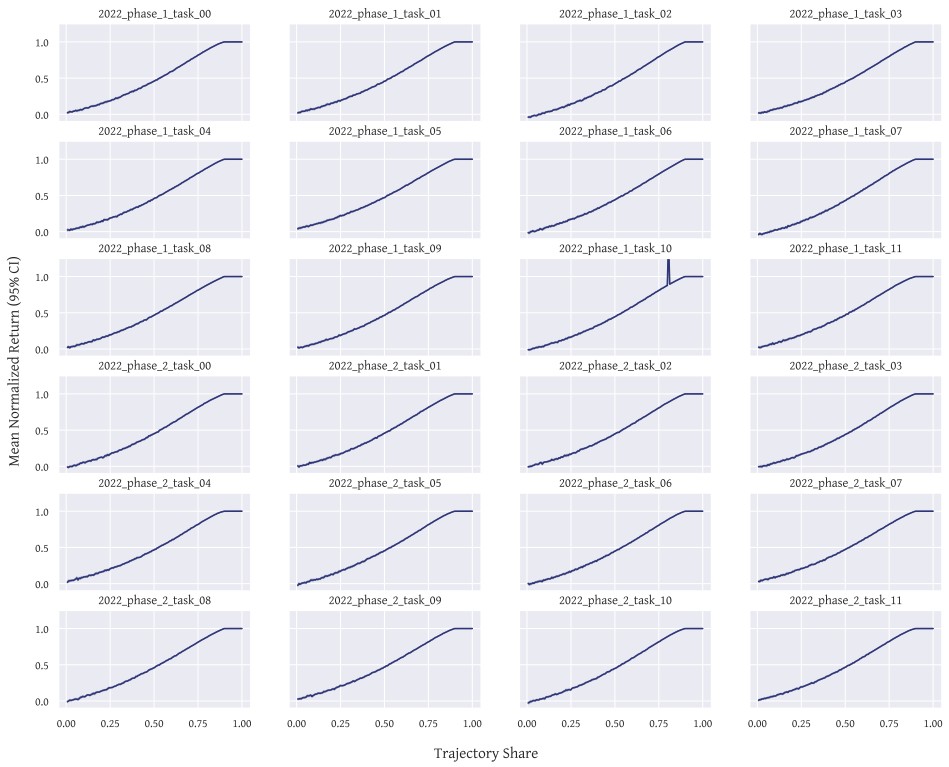

Figure 18: Mean normalized noise-distilled trajectories for CityLearn domain

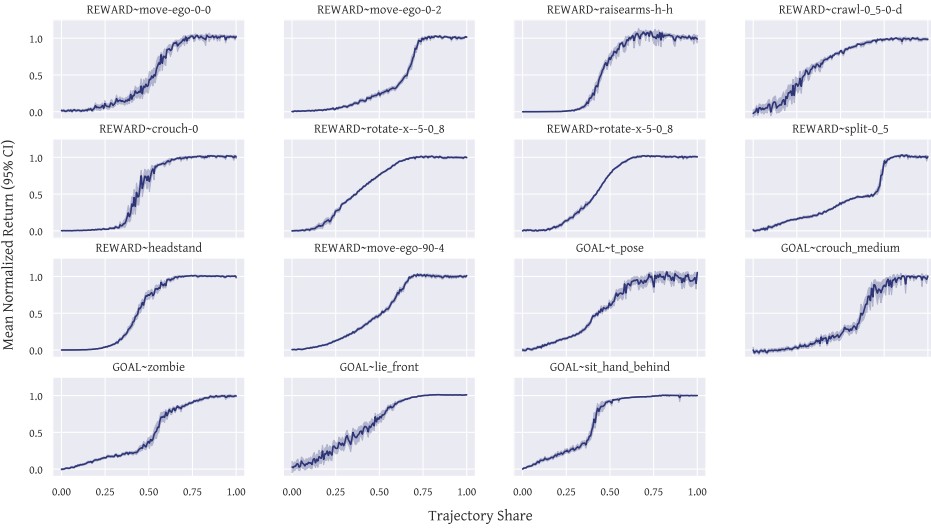

Figure 19: Mean normalized noise-distilled trajectories for HumEnv domain

# H  TASK-LEVEL PERFORMANCE

## H.1  COLD-START ONLINE

| Task Name | Split | Random Score | Expert Score | Normalized Return IQM |
|---|---|---|---|---|
| bin-picking-v2 | test | 1.9 | 410.0 ± 64.44 | 0.0 ± 0.0 |
| box-close-v2 | test | 82.0 | 583.9 ± 13.3 | -0.0 ± 0.0 |
| door-lock-v2 | test | 119.3 | 831.9 ± 9.22 | 0.6 ± 0.05 |
| door-unlock-v2 | test | 94.7 | 813.0 ± 2.07 | 0.2 ± 0.03 |
| hand-insert-v2 | test | 2.5 | 775.5 ± 5.46 | 0.1 ± 0.04 |
| assembly-v2 | train | 44.7 | 283.3 ± 27.63 | 1.0 ± 0.05 |
| basketball-v2 | train | 2.6 | 591.7 ± 19.84 | 1.0 ± 0.02 |
| button-press-topdown-v2 | train | 28.6 | 499.0 ± 31.24 | 0.5 ± 0.01 |
| button-press-topdown-wall-v2 | train | 30.1 | 508.4 ± 35.03 | 0.6 ± 0.12 |
| button-press-v2 | train | 31.5 | 646.8 ± 11.04 | 0.9 ± 0.17 |
| button-press-wall-v2 | train | 8.0 | 679.6 ± 9.74 | 0.7 ± 0.32 |
| coffee-button-v2 | train | 33.0 | 733.0 ± 29.68 | 1.0 ± 0.14 |
| coffee-pull-v2 | train | 4.2 | 349.8 ± 104.95 | 1.0 ± 0.03 |
| coffee-push-v2 | train | 4.2 | 570.1 ± 105.82 | 0.0 ± 0.01 |
| dial-turn-v2 | train | 29.8 | 794.2 ± 13.12 | 1.0 ± 0.01 |
| disassemble-v2 | train | 33.9 | 486.8 ± 88.9 | 0.2 ± 0.14 |
| door-close-v2 | train | 5.0 | 543.7 ± 34.04 | 1.0 ± 0.0 |
| door-open-v2 | train | 54.2 | 587.5 ± 20.2 | 1.0 ± 0.0 |
| drawer-close-v2 | train | 107.6 | 866.2 ± 4.77 | 1.0 ± 0.05 |
| drawer-open-v2 | train | 134.0 | 493.9 ± 2.1 | 1.0 ± 0.0 |
| faucet-close-v2 | train | 247.5 | 767.0 ± 8.36 | 1.0 ± 0.0 |
| faucet-open-v2 | train | 249.5 | 757.1 ± 8.85 | 1.0 ± 0.01 |
| hammer-v2 | train | 90.3 | 683.8 ± 23.29 | 1.0 ± 0.0 |
| handle-press-side-v2 | train | 53.6 | 843.9 ± 11.33 | 1.0 ± 0.01 |
| handle-press-v2 | train | 54.0 | 809.1 ± 53.55 | 1.0 ± 0.16 |
| handle-pull-side-v2 | train | 2.3 | 494.4 ± 49.06 | 0.7 ± 0.34 |
| handle-pull-v2 | train | 14.1 | 709.7 ± 5.59 | 0.3 ± 0.09 |
| lever-pull-v2 | train | 61.4 | 608.6 ± 40.05 | 0.6 ± 0.08 |
| peg-insert-side-v2 | train | 1.7 | 314.5 ± 153.35 | 0.0 ± 0.06 |
| peg-unplug-side-v2 | train | 5.8 | 474.3 ± 89.02 | 0.2 ± 0.31 |
| pick-out-of-hole-v2 | train | 1.5 | 381.2 ± 28.75 | -0.0 ± 0.0 |
| pick-place-v2 | train | 1.1 | 473.7 ± 122.34 | 0.1 ± 0.18 |
| pick-place-wall-v2 | train | 0.0 | 540.6 ± 86.62 | 0.4 ± 0.23 |
| plate-slide-back-side-v2 | train | 33.7 | 743.9 ± 60.52 | 0.8 ± 0.23 |
| plate-slide-back-v2 | train | 34.2 | 679.4 ± 78.11 | 1.0 ± 0.01 |
| plate-slide-side-v2 | train | 23.4 | 672.9 ± 130.16 | 1.0 ± 0.1 |
| plate-slide-v2 | train | 72.6 | 525.5 ± 154.19 | 0.9 ± 0.13 |
| push-back-v2 | train | 1.2 | 464.4 ± 92.37 | 0.0 ± 0.0 |
| push-v2 | train | 5.3 | 762.0 ± 35.22 | 1.0 ± 0.09 |
| push-wall-v2 | train | 6.2 | 746.4 ± 16.4 | 1.0 ± 0.0 |
| reach-v2 | train | 140.0 | 639.2 ± 142.86 | 0.8 ± 0.07 |
| reach-wall-v2 | train | 157.0 | 601.9 ± 140.25 | 1.2 ± 0.33 |
| shelf-place-v2 | train | 0.0 | 261.1 ± 22.94 | 0.9 ± 0.13 |
| soccer-v2 | train | 4.9 | 483.6 ± 48.56 | 0.2 ± 0.18 |
| stick-pull-v2 | train | 2.7 | 519.2 ± 9.43 | 0.9 ± 0.01 |
| stick-push-v2 | train | 2.7 | 635.9 ± 7.86 | 1.0 ± 0.0 |
| sweep-into-v2 | train | 16.3 | 795.2 ± 11.04 | -0.0 ± 0.12 |
| sweep-v2 | train | 12.5 | 499.6 ± 22.37 | 0.7 ± 0.24 |
| window-close-v2 | train | 56.9 | 594.2 ± 38.3 | 0.8 ± 0.29 |
| window-open-v2 | train | 43.2 | 618.4 ± 46.16 | 0.7 ± 0.19 |

Table 3: Online Evaluation Results for Meta-World

| Task Name | Split | Random Score | Expert Score | Normalized Return IQM |
|---|---|---|---|---|
| l-hard-lunar-lander-v1 | test | -0.4 | 1.0 ± 0.0 | 0.3 ± 0.24 |
| l-hard-pinball-v1 | test | -0.6 | 1.0 ± 0.0 | 0.1 ± 0.28 |
| l-mjc-walker-v1 | test | -0.3 | 1.0 ± 0.0 | 0.2 ± 0.0 |
| m-arm-hard-v1 | test | 0.0 | 1.0 ± 0.0 | 1.0 ± 0.0 |
| m-h14-thrustblock-v1 | test | -1.0 | 1.0 ± 0.0 | 0.0 ± 0.0 |
| s-h8-unicycle-balance-v1 | test | 0.4 | 1.0 ± 0.0 | 0.4 ± 0.5 |
| l-chain-lander-v1 | train | -0.3 | 1.0 ± 0.0 | 1.0 ± 0.0 |
| l-chain-thrust-v1 | train | 0.1 | 1.0 ± 0.0 | 1.0 ± 0.0 |
| l-grasp-easy-v1 | train | -0.4 | 1.0 ± 0.0 | 1.0 ± 0.01 |
| l-h1-car-left-v1 | train | 0.0 | 1.0 ± 0.0 | 0.3 ± 0.25 |
| l-h10-morph-direction-2-v1 | train | 0.0 | 1.0 ± 0.0 | 1.0 ± 0.0 |
| l-h14-simple-thruster-v1 | train | -0.4 | 1.0 ± 0.0 | 1.0 ± 0.0 |
| l-h3-car-right-v1 | train | 0.2 | 1.0 ± 0.0 | 1.0 ± 0.0 |
| l-h7-maze-1-v1 | train | -0.5 | 1.0 ± 0.0 | 0.6 ± 0.19 |
| l-h9-morph-direction-v1 | train | 0.0 | 1.0 ± 0.0 | 0.6 ± 0.2 |
| l-mjc-half-cheetah-easy-v1 | train | 0.0 | 1.0 ± 0.0 | 1.0 ± 0.0 |
| l-mjc-half-cheetah-v1 | train | -0.3 | 0.9 ± 0.0 | 0.3 ± 0.04 |
| l-mjc-hopper-easy-v1 | train | 0.3 | 1.0 ± 0.0 | 1.0 ± 0.13 |
| l-mjc-hopper-v1 | train | -0.3 | 1.0 ± 0.0 | 0.3 ± 0.18 |
| l-mjc-swimmer-v1 | train | 0.2 | 0.9 ± 0.0 | 0.8 ± 0.29 |
| l-mjc-walker-easy-v1 | train | 0.1 | 1.0 ± 0.0 | -0.2 ± 0.09 |
| l-pr-v1 | train | 0.6 | 1.0 ± 0.0 | 1.0 ± 0.0 |
| m-arm-left-v1 | train | 0.0 | 1.0 ± 0.0 | 1.0 ± 0.0 |
| m-arm-right-v1 | train | 0.1 | 1.0 ± 0.0 | 1.0 ± 0.0 |
| m-arm-up-v1 | train | -0.5 | 1.0 ± 0.0 | 1.0 ± 0.0 |
| m-h0-unicycle-v1 | train | -0.4 | 1.0 ± 0.0 | 1.0 ± 0.0 |
| m-h10-thrust-right-easy-v1 | train | 0.0 | 1.0 ± 0.0 | 0.8 ± 0.17 |
| m-h11-thrust-left-easy-v1 | train | 0.0 | 1.0 ± 0.0 | 1.0 ± 0.0 |
| m-h12-thrustfall-left-v1 | train | 0.0 | 1.0 ± 0.0 | 1.0 ± 0.0 |
| m-h13-thrustfall-right-v1 | train | 0.1 | 1.0 ± 0.0 | 1.0 ± 0.0 |
| m-h18-thrust-right-very-easy-v1 | train | 0.0 | 1.0 ± 0.0 | 1.0 ± 0.0 |
| m-h19-thrust-left-very-easy-v1 | train | 0.0 | 1.0 ± 0.0 | 1.0 ± 0.0 |
| m-h2-car-right-v1 | train | 0.0 | 1.0 ± 0.0 | 1.0 ± 0.0 |
| m-h3-car-thrust-v1 | train | -0.5 | 1.0 ± 0.0 | 1.0 ± 0.0 |
| m-h4-thrust-the-needle-v1 | train | -0.7 | 1.0 ± 0.0 | 1.0 ± 0.0 |
| m-h5-angry-birds-v1 | train | 0.4 | 1.0 ± 0.0 | 0.6 ± 0.16 |
| m-h6-thrust-over-v1 | train | -0.5 | 1.0 ± 0.0 | 1.0 ± 0.0 |
| m-h7-car-flip-v1 | train | -0.4 | 1.0 ± 0.0 | 1.0 ± 0.0 |
| m-h8-weird-vehicle-v1 | train | 0.2 | 1.0 ± 0.0 | 1.0 ± 0.0 |
| m-h9-spin-the-right-way-v1 | train | -0.3 | 1.0 ± 0.0 | 1.0 ± 0.0 |
| s-h0-weak-thrust-v1 | train | -0.5 | 1.0 ± 0.0 | 1.0 ± 0.0 |
| s-h1-thrust-over-ball-v1 | train | -0.3 | 1.0 ± 0.0 | 1.0 ± 0.0 |
| s-h2-one-wheel-car-v1 | train | 0.0 | 1.0 ± 0.0 | 1.0 ± 0.0 |
| s-h3-point-the-thruster-v1 | train | 0.0 | 1.0 ± 0.0 | 1.0 ± 0.0 |
| s-h4-thrust-aim-v1 | train | 0.3 | 1.0 ± 0.0 | 1.0 ± 0.0 |
| s-h5-rotate-fall-v1 | train | -0.6 | 1.0 ± 0.0 | 1.0 ± 0.0 |
| s-h6-unicycle-right-v1 | train | 0.0 | 1.0 ± 0.0 | 1.0 ± 0.0 |
| s-h7-unicycle-left-v1 | train | 0.1 | 1.0 ± 0.0 | 1.0 ± 0.0 |

Table 4: Online Evaluation Results for Kinetix

| Task Name | Split | Random Score | Expert Score | Normalized Return IQM |
|---|---|---|---|---|
| Ant-v4 | param_shift | -102.9 | 4989.7 ± 2058.04 | 1.0 ± 0.01 |
| HalfCheetah-v4 | param_shift | -264.2 | 7331.8 ± 420.5 | 1.0 ± 0.01 |
| Hopper-v4 | param_shift | 18.2 | 2264.3 ± 229.33 | 1.0 ± 0.47 |
| Humanoid-v4 | param_shift | 121.9 | 7373.8 ± 132.4 | 1.0 ± 0.0 |
| HumanoidStandup-v4 | param_shift | 33027.7 | 303552.9 ± 19279.02 | 1.0 ± 0.03 |
| InvertedDoublePendulum-v4 | param_shift | 58.5 | 9256.8 ± 293.62 | 1.0 ± 0.0 |
| InvertedPendulum-v4 | param_shift | 5.2 | 222.2 ± 9.37 | 0.8 ± 0.23 |
| Pusher-v4 | param_shift | -146.8 | -23.8 ± 0.57 | 1.0 ± 0.02 |
| Reacher-v4 | param_shift | -41.9 | -5.8 ± 1.06 | 1.0 ± 0.01 |
| Swimmer-v4 | param_shift | -0.3 | 95.3 ± 1.28 | 1.0 ± 0.02 |
| Walker2d-v4 | param_shift | 3.3 | 5169.3 ± 106.04 | 1.0 ± 0.01 |
| Ant-v4 | train | -58.0 | 6420.5 ± 17.13 | 1.0 ± 0.01 |
| HalfCheetah-v4 | train | -267.2 | 7786.1 ± 16.1 | 1.0 ± 0.01 |
| Hopper-v4 | train | 14.0 | 3390.4 ± 17.78 | 1.0 ± 0.16 |
| Humanoid-v4 | train | 117.9 | 7521.7 ± 12.92 | 1.0 ± 0.0 |
| HumanoidStandup-v4 | train | 34034.7 | 299534.8 ± 2187.55 | 1.1 ± 0.02 |
| InvertedDoublePendulum-v4 | train | 61.7 | 9359.5 ± 0.11 | 1.0 ± 0.0 |
| InvertedPendulum-v4 | train | 5.7 | 1000.0 ± 0.0 | 1.0 ± 0.0 |
| Pusher-v4 | train | -149.8 | -24.7 ± 1.61 | 1.0 ± 0.05 |
| Reacher-v4 | train | -43.0 | -5.6 ± 0.73 | 1.0 ± 0.02 |
| Swimmer-v4 | train | 1.7 | 96.3 ± 0.96 | 1.0 ± 0.03 |
| Walker2d-v4 | train | 2.7 | 5357.3 ± 13.14 | 1.0 ± 0.02 |

Table 5: Online Evaluation Results for MuJoCo

| Task Name | Split | Random Score | Expert Score | Normalized Return IQM |
|---|---|---|---|---|
| shadowhanddoorcloseoutward | test | 930.0 | 1380.0 ± 0.0 | -1.3 ± 0.01 |
| shadowhanddooropeninward | test | -15.0 | 410.0 ± 0.0 | 0.3 ± 0.03 |
| shadowhanddooropenoutward | test | 5.0 | 610.0 ± 0.0 | 0.1 ± 0.0 |
| shadowhandkettle | test | -200.0 | 53.0 ± 0.0 | -1.0 ± 0.01 |
| shadowhandblockstack | train | 90.0 | 285.0 ± 0.0 | 1.0 ± 0.09 |
| shadowhandbottlecap | train | 100.0 | 400.0 ± 0.0 | 1.1 ± 0.07 |
| shadowhandcatchabreast | train | 0.5 | 67.0 ± 0.0 | -0.0 ± 0.0 |
| shadowhandcatchover2underarm | train | 4.0 | 34.0 ± 0.0 | 1.0 ± 0.33 |
| shadowhandcatchunderarm | train | 0.8 | 25.0 ± 0.0 | 0.6 ± 0.27 |
| shadowhanddoorcloseinward | train | 1.2 | 8.8 ± 0.0 | 1.0 ± 0.01 |
| shadowhandgraspandplace | train | 5.0 | 500.0 ± 0.0 | 1.0 ± 0.16 |
| shadowhandliftunderarm | train | -41.5 | 400.0 ± 0.0 | 1.0 ± 0.01 |
| shadowhandpen | train | 0.0 | 190.0 ± 0.0 | 1.0 ± 0.03 |
| shadowhandpushblock | train | 230.0 | 450.0 ± 0.0 | 0.5 ± 0.02 |
| shadowhandreorientation | train | -50.0 | 3150.0 ± 0.0 | 0.4 ± 0.34 |
| shadowhandscissors | train | -40.0 | 725.0 ± 0.0 | 1.0 ± 0.02 |
| shadowhandswingcup | train | -490.0 | 4000.0 ± 0.0 | 1.0 ± 0.06 |
| shadowhandswitch | train | 45.0 | 280.0 ± 0.0 | 1.0 ± 0.0 |
| shadowhandtwocatchunderarm | train | 1.5 | 24.0 ± 0.0 | 0.7 ± 0.5 |

Table 6: Online Evaluation Results for Bi-DexHands

| Task Name | Split | Random Score | Expert Score | Normalized Return IQM |
|---|---|---|---|---|
| industrial-benchmark-100-v1 | test | -998.6 | -561.5 ± 0.0 | 1.0 ± 0.02 |
| industrial-benchmark-80-v1 | test | -779.9 | -485.4 ± 0.0 | 1.0 ± 0.0 |
| industrial-benchmark-85-v1 | test | -813.8 | -507.7 ± 0.0 | 1.0 ± 0.0 |
| industrial-benchmark-90-v1 | test | -883.4 | -521.9 ± 0.0 | 1.0 ± 0.0 |
| industrial-benchmark-95-v1 | test | -926.8 | -546.0 ± 0.0 | 1.0 ± 0.0 |
| industrial-benchmark-0-v1 | train | -357.4 | -180.8 ± 0.0 | 1.0 ± 0.0 |
| industrial-benchmark-10-v1 | train | -400.4 | -215.3 ± 0.0 | 1.0 ± 0.03 |
| industrial-benchmark-15-v1 | train | -444.4 | -230.0 ± 0.0 | 1.0 ± 0.0 |
| industrial-benchmark-20-v1 | train | -475.9 | -249.7 ± 0.0 | 0.9 ± 0.02 |
| industrial-benchmark-25-v1 | train | -461.1 | -272.9 ± 0.0 | 0.8 ± 0.05 |
| industrial-benchmark-30-v1 | train | -480.1 | -287.8 ± 0.0 | 1.0 ± 0.02 |
| industrial-benchmark-35-v1 | train | -501.7 | -314.4 ± 0.0 | 1.0 ± 0.01 |
| industrial-benchmark-40-v1 | train | -528.8 | -337.4 ± 0.0 | 0.9 ± 0.02 |
| industrial-benchmark-45-v1 | train | -547.8 | -361.0 ± 0.0 | 1.0 ± 0.01 |
| industrial-benchmark-5-v1 | train | -377.1 | -194.0 ± 0.0 | 1.0 ± 0.0 |
| industrial-benchmark-50-v1 | train | -574.4 | -378.3 ± 0.0 | 1.0 ± 0.0 |
| industrial-benchmark-55-v1 | train | -604.3 | -401.7 ± 0.0 | 1.0 ± 0.01 |
| industrial-benchmark-60-v1 | train | -636.3 | -430.1 ± 0.0 | 0.9 ± 0.02 |
| industrial-benchmark-65-v1 | train | -666.0 | -450.1 ± 0.0 | 0.9 ± 0.02 |
| industrial-benchmark-70-v1 | train | -695.5 | -470.8 ± 0.0 | 1.1 ± 0.0 |
| industrial-benchmark-75-v1 | train | -718.8 | -474.4 ± 0.0 | 1.0 ± 0.01 |

Table 7: Online Evaluation Results for Industrial-Benchmark

| Task Name | Split | Random Score | Expert Score | Normalized Return IQM |
|---|---|---|---|---|
| GOAL-t_pose | test | -3200.0 | -605.0 ± 0.0 | 0.2 ± 0.01 |
| REWARD-rotate-x-5-0_8 | test | 8.0 | 211.0 ± 0.0 | 0.1 ± 0.01 |
| REWARD-split-0_5 | test | 9.0 | 258.0 ± 0.0 | 0.0 ± 0.02 |
| GOAL-crouch_medium | train | -3120.0 | -341.0 ± 0.0 | 0.3 ± 0.04 |
| GOAL-lie_front | train | -3200.0 | -570.0 ± 0.0 | 1.0 ± 0.01 |
| GOAL-sit_hand_behind | train | -3200.0 | -535.0 ± 0.0 | 0.9 ± 0.01 |
| GOAL-zombie | train | -3200.0 | -410.0 ± 0.0 | 0.4 ± 0.07 |
| REWARD-crawl-0_5-0-d | train | 10.0 | 140.0 ± 0.0 | 0.9 ± 0.02 |
| REWARD-crouch-0 | train | 0.2 | 250.0 ± 0.0 | 1.0 ± 0.03 |
| REWARD-headstand | train | 0.0 | 247.0 ± 0.0 | 0.0 ± 0.0 |
| REWARD-move-ego-0-0 | train | 12.0 | 261.0 ± 0.0 | 0.2 ± 0.04 |
| REWARD-move-ego-0-2 | train | 10.0 | 265.0 ± 0.0 | 0.1 ± 0.02 |
| REWARD-move-ego-90-4 | train | 5.0 | 257.0 ± 0.0 | 0.3 ± 0.03 |
| REWARD-raisearms-h-h | train | 0.5 | 240.0 ± 0.0 | 0.0 ± 0.01 |
| REWARD-rotate-x--5-0_8 | train | 9.0 | 220.0 ± 0.0 | 0.5 ± 0.05 |

Table 8: Online Evaluation Results for HumEnv

| Task Name | Split | Random Score | Expert Score | Normalized Return IQM |
|---|---|---|---|---|
| 5zone-mixed_dec_jan_feb | test | -320.0 | -173.0 ± 0.0 | 0.4 ± 0.16 |
| 5zone-mixed_jun_jul_aug | test | -240.0 | -110.0 ± 0.0 | 1.0 ± 0.02 |
| 5zone-mixed_mar_apr_may | test | -330.0 | -90.0 ± 0.0 | 1.0 ± 0.13 |
| 5zone-mixed_oct_nov | test | -300.0 | -80.0 ± 0.0 | 0.9 ± 0.14 |
| 5zone-mixed_sep | test | -250.0 | -104.0 ± 0.0 | -0.1 ± 0.04 |
| warehouse-mixed_apr_may | test | -290.0 | -145.0 ± 0.0 | 1.2 ± 0.21 |
| warehouse-mixed_dec_jan_feb | test | -800.0 | -345.0 ± 0.0 | 0.7 ± 0.07 |
| warehouse-mixed_jul_aug | test | -650.0 | -340.0 ± 0.0 | 1.1 ± 0.03 |
| warehouse-mixed_jun | test | -370.0 | -180.0 ± 0.0 | 0.9 ± 0.02 |
| warehouse-mixed_mar | test | -660.0 | -250.0 ± 0.0 | 0.6 ± 0.06 |
| warehouse-mixed_sep | test | -340.0 | -200.0 ± 0.0 | 1.0 ± 0.03 |
| 5zone-cool_dec_jan_feb | train | -300.0 | -130.0 ± 0.0 | 0.9 ± 0.11 |
| 5zone-cool_jun_jul_aug | train | -300.0 | -118.0 ± 0.0 | 0.8 ± 0.38 |
| 5zone-cool_mar_apr_may | train | -300.0 | -50.0 ± 0.0 | 1.0 ± 0.07 |
| 5zone-cool_oct_nov | train | -300.0 | -58.0 ± 0.0 | 1.0 ± 0.07 |
| 5zone-cool_sep | train | -320.0 | -105.0 ± 0.0 | -0.0 ± 0.04 |
| 5zone-hot_dec_jan_feb | train | -300.0 | -50.0 ± 0.0 | 0.9 ± 0.01 |
| 5zone-hot_jun_jul_aug | train | -230.0 | -127.0 ± 0.0 | 1.1 ± 0.03 |
| 5zone-hot_mar_apr_may | train | -380.0 | -90.0 ± 0.0 | 0.6 ± 0.19 |
| 5zone-hot_oct_nov | train | -350.0 | -50.0 ± 0.0 | 0.8 ± 0.16 |
| 5zone-hot_sep | train | -240.0 | -120.0 ± 0.0 | 0.1 ± 0.02 |
| warehouse-cool_apr_may | train | -290.0 | -120.0 ± 0.0 | 0.9 ± 0.04 |
| warehouse-cool_dec_jan_feb | train | -700.0 | -300.0 ± 0.0 | 0.9 ± 0.1 |
| warehouse-cool_mar | train | -665.0 | -230.0 ± 0.0 | 0.7 ± 0.07 |
| warehouse-cool_nov | train | -500.0 | -180.0 ± 0.0 | 0.7 ± 0.09 |
| warehouse-cool_oct | train | -250.0 | -90.0 ± 0.0 | 0.9 ± 0.05 |
| warehouse-hot_apr_may | train | -1300.0 | -400.0 ± 0.0 | 0.7 ± 0.05 |
| warehouse-hot_dec_jan_feb | train | -175.0 | -80.0 ± 0.0 | 1.0 ± 0.13 |
| warehouse-hot_jun_jul_aug | train | -1000.0 | -500.0 ± 0.0 | 0.9 ± 0.07 |
| warehouse-hot_mar | train | -230.0 | -73.0 ± 0.0 | 0.9 ± 0.03 |
| warehouse-hot_nov | train | -530.0 | -100.0 ± 0.0 | 0.9 ± 0.03 |
| warehouse-hot_oct | train | -1100.0 | -360.0 ± 0.0 | 0.9 ± 0.04 |
| warehouse-hot_sep | train | -860.0 | -415.0 ± 0.0 | 1.0 ± 0.02 |

Table 9: Online Evaluation Results for SinerGym

| Task Name | Split | Random Score | Expert Score | Normalized Return IQM |
|---|---|---|---|---|
| cones_C_69 | test | 9.1 | 535.0 ± 0.0 | 0.9 ± 0.22 |
| def_C_574 | test | 9.4 | 611.4 ± 0.0 | 1.0 ± 0.01 |
| def_O_785 | test | 8.8 | 589.0 ± 0.0 | 1.0 ± 0.03 |
| def_T_971 | test | 6.7 | 454.7 ± 0.0 | 1.0 ± 0.0 |
| cones_C_520 | train | 10.0 | 813.4 ± 0.0 | 0.7 ± 0.15 |
| cones_C_574 | train | 11.4 | 616.2 ± 0.0 | 0.8 ± 0.14 |
| cones_C_785 | train | 10.0 | 645.8 ± 0.0 | 0.6 ± 0.31 |
| cones_C_85 | train | 11.4 | 719.2 ± 0.0 | 0.6 ± 0.37 |
| cones_C_859 | train | 8.2 | 577.4 ± 0.0 | 0.8 ± 0.2 |
| cones_S | train | 9.9 | 442.0 ± 0.0 | 0.9 ± 0.12 |
| def_C_520 | train | 7.9 | 921.5 ± 0.0 | 1.0 ± 0.01 |
| def_C_69 | train | 6.0 | 552.1 ± 0.0 | 1.0 ± 0.01 |
| def_C_785 | train | 5.7 | 649.4 ± 0.0 | 1.0 ± 0.0 |
| def_C_85 | train | 9.9 | 730.5 ± 0.0 | 1.0 ± 0.0 |
| def_C_859 | train | 7.8 | 588.1 ± 0.0 | 1.0 ± 0.01 |
| def_O_291 | train | 8.4 | 732.4 ± 0.0 | 1.0 ± 0.03 |
| def_O_647 | train | 8.2 | 568.9 ± 0.0 | 1.1 ± 0.03 |
| def_O_971 | train | 10.3 | 770.0 ± 0.0 | 1.0 ± 0.04 |
| def_S | train | 9.6 | 426.0 ± 0.0 | 1.0 ± 0.04 |
| def_T_855 | train | 10.7 | 485.3 ± 0.0 | 1.0 ± 0.0 |
| def_T_98 | train | 9.0 | 480.1 ± 0.0 | 1.0 ± 0.01 |

Table 10: Online Evaluation Results for MetaDrive

| Task Name | Split | Random Score | Expert Score | Normalized Return IQM |
|---|---|---|---|---|
| 2022_phase_1_task_10 | test | -1833.1 | -702.2 ± 0.0 | 0.8 ± 0.0 |
| 2022_phase_1_task_11 | test | -2386.1 | -1262.1 ± 0.0 | 0.8 ± 0.0 |
| 2022_phase_2_task_10 | test | -1724.5 | -611.3 ± 0.0 | 0.8 ± 0.0 |
| 2022_phase_2_task_11 | test | -2143.7 | -967.4 ± 0.0 | 0.8 ± 0.0 |
| 2022_phase_1_task_00 | train | -2212.6 | -1048.3 ± 0.0 | 1.0 ± 0.0 |
| 2022_phase_1_task_01 | train | -2017.0 | -831.4 ± 0.0 | 1.0 ± 0.0 |
| 2022_phase_1_task_02 | train | -1607.7 | -546.4 ± 0.0 | 1.0 ± 0.0 |
| 2022_phase_1_task_03 | train | -1602.2 | -506.8 ± 0.0 | 1.0 ± 0.0 |
| 2022_phase_1_task_04 | train | -2097.8 | -1054.4 ± 0.0 | 1.0 ± 0.0 |
| 2022_phase_1_task_05 | train | -2254.5 | -1238.1 ± 0.0 | 1.0 ± 0.0 |
| 2022_phase_1_task_06 | train | -1862.4 | -833.6 ± 0.0 | 1.0 ± 0.0 |
| 2022_phase_1_task_07 | train | -1511.0 | -433.7 ± 0.0 | 1.0 ± 0.0 |
| 2022_phase_1_task_08 | train | -1325.6 | -226.1 ± 0.0 | 1.0 ± 0.0 |
| 2022_phase_1_task_09 | train | -1518.1 | -453.1 ± 0.0 | 1.0 ± 0.0 |
| 2022_phase_2_task_00 | train | -2365.0 | -1232.1 ± 0.0 | 1.0 ± 0.0 |
| 2022_phase_2_task_01 | train | -2086.1 | -947.9 ± 0.0 | 1.0 ± 0.0 |
| 2022_phase_2_task_02 | train | -1678.5 | -594.0 ± 0.0 | 1.0 ± 0.0 |
| 2022_phase_2_task_03 | train | -1780.2 | -745.1 ± 0.0 | 1.0 ± 0.0 |
| 2022_phase_2_task_04 | train | -2408.6 | -1389.3 ± 0.0 | 1.0 ± 0.0 |
| 2022_phase_2_task_05 | train | -2604.0 | -1698.2 ± 0.0 | 1.0 ± 0.0 |
| 2022_phase_2_task_06 | train | -1965.7 | -934.7 ± 0.0 | 1.0 ± 0.0 |
| 2022_phase_2_task_07 | train | -1568.9 | -430.4 ± 0.0 | 1.0 ± 0.0 |
| 2022_phase_2_task_08 | train | -1270.2 | -241.3 ± 0.0 | 1.0 ± 0.0 |
| 2022_phase_2_task_09 | train | -1504.5 | -423.5 ± 0.0 | 1.0 ± 0.0 |

Table 11: Online Evaluation Results for CityLearn

| Task Name | Split | Random Score | Expert Score | Normalized Return IQM |
|---|---|---|---|---|
| aircraft_1 | test | -67000.0 | -0.5 ± 0.0 | 1.0 ± 0.0 |
| cable_5 | test | -56000.0 | -9.7 ± 0.0 | 1.0 ± 0.0 |
| pde_schrodinger | test | -6300000.0 | -2146.7 ± 0.0 | -4.5 ± 1.49 |
| aircraft_2 | train | -67000.0 | -0.9 ± 0.0 | 1.0 ± 0.0 |
| aircraft_4 | train | -10000.0 | -0.3 ± 0.0 | 1.0 ± 0.0 |
| aircraft_5 | train | -12000.0 | -1.5 ± 0.0 | 0.9 ± 0.05 |
| aircraft_6 | train | -720000.0 | -113.4 ± 0.0 | -17.5 ± 7.79 |
| cable_1 | train | -42000.0 | -23.3 ± 0.0 | 1.0 ± 0.0 |
| cable_2 | train | -49000.0 | -20.6 ± 0.0 | 1.0 ± 0.0 |
| cable_3 | train | -55000.0 | -15.2 ± 0.0 | 1.0 ± 0.0 |
| cable_4 | train | -55000.0 | -12.5 ± 0.0 | 1.0 ± 0.0 |
| pde_convection_diffusion_reaction | train | -58000.0 | -33.4 ± 0.0 | 0.3 ± 0.13 |

Table 12: Online Evaluation Results for ControlGym

## H.2    PROMPTED OFFLINE

| Task Name | Split | Random Score | Expert Score | Normalized Return IQM |
|---|---|---|---|---|
| bin-picking-v2 | test | 1.9 | 410.0 ± 64.37 | 0.0 ± 0.0 |
| box-close-v2 | test | 82.0 | 583.9 ± 13.28 | -0.0 ± 0.0 |
| door-lock-v2 | test | 119.3 | 831.9 ± 9.21 | 1.0 ± 0.0 |
| door-unlock-v2 | test | 94.7 | 813.0 ± 2.07 | 0.9 ± 0.01 |
| hand-insert-v2 | test | 2.5 | 775.5 ± 5.46 | 1.0 ± 0.0 |
| assembly-v2 | train | 44.7 | 283.3 ± 27.63 | 1.0 ± 0.0 |
| basketball-v2 | train | 2.6 | 591.7 ± 19.84 | 1.0 ± 0.01 |
| button-press-topdown-v2 | train | 28.6 | 499.0 ± 31.24 | 1.0 ± 0.0 |
| button-press-topdown-wall-v2 | train | 30.1 | 508.4 ± 35.03 | 1.0 ± 0.0 |
| button-press-v2 | train | 31.5 | 646.8 ± 11.04 | 1.0 ± 0.0 |
| button-press-wall-v2 | train | 8.0 | 679.6 ± 9.74 | 1.0 ± 0.0 |
| coffee-button-v2 | train | 33.0 | 733.0 ± 29.68 | 1.0 ± 0.0 |
| coffee-pull-v2 | train | 4.2 | 349.8 ± 104.95 | 1.0 ± 0.03 |
| coffee-push-v2 | train | 4.2 | 570.1 ± 105.82 | 1.0 ± 0.01 |
| dial-turn-v2 | train | 29.8 | 794.2 ± 13.12 | 1.0 ± 0.0 |
| disassemble-v2 | train | 33.9 | 486.8 ± 88.9 | 1.0 ± 0.12 |
| door-close-v2 | train | 5.0 | 543.7 ± 34.04 | 1.0 ± 0.01 |
| door-open-v2 | train | 54.2 | 587.5 ± 20.2 | 1.0 ± 0.01 |
| drawer-close-v2 | train | 107.6 | 866.2 ± 4.77 | 1.0 ± 0.0 |
| drawer-open-v2 | train | 134.0 | 493.9 ± 2.1 | 1.0 ± 0.0 |
| faucet-close-v2 | train | 247.5 | 767.0 ± 8.36 | 1.0 ± 0.0 |
| faucet-open-v2 | train | 249.5 | 757.1 ± 8.85 | 1.0 ± 0.0 |
| hammer-v2 | train | 90.3 | 683.8 ± 23.29 | 1.0 ± 0.0 |
| handle-press-side-v2 | train | 53.6 | 843.9 ± 11.33 | 1.0 ± 0.0 |
| handle-press-v2 | train | 54.0 | 809.1 ± 53.55 | 1.0 ± 0.04 |
| handle-pull-side-v2 | train | 2.3 | 494.4 ± 49.06 | 1.0 ± 0.01 |
| handle-pull-v2 | train | 14.1 | 709.7 ± 5.59 | 1.0 ± 0.0 |
| lever-pull-v2 | train | 61.4 | 608.6 ± 40.05 | 1.0 ± 0.02 |
| peg-insert-side-v2 | train | 1.7 | 314.5 ± 153.35 | 1.0 ± 0.02 |
| peg-unplug-side-v2 | train | 5.8 | 474.3 ± 89.02 | 1.0 ± 0.07 |
| pick-out-of-hole-v2 | train | 1.5 | 381.2 ± 28.75 | 1.0 ± 0.01 |
| pick-place-v2 | train | 1.1 | 473.7 ± 122.34 | 1.0 ± 0.05 |
| pick-place-wall-v2 | train | 0.0 | 540.6 ± 86.62 | 1.0 ± 0.03 |
| plate-slide-back-side-v2 | train | 33.7 | 743.9 ± 60.52 | 1.0 ± 0.0 |
| plate-slide-back-v2 | train | 34.2 | 679.4 ± 78.11 | 1.0 ± 0.0 |
| plate-slide-side-v2 | train | 23.4 | 672.9 ± 130.16 | 1.0 ± 0.1 |
| plate-slide-v2 | train | 72.6 | 525.5 ± 154.19 | 1.0 ± 0.05 |
| push-back-v2 | train | 1.2 | 464.4 ± 92.37 | 1.0 ± 0.03 |
| push-v2 | train | 5.3 | 762.0 ± 35.22 | 1.0 ± 0.0 |
| push-wall-v2 | train | 6.2 | 746.4 ± 16.4 | 1.0 ± 0.0 |
| reach-v2 | train | 140.0 | 639.2 ± 142.86 | 1.0 ± 0.02 |
| reach-wall-v2 | train | 157.0 | 601.9 ± 140.25 | 1.2 ± 0.35 |
| shelf-place-v2 | train | 0.0 | 261.1 ± 22.94 | 1.0 ± 0.04 |
| soccer-v2 | train | 4.9 | 483.6 ± 48.56 | 1.0 ± 0.08 |
| stick-pull-v2 | train | 2.7 | 519.2 ± 9.43 | 1.0 ± 0.0 |
| stick-push-v2 | train | 2.7 | 635.9 ± 7.86 | 1.0 ± 0.0 |
| sweep-into-v2 | train | 16.3 | 795.2 ± 11.04 | 1.0 ± 0.0 |
| sweep-v2 | train | 12.5 | 499.6 ± 22.37 | 1.0 ± 0.02 |
| window-close-v2 | train | 56.9 | 594.2 ± 38.3 | 1.0 ± 0.0 |
| window-open-v2 | train | 43.2 | 618.4 ± 46.16 | 1.0 ± 0.01 |

Table 13: Offline Evaluation Results for Meta-World

| Task Name | Split | Random Score | Expert Score | Normalized Return IQM |
|---|---|---|---|---|
| l-hard-lunar-lander-v1 | test | -0.4 | 1.0 ± 0.0 | -0.1 ± 0.36 |
| l-hard-pinball-v1 | test | -0.6 | 1.0 ± 0.0 | 0.4 ± 0.22 |
| l-mjc-walker-v1 | test | -0.3 | 1.0 ± 0.0 | 0.2 ± 0.0 |
| m-arm-hard-v1 | test | 0.0 | 1.0 ± 0.0 | 1.0 ± 0.0 |
| m-h14-thrustblock-v1 | test | -1.0 | 1.0 ± 0.0 | 0.0 ± 0.0 |
| s-h8-unicycle-balance-v1 | test | 0.4 | 1.0 ± 0.0 | 0.4 ± 0.2 |
| l-chain-lander-v1 | train | -0.3 | 1.0 ± 0.0 | 1.0 ± 0.0 |
| l-chain-thrust-v1 | train | 0.1 | 1.0 ± 0.0 | 1.0 ± 0.0 |
| l-grasp-easy-v1 | train | -0.4 | 1.0 ± 0.0 | 1.0 ± 0.02 |
| l-h1-car-left-v1 | train | 0.0 | 1.0 ± 0.0 | 0.3 ± 0.24 |
| l-h10-morph-direction-2-v1 | train | 0.0 | 1.0 ± 0.0 | 1.0 ± 0.0 |
| l-h14-simple-thruster-v1 | train | -0.4 | 1.0 ± 0.0 | 1.0 ± 0.0 |
| l-h3-car-right-v1 | train | 0.2 | 1.0 ± 0.0 | 1.0 ± 0.0 |
| l-h7-maze-1-v1 | train | -0.5 | 1.0 ± 0.0 | 0.8 ± 0.19 |
| l-h9-morph-direction-v1 | train | 0.0 | 1.0 ± 0.0 | 1.0 ± 0.0 |
| l-mjc-half-cheetah-easy-v1 | train | 0.0 | 1.0 ± 0.0 | 1.0 ± 0.06 |
| l-mjc-half-cheetah-v1 | train | -0.3 | 0.9 ± 0.0 | 1.1 ± 0.08 |
| l-mjc-hopper-easy-v1 | train | 0.3 | 1.0 ± 0.0 | 1.0 ± 0.0 |
| l-mjc-hopper-v1 | train | -0.3 | 1.0 ± 0.0 | 1.0 ± 0.05 |
| l-mjc-swimmer-v1 | train | 0.2 | 0.9 ± 0.0 | 1.1 ± 0.0 |
| l-mjc-walker-easy-v1 | train | 0.1 | 1.0 ± 0.0 | 0.9 ± 0.13 |
| l-pr-v1 | train | 0.6 | 1.0 ± 0.0 | 1.0 ± 0.0 |
| m-arm-left-v1 | train | 0.0 | 1.0 ± 0.0 | 1.0 ± 0.0 |
| m-arm-right-v1 | train | 0.1 | 1.0 ± 0.0 | 1.0 ± 0.0 |
| m-arm-up-v1 | train | -0.5 | 1.0 ± 0.0 | 1.0 ± 0.0 |
| m-h0-unicycle-v1 | train | -0.4 | 1.0 ± 0.0 | 1.0 ± 0.0 |
| m-h10-thrust-right-easy-v1 | train | 0.0 | 1.0 ± 0.0 | 1.0 ± 0.0 |
| m-h11-thrust-left-easy-v1 | train | 0.0 | 1.0 ± 0.0 | 1.0 ± 0.0 |
| m-h12-thrustfall-left-v1 | train | 0.0 | 1.0 ± 0.0 | 1.0 ± 0.0 |
| m-h13-thrustfall-right-v1 | train | 0.1 | 1.0 ± 0.0 | 1.0 ± 0.0 |
| m-h18-thrust-right-very-easy-v1 | train | 0.0 | 1.0 ± 0.0 | 1.0 ± 0.0 |
| m-h19-thrust-left-very-easy-v1 | train | 0.0 | 1.0 ± 0.0 | 1.0 ± 0.0 |
| m-h2-car-right-v1 | train | 0.0 | 1.0 ± 0.0 | 1.0 ± 0.0 |
| m-h3-car-thrust-v1 | train | -0.5 | 1.0 ± 0.0 | 1.0 ± 0.0 |
| m-h4-thrust-the-needle-v1 | train | -0.7 | 1.0 ± 0.0 | 1.0 ± 0.0 |
| m-h5-angry-birds-v1 | train | 0.4 | 1.0 ± 0.0 | 0.9 ± 0.11 |
| m-h6-thrust-over-v1 | train | -0.5 | 1.0 ± 0.0 | 1.0 ± 0.0 |
| m-h7-car-flip-v1 | train | -0.4 | 1.0 ± 0.0 | 1.0 ± 0.0 |
| m-h8-weird-vehicle-v1 | train | 0.2 | 1.0 ± 0.0 | 1.0 ± 0.0 |
| m-h9-spin-the-right-way-v1 | train | -0.3 | 1.0 ± 0.0 | 1.0 ± 0.0 |
| s-h0-weak-thrust-v1 | train | -0.5 | 1.0 ± 0.0 | 1.0 ± 0.0 |
| s-h1-thrust-over-ball-v1 | train | -0.3 | 1.0 ± 0.0 | 1.0 ± 0.0 |
| s-h2-one-wheel-car-v1 | train | 0.0 | 1.0 ± 0.0 | 1.0 ± 0.0 |
| s-h3-point-the-thruster-v1 | train | 0.0 | 1.0 ± 0.0 | 1.0 ± 0.0 |
| s-h4-thrust-aim-v1 | train | 0.3 | 1.0 ± 0.0 | 1.0 ± 0.0 |
| s-h5-rotate-fall-v1 | train | -0.6 | 1.0 ± 0.0 | 1.0 ± 0.0 |
| s-h6-unicycle-right-v1 | train | 0.0 | 1.0 ± 0.0 | 1.0 ± 0.0 |
| s-h7-unicycle-left-v1 | train | 0.1 | 1.0 ± 0.0 | 1.0 ± 0.0 |

Table 14: Offline Evaluation Results for Kinetix

| Task Name | Split | Random Score | Expert Score | Normalized Return IQM |
|---|---|---|---|---|
| Ant-v4 | param_shift | -103.1 | 4988.8 ± 2057.47 | 1.0 ± 0.01 |
| HalfCheetah-v4 | param_shift | -264.2 | 7331.8 ± 420.5 | 1.0 ± 0.01 |
| Hopper-v4 | param_shift | 18.9 | 2139.9 ± 286.52 | 0.8 ± 0.26 |
| Humanoid-v4 | param_shift | 121.9 | 7373.8 ± 132.4 | 1.0 ± 0.0 |
| HumanoidStandup-v4 | param_shift | 33027.7 | 303552.9 ± 19279.02 | 1.0 ± 0.03 |
| InvertedDoublePendulum-v4 | param_shift | 58.5 | 9256.8 ± 293.62 | 1.0 ± 0.0 |
| InvertedPendulum-v4 | param_shift | 5.5 | 222.1 ± 7.62 | 0.6 ± 0.14 |
| Pusher-v4 | param_shift | -146.8 | -23.8 ± 0.57 | 1.0 ± 0.02 |
| Reacher-v4 | param_shift | -41.9 | -5.8 ± 1.06 | 1.0 ± 0.01 |
| Swimmer-v4 | param_shift | -0.3 | 95.3 ± 1.28 | 1.0 ± 0.02 |
| Walker2d-v4 | param_shift | 3.1 | 5176.1 ± 98.82 | 1.0 ± 0.02 |
| Ant-v4 | train | -55.2 | 6421.3 ± 17.62 | 1.0 ± 0.01 |
| HalfCheetah-v4 | train | -267.2 | 7786.1 ± 16.1 | 1.0 ± 0.01 |
| Hopper-v4 | train | 13.8 | 3385.3 ± 16.47 | 1.1 ± 0.12 |
| Humanoid-v4 | train | 117.9 | 7521.7 ± 12.92 | 1.0 ± 0.0 |
| HumanoidStandup-v4 | train | 34034.7 | 299534.8 ± 2187.55 | 1.1 ± 0.1 |
| InvertedDoublePendulum-v4 | train | 61.7 | 9359.5 ± 0.11 | 1.0 ± 0.0 |
| InvertedPendulum-v4 | train | 5.7 | 1000.0 ± 0.0 | 1.0 ± 0.0 |
| Pusher-v4 | train | -149.8 | -24.7 ± 1.61 | 1.0 ± 0.03 |
| Reacher-v4 | train | -43.0 | -5.6 ± 0.73 | 1.0 ± 0.01 |
| Swimmer-v4 | train | 1.7 | 96.3 ± 0.96 | 1.0 ± 0.05 |
| Walker2d-v4 | train | 2.7 | 5357.3 ± 13.14 | 1.0 ± 0.01 |

Table 15: Offline Evaluation Results for MuJoCo

| Task Name | Split | Random Score | Expert Score | Normalized Return IQM |
|---|---|---|---|---|
| shadowhanddoorcloseoutward | test | 930.0 | 1380.0 ± 0.0 | -1.2 ± 0.01 |
| shadowhanddooropeninward | test | -15.0 | 410.0 ± 0.0 | 0.3 ± 0.03 |
| shadowhanddooropenoutward | test | 5.0 | 610.0 ± 0.0 | 0.1 ± 0.0 |
| shadowhandkettle | test | -200.0 | 53.0 ± 0.0 | -0.0 ± 0.03 |
| shadowhandblockstack | train | 90.0 | 285.0 ± 0.0 | 1.0 ± 0.08 |
| shadowhandbottlecap | train | 100.0 | 400.0 ± 0.0 | 1.1 ± 0.05 |
| shadowhandcatchabreast | train | 0.5 | 67.0 ± 0.0 | 0.1 ± 0.3 |
| shadowhandcatchover2underarm | train | 4.0 | 34.0 ± 0.0 | 0.9 ± 0.22 |
| shadowhandcatchunderarm | train | 0.8 | 25.0 ± 0.0 | 0.9 ± 0.15 |
| shadowhanddoorcloseinward | train | 1.2 | 8.8 ± 0.0 | 1.0 ± 0.01 |
| shadowhandgraspandplace | train | 5.0 | 500.0 ± 0.0 | 1.0 ± 0.01 |
| shadowhandliftunderarm | train | -41.5 | 400.0 ± 0.0 | 1.0 ± 0.01 |
| shadowhandpen | train | 0.0 | 190.0 ± 0.0 | 1.0 ± 0.02 |
| shadowhandpushblock | train | 230.0 | 450.0 ± 0.0 | 1.0 ± 0.01 |
| shadowhandreorientation | train | -50.0 | 3150.0 ± 0.0 | 0.6 ± 0.46 |
| shadowhandscissors | train | -40.0 | 725.0 ± 0.0 | 1.0 ± 0.0 |
| shadowhandswingcup | train | -490.0 | 4000.0 ± 0.0 | 1.0 ± 0.1 |
| shadowhandswitch | train | 45.0 | 280.0 ± 0.0 | 1.0 ± 0.0 |
| shadowhandtwocatchunderarm | train | 1.5 | 24.0 ± 0.0 | 0.4 ± 0.48 |

Table 16: Offline Evaluation Results for Bi-DexHands

| Task Name | Split | Random Score | Expert Score | Normalized Return IQM |
|---|---|---|---|---|
| industrial-benchmark-100-v1 | test | -998.6 | -561.5 ± 0.0 | 1.0 ± 0.0 |
| industrial-benchmark-80-v1 | test | -779.9 | -485.4 ± 0.0 | 0.8 ± 0.0 |
| industrial-benchmark-85-v1 | test | -813.8 | -507.7 ± 0.0 | 1.0 ± 0.01 |
| industrial-benchmark-90-v1 | test | -883.4 | -521.9 ± 0.0 | 1.0 ± 0.0 |
| industrial-benchmark-95-v1 | test | -926.8 | -546.0 ± 0.0 | 0.9 ± 0.0 |
| industrial-benchmark-0-v1 | train | -357.4 | -180.8 ± 0.0 | 1.0 ± 0.0 |
| industrial-benchmark-10-v1 | train | -400.4 | -215.3 ± 0.0 | 1.0 ± 0.0 |
| industrial-benchmark-15-v1 | train | -444.4 | -230.0 ± 0.0 | 1.0 ± 0.0 |
| industrial-benchmark-20-v1 | train | -475.9 | -249.7 ± 0.0 | 1.0 ± 0.01 |
| industrial-benchmark-25-v1 | train | -461.1 | -272.9 ± 0.0 | 1.0 ± 0.01 |
| industrial-benchmark-30-v1 | train | -480.1 | -287.8 ± 0.0 | 1.0 ± 0.01 |
| industrial-benchmark-35-v1 | train | -501.7 | -314.4 ± 0.0 | 1.0 ± 0.02 |
| industrial-benchmark-40-v1 | train | -528.8 | -337.4 ± 0.0 | 1.0 ± 0.01 |
| industrial-benchmark-45-v1 | train | -547.8 | -361.0 ± 0.0 | 1.0 ± 0.01 |
| industrial-benchmark-5-v1 | train | -377.1 | -194.0 ± 0.0 | 1.0 ± 0.0 |
| industrial-benchmark-50-v1 | train | -574.4 | -378.3 ± 0.0 | 1.0 ± 0.01 |
| industrial-benchmark-55-v1 | train | -604.3 | -401.7 ± 0.0 | 1.0 ± 0.01 |
| industrial-benchmark-60-v1 | train | -636.3 | -430.1 ± 0.0 | 1.0 ± 0.03 |
| industrial-benchmark-65-v1 | train | -666.0 | -450.1 ± 0.0 | 1.0 ± 0.0 |
| industrial-benchmark-70-v1 | train | -695.5 | -470.8 ± 0.0 | 1.1 ± 0.0 |
| industrial-benchmark-75-v1 | train | -718.8 | -474.4 ± 0.0 | 1.0 ± 0.01 |

Table 17: Offline Evaluation Results for Industrial-Benchmark

| Task Name | Split | Random Score | Expert Score | Normalized Return IQM |
|---|---|---|---|---|
| GOAL-t_pose | test | -3200.0 | -605.0 ± 0.0 | 0.2 ± 0.01 |
| REWARD-rotate-x-5-0_8 | test | 8.0 | 211.0 ± 0.0 | 0.0 ± 0.02 |
| REWARD-split-0_5 | test | 9.0 | 258.0 ± 0.0 | 0.1 ± 0.03 |
| GOAL-crouch_medium | train | -3120.0 | -341.0 ± 0.0 | 0.4 ± 0.15 |
| GOAL-lie_front | train | -3200.0 | -570.0 ± 0.0 | 1.0 ± 0.01 |
| GOAL-sit_hand_behind | train | -3200.0 | -535.0 ± 0.0 | 0.9 ± 0.05 |
| GOAL-zombie | train | -3200.0 | -410.0 ± 0.0 | 0.4 ± 0.08 |
| REWARD-crawl-0_5-0-d | train | 10.0 | 140.0 ± 0.0 | 1.0 ± 0.01 |
| REWARD-crouch-0 | train | 0.2 | 250.0 ± 0.0 | 1.0 ± 0.05 |
| REWARD-headstand | train | 0.0 | 247.0 ± 0.0 | 0.1 ± 0.01 |
| REWARD-move-ego-0-0 | train | 12.0 | 261.0 ± 0.0 | 0.2 ± 0.06 |
| REWARD-move-ego-0-2 | train | 10.0 | 265.0 ± 0.0 | 0.3 ± 0.04 |
| REWARD-move-ego-90-4 | train | 5.0 | 257.0 ± 0.0 | 0.3 ± 0.04 |
| REWARD-raisearms-h-h | train | 0.5 | 240.0 ± 0.0 | 0.1 ± 0.04 |
| REWARD-rotate-x–5-0_8 | train | 9.0 | 220.0 ± 0.0 | 0.7 ± 0.06 |

Table 18: Offline Evaluation Results for HumEnv

| Task Name | Split | Random Score | Expert Score | Normalized Return IQM |
|---|---|---|---|---|
| 5zone-mixed_dec_jan_feb | test | -320.0 | -173.0 ± 0.0 | 0.9 ± 0.09 |
| 5zone-mixed_jun_jul_aug | test | -240.0 | -110.0 ± 0.0 | 1.0 ± 0.03 |
| 5zone-mixed_mar_apr_may | test | -330.0 | -90.0 ± 0.0 | 1.2 ± 0.08 |
| 5zone-mixed_oct_nov | test | -300.0 | -80.0 ± 0.0 | 1.1 ± 0.07 |
| 5zone-mixed_sep | test | -250.0 | -104.0 ± 0.0 | 0.7 ± 0.02 |
| warehouse-mixed_apr_may | test | -290.0 | -145.0 ± 0.0 | 1.0 ± 0.15 |
| warehouse-mixed_dec_jan_feb | test | -800.0 | -345.0 ± 0.0 | 0.9 ± 0.03 |
| warehouse-mixed_jul_aug | test | -650.0 | -340.0 ± 0.0 | 1.1 ± 0.03 |
| warehouse-mixed_jun | test | -370.0 | -180.0 ± 0.0 | 0.9 ± 0.02 |
| warehouse-mixed_mar | test | -660.0 | -250.0 ± 0.0 | 1.0 ± 0.01 |
| warehouse-mixed_sep | test | -340.0 | -200.0 ± 0.0 | 1.1 ± 0.02 |
| 5zone-cool_dec_jan_feb | train | -300.0 | -130.0 ± 0.0 | 1.0 ± 0.11 |
| 5zone-cool_jun_jul_aug | train | -300.0 | -118.0 ± 0.0 | 1.0 ± 0.05 |
| 5zone-cool_mar_apr_may | train | -300.0 | -50.0 ± 0.0 | 1.0 ± 0.09 |
| 5zone-cool_oct_nov | train | -300.0 | -58.0 ± 0.0 | 1.0 ± 0.09 |
| 5zone-cool_sep | train | -320.0 | -105.0 ± 0.0 | 0.9 ± 0.01 |
| 5zone-hot_dec_jan_feb | train | -300.0 | -50.0 ± 0.0 | 1.0 ± 0.05 |
| 5zone-hot_jun_jul_aug | train | -230.0 | -127.0 ± 0.0 | 1.0 ± 0.03 |
| 5zone-hot_mar_apr_may | train | -380.0 | -90.0 ± 0.0 | 1.0 ± 0.04 |
| 5zone-hot_oct_nov | train | -350.0 | -50.0 ± 0.0 | 1.0 ± 0.03 |
| 5zone-hot_sep | train | -240.0 | -120.0 ± 0.0 | 0.9 ± 0.02 |
| warehouse-cool_apr_may | train | -290.0 | -120.0 ± 0.0 | 1.0 ± 0.01 |
| warehouse-cool_dec_jan_feb | train | -700.0 | -300.0 ± 0.0 | 1.0 ± 0.05 |
| warehouse-cool_mar | train | -665.0 | -230.0 ± 0.0 | 1.0 ± 0.0 |
| warehouse-cool_nov | train | -500.0 | -180.0 ± 0.0 | 1.0 ± 0.01 |
| warehouse-cool_oct | train | -250.0 | -90.0 ± 0.0 | 1.0 ± 0.02 |
| warehouse-hot_apr_may | train | -1300.0 | -400.0 ± 0.0 | 1.0 ± 0.03 |
| warehouse-hot_dec_jan_feb | train | -175.0 | -80.0 ± 0.0 | 1.0 ± 0.15 |
| warehouse-hot_jun_jul_aug | train | -1000.0 | -500.0 ± 0.0 | 0.7 ± 0.2 |
| warehouse-hot_mar | train | -230.0 | -73.0 ± 0.0 | 1.0 ± 0.01 |
| warehouse-hot_nov | train | -530.0 | -100.0 ± 0.0 | 1.0 ± 0.01 |
| warehouse-hot_oct | train | -1100.0 | -360.0 ± 0.0 | 1.0 ± 0.01 |
| warehouse-hot_sep | train | -860.0 | -415.0 ± 0.0 | 1.0 ± 0.01 |

Table 19: Offline Evaluation Results for SinerGym

| Task Name | Split | Random Score | Expert Score | Normalized Return IQM |
|---|---|---|---|---|
| cones_C_69 | test | 9.1 | 535.0 ± 0.0 | 0.7 ± 0.31 |
| def_C_574 | test | 9.4 | 611.4 ± 0.0 | 1.0 ± 0.01 |
| def_O_785 | test | 8.8 | 589.0 ± 0.0 | 1.1 ± 0.03 |
| def_T_971 | test | 6.7 | 454.7 ± 0.0 | 1.0 ± 0.01 |
| cones_C_520 | train | 10.0 | 813.4 ± 0.0 | 1.0 ± 0.06 |
| cones_C_574 | train | 11.4 | 616.2 ± 0.0 | 0.8 ± 0.17 |
| cones_C_785 | train | 10.0 | 645.8 ± 0.0 | 0.9 ± 0.19 |
| cones_C_85 | train | 11.4 | 719.2 ± 0.0 | 0.6 ± 0.33 |
| cones_C_859 | train | 8.2 | 577.4 ± 0.0 | 1.0 ± 0.2 |
| cones_S | train | 9.9 | 442.0 ± 0.0 | 0.7 ± 0.19 |
| def_C_520 | train | 7.9 | 921.5 ± 0.0 | 1.0 ± 0.01 |
| def_C_69 | train | 6.0 | 552.1 ± 0.0 | 1.0 ± 0.01 |
| def_C_785 | train | 5.7 | 649.4 ± 0.0 | 1.0 ± 0.0 |
| def_C_85 | train | 9.9 | 730.5 ± 0.0 | 1.0 ± 0.0 |
| def_C_859 | train | 7.8 | 588.1 ± 0.0 | 1.0 ± 0.01 |
| def_O_291 | train | 8.4 | 732.4 ± 0.0 | 1.0 ± 0.03 |
| def_O_647 | train | 8.2 | 568.9 ± 0.0 | 1.0 ± 0.02 |
| def_O_971 | train | 10.3 | 770.0 ± 0.0 | 1.0 ± 0.04 |
| def_S | train | 9.6 | 426.0 ± 0.0 | 1.0 ± 0.05 |
| def_T_855 | train | 10.7 | 485.3 ± 0.0 | 1.0 ± 0.0 |
| def_T_98 | train | 9.0 | 480.1 ± 0.0 | 1.0 ± 0.01 |

Table 20: Offline Evaluation Results for MetaDrive

| Task Name | Split | Random Score | Expert Score | Normalized Return IQM |
|---|---|---|---|---|
| 2022_phase_1_task_10 | test | -1833.1 | -702.2 ± 0.0 | 0.8 ± 0.0 |
| 2022_phase_1_task_11 | test | -2386.1 | -1262.1 ± 0.0 | 0.8 ± 0.0 |
| 2022_phase_2_task_10 | test | -1724.5 | -611.3 ± 0.0 | 0.8 ± 0.0 |
| 2022_phase_2_task_11 | test | -2143.7 | -967.4 ± 0.0 | 0.8 ± 0.0 |
| 2022_phase_1_task_00 | train | -2212.6 | -1048.3 ± 0.0 | 1.0 ± 0.0 |
| 2022_phase_1_task_01 | train | -2017.0 | -831.4 ± 0.0 | 1.0 ± 0.0 |
| 2022_phase_1_task_02 | train | -1607.7 | -546.4 ± 0.0 | 1.0 ± 0.0 |
| 2022_phase_1_task_03 | train | -1602.2 | -506.8 ± 0.0 | 1.0 ± 0.0 |
| 2022_phase_1_task_04 | train | -2097.8 | -1054.4 ± 0.0 | 1.0 ± 0.0 |
| 2022_phase_1_task_05 | train | -2254.5 | -1238.1 ± 0.0 | 1.0 ± 0.0 |
| 2022_phase_1_task_06 | train | -1862.4 | -833.6 ± 0.0 | 1.0 ± 0.0 |
| 2022_phase_1_task_07 | train | -1511.0 | -433.7 ± 0.0 | 1.0 ± 0.0 |
| 2022_phase_1_task_08 | train | -1325.6 | -226.1 ± 0.0 | 1.0 ± 0.0 |
| 2022_phase_1_task_09 | train | -1518.1 | -453.1 ± 0.0 | 1.0 ± 0.0 |
| 2022_phase_2_task_00 | train | -2365.0 | -1232.1 ± 0.0 | 1.0 ± 0.0 |
| 2022_phase_2_task_01 | train | -2086.1 | -947.9 ± 0.0 | 1.0 ± 0.0 |
| 2022_phase_2_task_02 | train | -1678.5 | -594.0 ± 0.0 | 1.0 ± 0.0 |
| 2022_phase_2_task_03 | train | -1780.2 | -745.1 ± 0.0 | 1.0 ± 0.0 |
| 2022_phase_2_task_04 | train | -2408.6 | -1389.3 ± 0.0 | 1.0 ± 0.0 |
| 2022_phase_2_task_05 | train | -2604.0 | -1698.2 ± 0.0 | 1.0 ± 0.0 |
| 2022_phase_2_task_06 | train | -1965.7 | -934.7 ± 0.0 | 1.0 ± 0.0 |
| 2022_phase_2_task_07 | train | -1568.9 | -430.4 ± 0.0 | 1.0 ± 0.0 |
| 2022_phase_2_task_08 | train | -1270.2 | -241.3 ± 0.0 | 1.0 ± 0.0 |
| 2022_phase_2_task_09 | train | -1504.5 | -423.5 ± 0.0 | 1.0 ± 0.0 |

Table 21: Offline Evaluation Results for CityLearn

| Task
Name | Split | Random
Score | Expert
Score | Normalized
Return IQM |
|---|---|---|---|---|
| aircraft_1 | test | -67000.0 | -0.5 ± 0.0 | 1.0 ± 0.0 |
| cable_5 | test | -56000.0 | -9.7 ± 0.0 | 1.0 ± 0.0 |
| pde_schrodinger | test | -6300000.0 | -2146.7 ± 0.0 | 1.0 ± 0.0 |
| aircraft_2 | train | -67000.0 | -0.9 ± 0.0 | 1.0 ± 0.0 |
| aircraft_4 | train | -10000.0 | -0.3 ± 0.0 | 1.0 ± 0.0 |
| aircraft_5 | train | -12000.0 | -1.5 ± 0.0 | 1.0 ± 0.0 |
| aircraft_6 | train | -720000.0 | -113.4 ± 0.0 | -2.8 ± 1.78 |
| cable_1 | train | -42000.0 | -23.3 ± 0.0 | 1.0 ± 0.0 |
| cable_2 | train | -49000.0 | -20.6 ± 0.0 | 1.0 ± 0.0 |
| cable_3 | train | -55000.0 | -15.2 ± 0.0 | 1.0 ± 0.0 |
| cable_4 | train | -55000.0 | -12.5 ± 0.0 | 1.0 ± 0.0 |
| pde_convection_diffusion_reaction | train | -58000.0 | -33.4 ± 0.0 | 1.0 ± 0.0 |

Table 22: Offline Evaluation Results for ControlGym

# I    DATASET SIZE AND METADATA

| Task
Name | Trajectory
Length | Number of
Trajectories | Mean
Eplen. | Obs.
Shape | Action
Shape | Reward
Scale | Split |
|---|---|---|---|---|---|---|---|
| InvertedPendulum-v4 | 1000488 | 10 | 68.2 | (4,) | (1,) | 0.1 | train |
| InvertedDoublePendulum-v4 | 1000480 | 10 | 55.7 | (11,) | (1,) | 0.1 | train |
| Humanoid-v4 | 1000488 | 15 | 207.8 | (376,) | (17,) | 0.1 | train |
| Walker2d-v4 | 1000586 | 10 | 350.4 | (17,) | (6,) | 0.1 | train |
| Hopper-v4 | 1000462 | 10 | 227.7 | (11,) | (3,) | 0.1 | train |
| Ant-v4 | 1000653 | 10 | 524.7 | (27,) | (8,) | 0.1 | train |
| HumanoidStandup-v4 | 1000000 | 15 | 1000.0 | (376,) | (17,) | 0.01 | train |
| Swimmer-v4 | 1000000 | 5 | 1000.0 | (8,) | (2,) | 1.0 | train |
| Reacher-v4 | 1000000 | 5 | 50.0 | (11,) | (2,) | 0.1 | train |
| Pusher-v4 | 1000000 | 5 | 100.0 | (23,) | (7,) | 0.1 | train |
| HalfCheetah-v4 | 1000000 | 5 | 1000.0 | (17,) | (6,) | 0.1 | train |

Table 23: Dataset Metadata for MuJoCo

| Task Name | Trajectory Length | Number of Trajectories | Mean Eplen. | Obs. Shape | Action Shape | Reward Scale | Split |
|---|---|---|---|---|---|---|---|
| shadowhandblockstack | 250140 | 10 | 248.6 | (428,) | (52,) | 1 | train |
| shadowhandbottlecap | 135065 | 10 | 123.7 | (420,) | (52,) | 1 | train |
| shadowhandcatchabreast | 165078 | 10 | 98.5 | (422,) | (52,) | 1 | train |
| shadowhandcatchover2underarm | 100034 | 10 | 55.6 | (422,) | (52,) | 1 | train |
| shadowhandcatchunderarm | 100029 | 10 | 64.9 | (422,) | (52,) | 1 | train |
| shadowhanddoorcloseinward | 100014 | 10 | 23.8 | (417,) | (52,) | 1 | train |
| shadowhandgraspandplace | 500250 | 8 | 332.8 | (425,) | (52,) | 1 | train |
| shadowhandliftunderarm | 500301 | 10 | 455.0 | (417,) | (52,) | 1 | train |
| shadowhandpen | 135077 | 10 | 120.5 | (417,) | (52,) | 1 | train |
| shadowhandpushblock | 135065 | 10 | 123.2 | (428,) | (52,) | 1 | train |
| shadowhandreorientation | 600228 | 7 | 463.8 | (422,) | (40,) | 1 | train |
| shadowhandscissors | 175075 | 10 | 149.0 | (417,) | (52,) | 1 | train |
| shadowhandswingcup | 320225 | 10 | 299.0 | (417,) | (52,) | 1 | train |
| shadowhandswitch | 135036 | 10 | 124.0 | (417,) | (52,) | 1 | train |
| shadowhandtwocatchunderarm | 100027 | 10 | 65.1 | (446,) | (52,) | 1 | train |
| shadowhanddoorcloseoutward | 265185 | 10 | 249.0 | (417,) | (52,) | 1 | test |
| shadowhanddooropeninward | 265164 | 10 | 249.0 | (417,) | (52,) | 1 | test |
| shadowhanddooropenoutward | 250242 | 10 | 249.0 | (417,) | (52,) | 1 | test |
| shadowhandkettle | 135033 | 10 | 124.0 | (417,) | (52,) | 1 | test |
| shadowhandover | 100040 | 10 | 64.4 | (398,) | (40,) | 1 | test |

Table 24: Dataset Metadata for Bi-DexHands

| Task Name | Trajectory Length | Number of Trajectories | Mean Eplen. | Obs. Shape | Action Shape | Reward Scale | Split |
|---|---|---|---|---|---|---|---|
| assembly-v2 | 100000 | 15 | 100.0 | (39,) | (4,) | 1 | train |
| basketball-v2 | 100000 | 15 | 100.0 | (39,) | (4,) | 1 | train |
| button-press-topdown-v2 | 100000 | 15 | 100.0 | (39,) | (4,) | 1 | train |
| button-press-topdown-wall-v2 | 100000 | 15 | 100.0 | (39,) | (4,) | 1 | train |
| button-press-v2 | 100000 | 15 | 100.0 | (39,) | (4,) | 1 | train |
| button-press-wall-v2 | 100000 | 15 | 100.0 | (39,) | (4,) | 1 | train |
| coffee-button-v2 | 100000 | 15 | 100.0 | (39,) | (4,) | 1 | train |
| coffee-pull-v2 | 100000 | 17 | 100.0 | (39,) | (4,) | 1 | train |
| coffee-push-v2 | 100000 | 15 | 100.0 | (39,) | (4,) | 1 | train |
| dial-turn-v2 | 100000 | 15 | 100.0 | (39,) | (4,) | 1 | train |
| disassemble-v2 | 100000 | 15 | 100.0 | (39,) | (4,) | 1 | train |
| door-close-v2 | 100000 | 15 | 100.0 | (39,) | (4,) | 1 | train |
| door-open-v2 | 100000 | 15 | 100.0 | (39,) | (4,) | 1 | train |
| drawer-close-v2 | 100000 | 15 | 100.0 | (39,) | (4,) | 1 | train |
| drawer-open-v2 | 100000 | 15 | 100.0 | (39,) | (4,) | 1 | train |
| faucet-close-v2 | 100000 | 15 | 100.0 | (39,) | (4,) | 1 | train |
| faucet-open-v2 | 100000 | 15 | 100.0 | (39,) | (4,) | 1 | train |
| hammer-v2 | 100000 | 15 | 100.0 | (39,) | (4,) | 1 | train |
| handle-press-side-v2 | 100000 | 15 | 100.0 | (39,) | (4,) | 1 | train |
| handle-press-v2 | 100000 | 15 | 100.0 | (39,) | (4,) | 1 | train |
| handle-pull-side-v2 | 100000 | 15 | 100.0 | (39,) | (4,) | 1 | train |
| handle-pull-v2 | 100000 | 15 | 100.0 | (39,) | (4,) | 1 | train |
| lever-pull-v2 | 100000 | 13 | 100.0 | (39,) | (4,) | 1 | train |
| peg-insert-side-v2 | 100000 | 12 | 100.0 | (39,) | (4,) | 1 | train |
| peg-unplug-side-v2 | 100000 | 15 | 100.0 | (39,) | (4,) | 1 | train |
| pick-out-of-hole-v2 | 100000 | 15 | 100.0 | (39,) | (4,) | 1 | train |
| pick-place-v2 | 100000 | 15 | 100.0 | (39,) | (4,) | 1 | train |
| pick-place-wall-v2 | 100000 | 15 | 100.0 | (39,) | (4,) | 1 | train |
| plate-slide-back-side-v2 | 100000 | 15 | 100.0 | (39,) | (4,) | 1 | train |
| plate-slide-back-v2 | 100000 | 15 | 100.0 | (39,) | (4,) | 1 | train |
| plate-slide-side-v2 | 100000 | 15 | 100.0 | (39,) | (4,) | 1 | train |
| plate-slide-v2 | 100000 | 15 | 100.0 | (39,) | (4,) | 1 | train |
| push-back-v2 | 100000 | 15 | 100.0 | (39,) | (4,) | 1 | train |
| push-v2 | 100000 | 15 | 100.0 | (39,) | (4,) | 1 | train |
| push-wall-v2 | 100000 | 15 | 100.0 | (39,) | (4,) | 1 | train |
| reach-v2 | 100000 | 15 | 100.0 | (39,) | (4,) | 1 | train |
| reach-wall-v2 | 100000 | 15 | 100.0 | (39,) | (4,) | 1 | train |
| shelf-place-v2 | 100000 | 15 | 100.0 | (39,) | (4,) | 1 | train |
| soccer-v2 | 100000 | 13 | 100.0 | (39,) | (4,) | 1 | train |
| stick-pull-v2 | 100000 | 15 | 100.0 | (39,) | (4,) | 1 | train |
| stick-push-v2 | 100000 | 15 | 100.0 | (39,) | (4,) | 1 | train |
| sweep-into-v2 | 100000 | 15 | 100.0 | (39,) | (4,) | 1 | train |
| sweep-v2 | 100000 | 15 | 100.0 | (39,) | (4,) | 1 | train |
| window-close-v2 | 100000 | 15 | 100.0 | (39,) | (4,) | 1 | train |
| window-open-v2 | 100000 | 15 | 100.0 | (39,) | (4,) | 1 | train |
| bin-picking-v2 | 100000 | 15 | 100.0 | (39,) | (4,) | 1 | test |
| box-close-v2 | 100000 | 15 | 100.0 | (39,) | (4,) | 1 | test |
| door-lock-v2 | 100000 | 15 | 100.0 | (39,) | (4,) | 1 | test |
| door-unlock-v2 | 100000 | 15 | 100.0 | (39,) | (4,) | 1 | test |
| hand-insert-v2 | 100000 | 15 | 100.0 | (39,) | (4,) | 1 | test |

Table 25: Dataset Metadata for Meta-World

| Task Name | Trajectory Length | Number of Trajectories | Mean Eplen. | Obs. Shape | Action Shape | Reward Scale | Split |
|---|---|---|---|---|---|---|---|
| def_S | 300184 | 10 | 232.9 | (275,) | (2,) | 1 | train |
| def_C_69 | 400192 | 10 | 301.1 | (275,) | (2,) | 1 | train |
| def_C_859 | 450235 | 10 | 283.1 | (275,) | (2,) | 1 | train |
| def_C_574 | 500242 | 10 | 263.3 | (275,) | (2,) | 1 | test |
| def_C_85 | 500302 | 10 | 286.1 | (275,) | (2,) | 1 | train |
| def_C_785 | 500188 | 10 | 293.8 | (275,) | (2,) | 1 | train |
| def_C_520 | 600292 | 10 | 258.4 | (275,) | (2,) | 1 | train |
| def_O_785 | 600207 | 10 | 364.8 | (275,) | (2,) | 1 | test |
| def_O_291 | 700325 | 10 | 357.4 | (275,) | (2,) | 1 | train |
| def_O_647 | 600323 | 10 | 350.7 | (275,) | (2,) | 1 | train |
| def_O_971 | 750438 | 10 | 355.5 | (275,) | (2,) | 1 | train |
| def_T_971 | 400184 | 10 | 365.6 | (275,) | (2,) | 1 | test |
| def_T_855 | 400192 | 10 | 320.2 | (275,) | (2,) | 1 | train |
| def_T_98 | 400188 | 10 | 363.7 | (275,) | (2,) | 1 | train |
| cones_S | 500224 | 10 | 451.8 | (275,) | (2,) | 1 | train |
| cones_C_69 | 700312 | 10 | 458.0 | (275,) | (2,) | 1 | test |
| cones_C_859 | 700433 | 10 | 461.5 | (275,) | (2,) | 1 | train |
| cones_C_574 | 800384 | 10 | 518.5 | (275,) | (2,) | 1 | train |
| cones_C_85 | 850386 | 10 | 532.8 | (275,) | (2,) | 1 | train |
| cones_C_785 | 800457 | 10 | 495.7 | (275,) | (2,) | 1 | train |
| cones_C_520 | 1000367 | 10 | 587.3 | (275,) | (2,) | 1 | train |

Table 26: Dataset Metadata for MetaDrive

| Task Name | Trajectory Length | Number of Trajectories | Mean Eplen. | Obs. Shape | Action Shape | Reward Scale | Split |
|---|---|---|---|---|---|---|---|
| 2022_phase_1_task_00 | 533628 | 10 | 729.0 | (44,) | (5,) | 0.25 | train |
| 2022_phase_1_task_01 | 533628 | 10 | 729.0 | (44,) | (5,) | 0.25 | train |
| 2022_phase_1_task_02 | 533628 | 10 | 729.0 | (44,) | (5,) | 0.35 | train |
| 2022_phase_1_task_03 | 533628 | 10 | 729.0 | (44,) | (5,) | 0.35 | train |
| 2022_phase_1_task_04 | 533628 | 10 | 729.0 | (44,) | (5,) | 0.3 | train |
| 2022_phase_1_task_05 | 533628 | 10 | 729.0 | (44,) | (5,) | 0.25 | train |
| 2022_phase_1_task_06 | 533628 | 10 | 729.0 | (44,) | (5,) | 0.3 | train |
| 2022_phase_1_task_07 | 533628 | 10 | 729.0 | (44,) | (5,) | 0.35 | train |
| 2022_phase_1_task_08 | 533628 | 10 | 729.0 | (44,) | (5,) | 0.4 | train |
| 2022_phase_1_task_09 | 533628 | 10 | 729.0 | (44,) | (5,) | 0.35 | train |
| 2022_phase_1_task_10 | 533628 | 10 | 718.7 | (44,) | (5,) | 0.3 | test |
| 2022_phase_1_task_11 | 533628 | 10 | 729.0 | (44,) | (5,) | 0.25 | test |
| 2022_phase_2_task_00 | 533628 | 10 | 729.0 | (44,) | (5,) | 0.25 | train |
| 2022_phase_2_task_01 | 533628 | 10 | 729.0 | (44,) | (5,) | 0.25 | train |
| 2022_phase_2_task_02 | 533628 | 10 | 729.0 | (44,) | (5,) | 0.35 | train |
| 2022_phase_2_task_03 | 533628 | 10 | 729.0 | (44,) | (5,) | 0.35 | train |
| 2022_phase_2_task_04 | 533628 | 10 | 729.0 | (44,) | (5,) | 0.2 | train |
| 2022_phase_2_task_05 | 533628 | 10 | 729.0 | (44,) | (5,) | 0.2 | train |
| 2022_phase_2_task_06 | 533628 | 10 | 729.0 | (44,) | (5,) | 0.25 | train |
| 2022_phase_2_task_07 | 533628 | 10 | 729.0 | (44,) | (5,) | 0.35 | train |
| 2022_phase_2_task_08 | 533628 | 10 | 729.0 | (44,) | (5,) | 0.4 | train |
| 2022_phase_2_task_09 | 533628 | 10 | 729.0 | (44,) | (5,) | 0.35 | train |
| 2022_phase_2_task_10 | 533628 | 10 | 729.0 | (44,) | (5,) | 0.3 | test |
| 2022_phase_2_task_11 | 533628 | 10 | 729.0 | (44,) | (5,) | 0.25 | test |

Table 27: Dataset Metadata for CityLearn

| Task Name | Trajectory Length | Number of Trajectories | Mean Eplen. | Obs. Shape | Action Shape | Reward Scale | Split |
|---|---|---|---|---|---|---|---|
| s-h0-weak-thrust-v1 | 100012 | 15 | 25.0 | (140,) | (6,) | 1 | train |
| s-h1-thrust-over-ball-v1 | 100014 | 15 | 22.3 | (140,) | (6,) | 1 | train |
| s-h2-one-wheel-car-v1 | 100104 | 15 | 209.3 | (140,) | (6,) | 1 | train |
| s-h3-point-the-thruster-v1 | 100019 | 15 | 90.1 | (140,) | (6,) | 1 | train |
| s-h4-thrust-aim-v1 | 100012 | 15 | 42.5 | (140,) | (6,) | 1 | train |
| s-h5-rotate-fall-v1 | 100013 | 15 | 27.0 | (140,) | (6,) | 1 | train |
| s-h6-unicycle-right-v1 | 100084 | 14 | 232.0 | (140,) | (6,) | 1 | train |
| s-h7-unicycle-left-v1 | 100074 | 14 | 191.9 | (140,) | (6,) | 1 | train |
| s-h8-unicycle-balance-v1 | 100012 | 15 | 25.1 | (140,) | (6,) | 1 | test |
| m-h11-thrust-left-easy-v1 | 100014 | 15 | 54.6 | (247,) | (6,) | 1 | train |
| m-h19-thrust-left-very-easy-v1 | 100082 | 15 | 138.0 | (247,) | (6,) | 1 | train |
| m-h5-angry-birds-v1 | 100015 | 15 | 72.7 | (247,) | (6,) | 1 | train |
| m-arm-right-v1 | 100007 | 15 | 46.7 | (247,) | (6,) | 1 | train |
| m-h12-thrustfall-left-v1 | 100012 | 15 | 41.6 | (247,) | (6,) | 1 | train |
| m-h7-car-flip-v1 | 100013 | 15 | 31.7 | (247,) | (6,) | 1 | train |
| m-h2-car-right-v1 | 100072 | 15 | 206.2 | (247,) | (6,) | 1 | train |
| m-h8-weird-vehicle-v1 | 100072 | 15 | 215.2 | (247,) | (6,) | 1 | train |
| m-h0-unicycle-v1 | 100021 | 15 | 29.4 | (247,) | (6,) | 1 | train |
| m-h14-thrustblock-v1 | 100017 | 15 | 27.3 | (247,) | (6,) | 1 | test |
| m-h3-car-thrust-v1 | 100019 | 15 | 54.2 | (247,) | (6,) | 1 | train |
| m-h9-spin-the-right-way-v1 | 100011 | 15 | 36.7 | (247,) | (6,) | 1 | train |
| m-arm-hard-v1 | 100009 | 15 | 46.4 | (247,) | (6,) | 1 | test |
| m-h10-thrust-right-easy-v1 | 100014 | 15 | 49.6 | (247,) | (6,) | 1 | train |
| m-h18-thrust-right-very-easy-v1 | 100047 | 15 | 175.6 | (247,) | (6,) | 1 | train |
| m-h4-thrust-the-needle-v1 | 100011 | 15 | 25.3 | (247,) | (6,) | 1 | train |
| m-arm-left-v1 | 100012 | 15 | 80.0 | (247,) | (6,) | 1 | train |
| m-arm-up-v1 | 100015 | 15 | 54.5 | (247,) | (6,) | 1 | train |
| m-h13-thrustfall-right-v1 | 100009 | 15 | 36.5 | (247,) | (6,) | 1 | train |
| m-h6-thrust-over-v1 | 100011 | 15 | 17.8 | (247,) | (6,) | 1 | train |
| l-chain-lander-v1 | 100018 | 15 | 33.8 | (679,) | (6,) | 1 | train |
| l-mjc-half-cheetah-v1 | 100095 | 15 | 194.1 | (679,) | (6,) | 1 | train |
| l-h7-maze-1-v1 | 100082 | 15 | 116.1 | (679,) | (6,) | 1 | train |
| l-pr-v1 | 100020 | 15 | 67.5 | (679,) | (6,) | 1 | train |
| l-h1-car-left-v1 | 100094 | 15 | 186.2 | (679,) | (6,) | 1 | train |
| l-hard-pinball-v1 | 100038 | 15 | 106.5 | (679,) | (6,) | 1 | test |
| l-mjc-swimmer-v1 | 100095 | 15 | 230.8 | (679,) | (6,) | 1 | train |
| l-h3-car-right-v1 | 100067 | 15 | 179.5 | (679,) | (6,) | 1 | train |
| l-mjc-half-cheetah-easy-v1 | 100101 | 15 | 234.2 | (679,) | (6,) | 1 | train |
| l-mjc-walker-easy-v1 | 100042 | 15 | 169.6 | (679,) | (6,) | 1 | train |
| l-chain-thrust-v1 | 100007 | 15 | 58.4 | (679,) | (6,) | 1 | train |
| l-mjc-walker-v1 | 100038 | 15 | 106.6 | (679,) | (6,) | 1 | test |
| l-h10-morph-direction-2-v1 | 100031 | 15 | 107.1 | (679,) | (6,) | 1 | train |
| l-h9-morph-direction-v1 | 100075 | 15 | 90.4 | (679,) | (6,) | 1 | train |
| l-h14-simple-thruster-v1 | 100014 | 15 | 28.4 | (679,) | (6,) | 1 | train |
| l-hard-lunar-lander-v1 | 100039 | 15 | 42.4 | (679,) | (6,) | 1 | test |
| l-mjc-hopper-easy-v1 | 100043 | 15 | 172.2 | (679,) | (6,) | 1 | train |
| l-mjc-hopper-v1 | 100018 | 15 | 65.3 | (679,) | (6,) | 1 | train |
| l-grasp-easy-v1 | 100003 | 15 | 30.4 | (679,) | (6,) | 1 | train |

Table 28: Dataset Metadata for Kinetix

| Task Name | Trajectory Length | Number of Trajectories | Mean Eplen. | Obs. Shape | Action Shape | Reward Scale | Split |
|---|---|---|---|---|---|---|---|
| industrial-benchmark-0-v1 | 300000 | 15 | 250.0 | (6,) | (3,) | 1 | train |
| industrial-benchmark-5-v1 | 300000 | 15 | 250.0 | (6,) | (3,) | 1 | train |
| industrial-benchmark-10-v1 | 300000 | 15 | 250.0 | (6,) | (3,) | 1 | train |
| industrial-benchmark-15-v1 | 300000 | 15 | 250.0 | (6,) | (3,) | 1 | train |
| industrial-benchmark-20-v1 | 300000 | 15 | 250.0 | (6,) | (3,) | 1 | train |
| industrial-benchmark-25-v1 | 300000 | 15 | 250.0 | (6,) | (3,) | 1 | train |
| industrial-benchmark-30-v1 | 300000 | 15 | 250.0 | (6,) | (3,) | 1 | train |
| industrial-benchmark-35-v1 | 300000 | 15 | 250.0 | (6,) | (3,) | 1 | train |
| industrial-benchmark-40-v1 | 300000 | 15 | 250.0 | (6,) | (3,) | 1 | train |
| industrial-benchmark-45-v1 | 300000 | 15 | 250.0 | (6,) | (3,) | 1 | train |
| industrial-benchmark-50-v1 | 300000 | 15 | 250.0 | (6,) | (3,) | 1 | train |
| industrial-benchmark-55-v1 | 300000 | 15 | 250.0 | (6,) | (3,) | 1 | train |
| industrial-benchmark-60-v1 | 300000 | 15 | 250.0 | (6,) | (3,) | 1 | train |
| industrial-benchmark-65-v1 | 300000 | 15 | 250.0 | (6,) | (3,) | 1 | train |
| industrial-benchmark-70-v1 | 300000 | 15 | 250.0 | (6,) | (3,) | 1 | train |
| industrial-benchmark-75-v1 | 300000 | 15 | 250.0 | (6,) | (3,) | 1 | train |
| industrial-benchmark-80-v1 | 300000 | 15 | 250.0 | (6,) | (3,) | 1 | test |
| industrial-benchmark-85-v1 | 300000 | 15 | 250.0 | (6,) | (3,) | 1 | test |
| industrial-benchmark-90-v1 | 300000 | 15 | 250.0 | (6,) | (3,) | 1 | test |
| industrial-benchmark-95-v1 | 300000 | 15 | 250.0 | (6,) | (3,) | 1 | test |
| industrial-benchmark-100-v1 | 300000 | 15 | 250.0 | (6,) | (3,) | 1 | test |

Table 29: Dataset Metadata for Industrial-Benchmark

| Task Name | Trajectory Length | Number of Trajectories | Mean Eplen. | Obs. Shape | Action Shape | Reward Scale | Split |
|---|---|---|---|---|---|---|---|
| REWARD-move-ego-0-0 | 300000 | 10 | 300.0 | (358,) | (69,) | 1 | train |
| REWARD-move-ego-0-2 | 300000 | 10 | 300.0 | (358,) | (69,) | 1 | train |
| REWARD-raisearms-h-h | 300000 | 10 | 300.0 | (358,) | (69,) | 1 | train |
| REWARD-crawl-0_5-0-d | 300000 | 10 | 300.0 | (358,) | (69,) | 1 | train |
| REWARD-crouch-0 | 300000 | 10 | 300.0 | (358,) | (69,) | 1 | train |
| REWARD-rotate-x–5-0_8 | 300000 | 10 | 300.0 | (358,) | (69,) | 1 | train |
| REWARD-rotate-x-5-0_8 | 300000 | 10 | 300.0 | (358,) | (69,) | 1 | test |
| REWARD-split-0_5 | 300000 | 10 | 300.0 | (358,) | (69,) | 1 | test |
| REWARD-headstand | 300000 | 10 | 300.0 | (358,) | (69,) | 1 | train |
| REWARD-move-ego-90-4 | 300000 | 10 | 300.0 | (358,) | (69,) | 1 | train |
| GOAL-t_pose | 300000 | 10 | 300.0 | (358,) | (69,) | 1 | test |
| GOAL-crouch_medium | 300000 | 10 | 300.0 | (358,) | (69,) | 1 | train |
| GOAL-zombie | 300000 | 10 | 300.0 | (358,) | (69,) | 1 | train |
| GOAL-lie_front | 300000 | 10 | 300.0 | (358,) | (69,) | 1 | train |
| GOAL-sit_hand_behind | 300000 | 10 | 300.0 | (358,) | (69,) | 1 | train |

Table 30: Dataset Metadata for HumEnv

| Task Name | Trajectory Length | Number of Trajectories | Mean Eplen. | Obs. Shape | Action Shape | Reward Scale | Split |
|---|---|---|---|---|---|---|---|
| aircraft_1 | 1000000 | 10 | 1000.0 | (3,) | (6,) | 0.005 | test |
| aircraft_2 | 1000000 | 10 | 1000.0 | (3,) | (6,) | 0.005 | train |
| aircraft_4 | 1000000 | 10 | 1000.0 | (2,) | (5,) | 0.02 | train |
| aircraft_5 | 1000000 | 10 | 1000.0 | (5,) | (11,) | 0.02 | train |
| aircraft_6 | 1000000 | 10 | 1000.0 | (5,) | (14,) | 0.0005 | train |
| cable_1 | 1000000 | 10 | 1000.0 | (2,) | (2,) | 0.01 | train |
| cable_2 | 1000000 | 10 | 1000.0 | (2,) | (2,) | 0.01 | train |
| cable_3 | 1000000 | 10 | 1000.0 | (2,) | (2,) | 0.01 | train |
| cable_4 | 1000000 | 10 | 1000.0 | (2,) | (2,) | 0.01 | train |
| cable_5 | 1000000 | 10 | 1000.0 | (2,) | (2,) | 0.01 | test |
| pde_convection_diffusion_reaction | 1000000 | 10 | 100.0 | (10,) | (1,) | 0.0077 | train |
| pde_schrodinger | 1000000 | 10 | 500.0 | (10,) | (1,) | 0.0001 | test |

Table 31: Dataset Metadata for ControlGym

| Task Name | Trajectory Length | Number of Trajectories | Mean Eplen. | Obs. Shape | Action Shape | Reward Scale | Split |
|---|---|---|---|---|---|---|---|
| 5zone-cool_dec_jan_feb | 140340 | 10 | 721.2 | (17,) | (2,) | 1 | train |
| 5zone-cool_jun_jul_aug | 140380 | 10 | 736.1 | (17,) | (2,) | 1 | train |
| 5zone-cool_mar_apr_may | 140438 | 10 | 736.4 | (17,) | (2,) | 1 | train |
| 5zone-cool_oct_nov | 140515 | 10 | 731.8 | (17,) | (2,) | 1 | train |
| 5zone-cool_sep | 140400 | 10 | 720.0 | (17,) | (2,) | 1 | train |
| 5zone-hot_dec_jan_feb | 140136 | 10 | 720.5 | (17,) | (2,) | 1 | train |
| 5zone-hot_jun_jul_aug | 140419 | 10 | 736.0 | (17,) | (2,) | 1 | train |
| 5zone-hot_mar_apr_may | 140527 | 10 | 736.1 | (17,) | (2,) | 1 | train |
| 5zone-hot_oct_nov | 140392 | 10 | 732.4 | (17,) | (2,) | 1 | train |
| 5zone-hot_sep | 140400 | 10 | 720.0 | (17,) | (2,) | 1 | train |
| 5zone-mixed_dec_jan_feb | 140388 | 10 | 720.7 | (17,) | (2,) | 1 | test |
| 5zone-mixed_jun_jul_aug | 140373 | 10 | 735.7 | (17,) | (2,) | 1 | test |
| 5zone-mixed_mar_apr_may | 140469 | 10 | 735.9 | (17,) | (2,) | 1 | test |
| 5zone-mixed_oct_nov | 140448 | 10 | 731.5 | (17,) | (2,) | 1 | test |
| 5zone-mixed_sep | 140400 | 10 | 720.0 | (17,) | (2,) | 1 | test |
| warehouse-cool_apr_may | 140474 | 10 | 732.4 | (22,) | (2,) | 1 | train |
| warehouse-cool_dec_jan_feb | 140215 | 10 | 718.8 | (22,) | (2,) | 1 | train |
| warehouse-cool_mar | 140616 | 10 | 744.0 | (22,) | (2,) | 1 | train |
| warehouse-cool_nov | 140400 | 10 | 720.0 | (22,) | (2,) | 1 | train |
| warehouse-cool_oct | 140616 | 10 | 744.0 | (22,) | (2,) | 1 | train |
| warehouse-hot_apr_may | 140364 | 10 | 732.2 | (22,) | (2,) | 1 | train |
| warehouse-hot_dec_jan_feb | 140323 | 10 | 719.2 | (22,) | (2,) | 1 | train |
| warehouse-hot_jun_jul_aug | 140412 | 10 | 735.9 | (22,) | (2,) | 1 | train |
| warehouse-hot_mar | 140616 | 10 | 744.0 | (22,) | (2,) | 1 | train |
| warehouse-hot_nov | 140400 | 10 | 720.0 | (22,) | (2,) | 1 | train |
| warehouse-hot_oct | 140616 | 10 | 744.0 | (22,) | (2,) | 1 | train |
| warehouse-hot_sep | 140400 | 10 | 720.0 | (22,) | (2,) | 1 | train |
| warehouse-mixed_apr_may | 140296 | 10 | 731.8 | (22,) | (2,) | 1 | test |
| warehouse-mixed_dec_jan_feb | 140428 | 10 | 720.2 | (22,) | (2,) | 1 | test |
| warehouse-mixed_jul_aug | 140616 | 10 | 744.0 | (22,) | (2,) | 1 | test |
| warehouse-mixed_jun | 140400 | 10 | 720.0 | (22,) | (2,) | 1 | test |
| warehouse-mixed_mar | 140616 | 10 | 744.0 | (22,) | (2,) | 1 | test |
| warehouse-mixed_sep | 140400 | 10 | 720.0 | (22,) | (2,) | 1 | test |

Table 32: Dataset Metadata for SinerGym

## J  DOES MULTI-DOMAIN HELP GENERALIZATION

To understand if multi-domain setting helps reach the better performance, we trained separate single-domain models for several representative domains under exactly the same conditions as the 10-domain model (identical architecture size, hyperparameters, and number of seen tokens per domain).

The results in the Table 33 show that the multi-domain model either outperforms or matches the single-domain models on both online and offline metrics, with particularly clear gains on Meta-World and Industrial Benchmark, and no systematic degradation elsewhere. This suggests that, despite using group-specific encoders, the shared backbone does exploit cross-domain data and yields measurable benefits over training separate models.

| Domain | Online Single | Online Full | Offline Single | Offline Full |
|---|---|---|---|---|
| Industrial Benchmark (Test) | $0.950 \pm 0.004$ | $\mathbf{0.980 \pm 0.007}$ | $0.950 \pm 0.007$ | $\mathbf{0.960 \pm 0.009}$ |
| Meta-World (Test) | $-0.000 \pm 0.001$ | $\mathbf{0.140 \pm 0.023}$ | $0.460 \pm 0.029$ | $\mathbf{0.690 \pm 0.036}$ |
| SinerGym (Test) | $0.900 \pm 0.026$ | $\mathbf{0.910 \pm 0.020}$ | $0.970 \pm 0.010$ | $0.960 \pm 0.017$ |
| Industrial Benchmark (Train) | $0.940 \pm 0.009$ | $\mathbf{0.990 \pm 0.006}$ | $1.010 \pm 0.001$ | $\mathbf{1.020 \pm 0.001}$ |
| Meta-World (Train) | $0.650 \pm 0.015$ | $\mathbf{0.850 \pm 0.080}$ | $0.990 \pm 0.001$ | $\mathbf{1.000 \pm 0.000}$ |
| SinerGym (Train) | $0.960 \pm 0.031$ | $\mathbf{1.000 \pm 0.026}$ | $\mathbf{1.010 \pm 0.004}$ | $0.990 \pm 0.017$ |

Table 33: Single-domain and multi-domain models comparison

## K  COMPARISON WITH GAUSSIAN HEADS

To ensure that Flow matching head is a suitable choice, we conducted an additional ablation comparing standard Gaussian policy heads (GH) with flow-matching heads (FM). We trained single-domain models with GH and with FM, keeping the number of parameters, batch size, optimization hyperparameters, and number of update steps identical for each pair of models. The results are summarized in the Table 34.

On unseen test tasks, FM heads perform at least as well as GH in three out of four domains, and substantially better in several cases: on SinerGym the offline test score improves from 0.790 to 0.970, on HumEnv from 0.070 to 0.130 and on Meta-World from 0.390 to 0.460. The only exception is Industrial Benchmark, where GH achieves slightly higher test performance than FM.

On training tasks, the picture is mixed: GH achieves higher scores in some cases (e.g., Industrial Benchmark and Meta-World), while FM is comparable or better in others (e.g., HumEnv, SinerGym). This suggests that GH can sometimes fit the training tasks more tightly, whereas FM heads tend to provide better or more robust generalization on held-out test tasks in several domains.

| Domain | Online GH | Online FM | Offline GH | Offline FM |
|---|---|---|---|---|
| HumEnv (Test) | $\mathbf{0.070 \pm 0.023}$ | $\mathbf{0.070 \pm 0.030}$ | $0.070 \pm 0.020$ | $\mathbf{0.130 \pm 0.020}$ |
| Industrial Benchmark (Test) | $\mathbf{0.990 \pm 0.003}$ | $0.950 \pm 0.004$ | $\mathbf{0.990 \pm 0.007}$ | $0.950 \pm 0.007$ |
| Meta-World (Test) | $0.000 \pm 0.001$ | $-0.000 \pm 0.001$ | $0.390 \pm 0.029$ | $\mathbf{0.460 \pm 0.034}$ |
| SinerGym (Test) | $0.400 \pm 0.087$ | $\mathbf{0.900 \pm 0.025}$ | $0.790 \pm 0.046$ | $\mathbf{0.970 \pm 0.010}$ |
| HumEnv (Train) | $0.440 \pm 0.025$ | $0.440 \pm 0.044$ | $0.440 \pm 0.025$ | $0.440 \pm 0.024$ |
| Industrial Benchmark (Train) | $\mathbf{1.020 \pm 0.001}$ | $0.940 \pm 0.008$ | $\mathbf{1.020 \pm 0.001}$ | $1.010 \pm 0.001$ |
| Meta-World (Train) | $\mathbf{0.960 \pm 0.002}$ | $0.650 \pm 0.014$ | $0.960 \pm 0.002$ | $\mathbf{0.990 \pm 0.001}$ |
| SinerGym (Train) | $0.920 \pm 0.009$ | $\mathbf{0.960 \pm 0.030}$ | $0.920 \pm 0.009$ | $\mathbf{0.970 \pm 0.010}$ |

Table 34: Gaussian and Flow Matching Heads single-domain models comparison

## L   COMPARISON WITH VINTIX ON ALL DOMAINS

While Vintix was trained and evaluated on only 4 domains, we were interested in a full comparison across all 10 domains in test set. The per-domain results are presented in Table 35. To ensure a fair comparison, we matched Vintix's model size to our setup. As shown, Vintix performs substantially worse than our model.

| Domain | Vintix Online | Vintix II Online | Vintix Offline | Vintix II Offline |
|---|---|---|---|---|
| Bi-DexHands | $-0.45 \pm 0.08$ | $\mathbf{-0.43 \pm 0.08}$ | $-0.42 \pm 0.08$ | $\mathbf{0.07 \pm 0.05}$ |
| CityLearn | $0.01 \pm 0.02$ | $\mathbf{0.78 \pm 0.00}$ | $0.77 \pm 0.00$ | $\mathbf{0.78 \pm 0.00}$ |
| ControlGym | $0.97 \pm 0.03$ | $\mathbf{0.99 \pm 0.31}$ | $1.00 \pm 0.00$ | $1.00 \pm 0.00$ |
| HumEnv | $0.04 \pm 0.06$ | $\mathbf{0.09 \pm 0.02}$ | $0.06 \pm 0.02$ | $\mathbf{0.07 \pm 0.01}$ |
| Industrial Benchmark | $0.42 \pm 0.05$ | $\mathbf{0.98 \pm 0.00}$ | $0.86 \pm 0.05$ | $\mathbf{0.96 \pm 0.01}$ |
| Kinetix | $0.20 \pm 0.05$ | $\mathbf{0.23 \pm 0.11}$ | $0.17 \pm 0.05$ | $\mathbf{0.23 \pm 0.05}$ |
| Meta-World | $0.05 \pm 0.01$ | $\mathbf{0.14 \pm 0.02}$ | $0.09 \pm 0.01$ | $\mathbf{0.69 \pm 0.03}$ |
| MetaDrive | $1.02 \pm 0.00$ | $1.02 \pm 0.21$ | $1.02 \pm 0.00$ | $1.02 \pm 0.00$ |
| MuJoCo | $0.98 \pm 0.01$ | $\mathbf{1.00 \pm 0.00}$ | $1.00 \pm 0.00$ | $1.00 \pm 0.00$ |
| SinerGym | $0.04 \pm 0.06$ | $\mathbf{0.86 \pm 0.07}$ | $0.08 \pm 0.02$ | $\mathbf{0.92 \pm 0.02}$ |

Table 35: Comparison between Vintix and Vintix II in online and offline settings across domains on test set

