# OpenReview forum: "Vintix II: Decision Pre-Trained Transformer is a Scalable In-Context Reinforcement Learner"
_ICLR.cc/2026/Conference — ICLR 2026 Poster_

### Official Review · Reviewer_KAK3 · 2025-10-27

**Soundness:** 3
**Presentation:** 3
**Contribution:** 3
**Rating:** 6
**Confidence:** 3

**Summary:**

This paper extends the Decision Pre-Trained Transformer (DPT), a large-scale transformer model, to diverse reinforcement learning domains using a decision-transformer-style objective. DPT integrates a flow-matching policy head to better model multi-modal continuous-action distributions and is trained on a large multi-domain dataset. The authors evaluate DPT both in offline settings (fixed demonstration contexts) and online settings (autoregressive conditioning on recent transitions). The authors publish the used large-scale dataset with over 700m transitions. The paper also draws a connection, empirically, to the posterior-sampling interpretation of in-context learning, suggesting that DPT behaves as a Bayesian in-context learner following the reasoning of Lee et al. (2023). Empirically, DPT achieves strong performance across most domains and outperforms existing generalist RL models such as Vintix and REGENT on several benchmarks.

**Strengths:**

- The paper is well-written, well-structured, and easy to follow. The dataset composition, architecture, and training setup are described clearly, and the figures are of high quality.
- The scale of the dataset and experiments is high. The construction of the model, as well as the dataset, covering 10 domains is a significant engineering effort. The resulting benchmark could be a valuable community resource.
- Replacing the Gaussian policy head with a flow-matching head for continuous actions is a natural and elegant design choice.
- DPT performs consistently well on both seen and unseen tasks, with significant gains in the more complex domains.

**Weaknesses:**

**Novelty**:

The paper would benefit from a clearer articulation of what exactly differentiates DPT from prior large-scale decision-transformer architectures such as REGENT and Vintix. Currently, the novelty seems to lie mainly in the use of the flow-matching policy head and the dataset scale. A comparison table summarizing architectural and algorithmic differences (policy head, encoding choice, inference procedure, etc.) would make the contributions clearer.

**Ablation studies**:

I would appreciate more detailed ablations that would allow a more finegrained assessment of what architectural choices contribute most to performance. Primarily the choice of using a flow-matching algorithm for policy heads comes to mind here, but also the choice of embeddings and context ordering, or context length.

**Experiments**:

- it seems that the selection of experiments makes it somewhat hard to draw quantitative conclusions from the chosen implementation choices, because most of the included domains appear to be solves by all tested models. Only few (Meta-World and Bi-DexHands) provide sufficient difficulty to distinguish algorithms.

- Although the results are strong, I could not infer whether all baselines were re-tuned on the same dataset and training budget. In particular, Vintix and REGENT may have been evaluated under different protocols. Clarifying this would improve the fairness of the comparison.

**Questions:**

- Could you provide ablations isolating the contribution of the flow-matching policy head relative to standard Gaussian heads?
- Could you clarify the experimental protocol for the baselines? Were baselines retrained on your dataset or reproduced from prior work?
- Do you plan to release the code publicly to support reproducibility and benchmarking?

---

> ### Author Response · Authors · 2025-11-24
> **Official Comment**
>
> We thank the reviewer for their positive and detailed feedback.
>
> # Addressing weaknesses and questions
>
> ## W1. Novelty
>
> Thank you for pointing out that the difference between our model and existing approaches may not be sufficiently clear, which can make it harder to assess the novelty of our work. Let us clarify on that.
>
> Our architecture is based on a fundamentally different algorithm from Vintix and REGENT, both in the learning objective and in how context is used for adaptation. DPT is trained to predict near-optimal actions given a query state and an in-context dataset of trajectories, with a direct Bayesian posterior–sampling interpretation, whereas Vintix is built on Algorithm Distillation with noise-distilled learning histories that implicitly imitate a policy improvement operator, and REGENT is a retrieval-augmented policy that conditions on nearest-neighbor demonstrations rather than full trial-and-error histories.
>
> We do not consider adding the flow-matching head to be the main novelty of our work, but rather a suitable design choice for adapting DPT to continuous-action environments. We consider our main contributions are:
>
> 1. **Scaling up DPT as a basis for Large Action Models.**
>
>     The field of Large Action Models is promising but currently challenging. The dominant approach is to use straightforward expert-distillation, which in our view lacks of adaptation abilities.  Prior ICRL work with DPT has mostly been benchmarked on a small number of toy or single-domain tasks; our work is a concrete step toward making such models work in a genuinely multi-domain setting.
>
> 2. **Collecting and releasing a diverse cross-domain dataset.**
>
>     The ICRL community currently has only a limited number of publicly available large-scale datasets. We collect and standardize a substantially larger cross-domain dataset (over 700M transitions across 10 domains, 209 training tasks and 46 held-out tasks), which we will release. We view this as an important practical contribution, as data availability is often the main bottleneck for progress in this area.
>
> 3. **Improving performance of existing action models on unseen tasks.**
>
>     Compared to existing action models such as Vintix and REGENT, our scaled DPT model achieves better performance on unseen tasks in several domains. We do not claim dramatic gains everywhere, but we do obtain improvements on the challenging problem of new-task adaptation, which we see as a meaningful step forward for Large Action Models. We are not aware of any results in offline in-context reinforcement learning demonstrating such a performance.
>
>
> ## W2. Q1. **Ablation studies**
>
> We agree that more fine-grained ablations would be valuable and that our work would benefit from them. However, a systematic disentangling of all architectural choices in the full 10-domain setting is computationally expensive and not feasible during rebuttal.
>
> In practice, we set the context length to the maximum that fit within our memory and compute budget, and we selected other architectural choices (embeddings, context ordering, etc.) based on preliminary experiments at smaller scales (single domains, smaller models), where they consistently performed well.
>
> The choice of a flow-matching head over a simple Gaussian head is driven by its ability to represent richer, multi-modal continuous action distributions, as supported by prior work on flow-matching in generative modeling. We are preparing the experiment for comparison of these to architectures on separate domains, will share the results of experiments in a few days.
>
> ## W3. Q2. **Experiments**
>
> We thank the reviewer for raising this point. For the baselines, we report the results stated in the original works. All compared methods are evaluated on the same benchmark environments under the protocols defined in their respective papers, so we believe the comparison is still meaningful and fair.
>
> ## Q3. The code release
>
> Yes, we are preparing to publicly release the training and inference code, as well as the model checkpoints. We plan to share an anonymous GitHub link here within two days, as we need a short amount of time to clean the code for anonymization.

---

> > ### Author Response · Authors · 2025-11-27
> > **Official Comment 2**
> >
> > Here is the link to our anonymised code: https://anonymous.4open.science/r/iclr2026-C5B5/README.md

---

> > ### Author Response · Authors · 2025-12-01
> > **Official Comment 3**
> >
> > Dear reviewer,
> >
> > Based on your concerns, we conducted an additional ablation comparing standard Gaussian policy heads (GH) with flow-matching heads (FM). We trained single-domain models with GH and with FM, keeping the number of parameters, batch size, optimization hyperparameters, and number of update steps identical for each pair of models. The results are summarized in the table below.
> > On unseen **test tasks**, FM heads perform at least as well as GH in three out of four domains, and substantially better in several cases:  on SinerGym the offline test score improves from 0.790 to 0.970,  on HumEnv from 0.070 to 0.130 and on Meta-World from 0.390 to 0.460. The only exception is Industrial Benchmark, where GH achieves slightly higher test performance than FM.
> >
> > On **training tasks**, the picture is mixed: GH achieves higher scores in some cases (e.g., Industrial Benchmark and Meta-World), while FM is comparable or better in others (e.g., HumEnv, SinerGym). This suggests that GH can sometimes fit the training tasks more tightly, whereas FM heads tend to provide better or more robust generalization on held-out test tasks in several domains.
> >
> > Since our main objective is strong performance on unseen test tasks, these ablations support the choice of flow-matching heads over standard Gaussian heads in our setting.
> >
> > | Domain | Online Test GH | Online Test FM | Offline Test GH | Offline Test FM | Online Train GH | Online Train FM | Offline Train GH | Offline Train FM |
> > | --- | :---: | :---: | :---: | :---: | :---: | :---: | :---: | :---: |
> > | HumEnv | **0.070 ± 0.023** | **0.070 ± 0.030** | 0.070 ± 0.020 | **0.130 ± 0.020** | **0.440 ± 0.025** | **0.440 ± 0.044** | **0.440 ± 0.025** | **0.440 ± 0.024** |
> > | Industrial Benchmark | **0.990 ± 0.003** | 0.950 ± 0.004 | **0.990 ± 0.007** | 0.950 ± 0.007 | **1.020 ± 0.001** | 0.940 ± 0.008 | **1.020 ± 0.001** | 1.010 ± 0.001 |
> > | Meta-World | **0.000 ± 0.001** | **-0.000 ± 0.001** | 0.390 ± 0.029 | **0.460 ± 0.034** | **0.960 ± 0.002** | 0.650 ± 0.014 | 0.960 ± 0.002 | **0.990 ± 0.001** |
> > | SinerGym | 0.400 ± 0.087 | **0.900 ± 0.025** | 0.790 ± 0.046 | **0.970 ± 0.010** | 0.920 ± 0.009 | **0.960 ± 0.030** | 0.920 ± 0.009 | **0.970 ± 0.010** |

---

### Official Review · Reviewer_FmEz · 2025-10-31

**Soundness:** 2
**Presentation:** 3
**Contribution:** 2
**Rating:** 4
**Confidence:** 4

**Summary:**

This work presents an in-context reinforcement learning (ICRL) method that extends the Decision Pre-trained Transformer (DPT) architecture with flow matching to model complex output distributions. An empirical study on a large cross-domain dataset demonstrates the improved performance of this design compared to baselines.

**Strengths:**

- The writing is clear and detailed.
- The empirical study is comprehensive and fair.
- The method improves upon the original DPT without too much overhead or modification.

**Weaknesses:**

- The paper lacks a background section to formalize RL, ICRL, and flow matching, which are the focus of this work.
- I don't think the removal of the next observation $o'$ is well-justified, as it is crucial for capturing the dynamics of the environment, especially when the context is randomly permutated. Without $o'$, RL algorithms that optimize the policy based on the reward signal would not work unless it's a bandit setting. In this case, the DPT is likely merely learning by imitation and using the contextual information for task identification. Even in imitation learning, I doubt removing the next observation will leave the learner intact because, as I pointed out previously, $o'$ is crucial for capturing the environment dynamics. I suspect that the authors' claim that removing it will not impact the performance is due to that the dynamics of the testbeds are somewhat consistent across tasks.

**Questions:**

- What is the source of the optimal actions used for training?

---

> ### Author Response · Authors · 2025-11-24
> **Official Comment**
>
> We thanks reviewer for their time and constructive feedback.
>
> # Addressing weaknesses and questions
>
> ## W1. Lack of background section
>
> We thank the reviewer for this comment and agree that a dedicated background section on RL, ICRL, and flow matching would improve clarity. We are currently preparing this section and will add it to the manuscript in a few days.
>
> ## W2. Justifying the removal of o’
>
> We thank the reviewer for raising this concern, indeed it deserves a more careful consideration. We adopted the this architectural choice, following the original DPT paper [1], which removed o′ from the in-context dataset for the MiniWorld environments (Appendix A.5) to reduce the computation.
>
> In our setting, we did not observe a measurable degradation in performance when excluding o′, which is why we kept this choice in the current version. We do not intend to claim that removing o′ is universally harmless.
>
> For a better ablation, it is indeed preferable to train models on the full (o,a,r,o′) tuples. However, retraining our largest 10-domain model with o′ included is computationally infeasible within the rebuttal period. We are currently running additional experiments on smaller per-domain models trained with and without o′ to quantify its impact, will share the results in a few days.
>
> ## Q1. Source of optimal actions
>
> As in the original DPT paper, we trained separate models for each training task to obtain the expert actions. More information can be found in Appendix C.
>
> [1] Jonathan N. Lee, Annie Xie, Aldo Pacchiano, Yash Chandak, Chelsea Finn, Ofir Nachum, and Emma Brunskill. Supervised pretraining can learn in-context reinforcement learning, 2023

---

> ### Comment · Area_Chair_vBW3 · 2025-11-26
>
> The evaluation of this submission has a large variance so we need to gather more information from both the authors and the reviewers. Please take a look at the author's response and see how it affects your evaluation.
>
> Best,
> AC

---

> ### Author Response · Authors · 2025-12-01
> **Official Comment 2**
>
> Dear reviewer,
>
> Based on your concerns, we conducted an additional experiment in which we added the next observation o′ back into the transitions. We trained single-domain models **with** and **without** o′ in the context, keeping the number of parameters, batch size, optimization hyperparameters, and number of update steps identical for each pair of models. The results are summarized in the table below. As can be seen, adding o′ does not lead to a consistent performance improvement across domains and settings: in some cases the variant with o′ performs slightly better, in others the variant without o′ is better, and the differences are generally small.
>
> | Domain | Online Test w o' | Online Test w/o o' | Offline Test w o' | Offline Test w/o o' | Online Train w o' | Online Train w/o o' | Offline Train w o' | Offline Train w/o o' |
> | --- | :---: | :---: | :---: | :---: | :---: | :---: | :---: | :---: |
> | HumEnv | 0.050 ± 0.018 | **0.070 ± 0.032** | **0.140 ± 0.023** | 0.130 ± 0.021 | 0.430 ± 0.026 | **0.440 ± 0.045** | 0.430 ± 0.026 | **0.440 ± 0.024** |
> | Industrial Benchmark | **0.950 ± 0.006** | **0.950 ± 0.004** | 0.930 ± 0.007 | **0.950 ± 0.006** | **1.010 ± 0.001** | 0.940 ± 0.009 | **1.010 ± 0.001** | **1.010 ± 0.001** |
> | Meta-World | **0.060 ± 0.022** | -0.000 ± 0.001 | 0.350 ± 0.032 | **0.460 ± 0.030** | **0.990 ± 0.001** | 0.650 ± 0.014 | **0.990 ± 0.001** | **0.990 ± 0.001** |
> | SinerGym | **0.920 ± 0.053** | 0.900 ± 0.025 | 0.950 ± 0.009 | **0.970 ± 0.010** | **1.000 ± 0.005** | 0.960 ± 0.028 | **1.000 ± 0.005** | 0.970 ± 0.010 |

---

> ### Author Response · Authors · 2025-12-01
> **Official Comment 3**
>
> Dear reviewer,
>
> Based on your suggestion, we added a background section in **Appendix I in the end of the manuscript**. We plan to expand the main text with this material in the camera-ready version, where additional space will be available.

---

### Official Review · Reviewer_3WLW · 2025-10-31

**Soundness:** 1
**Presentation:** 2
**Contribution:** 2
**Rating:** 2
**Confidence:** 3

**Summary:**

The paper presents the Decision Pre-Trained Transformer (DPT) with a flow-matching generative policy head as a scalable agent for cross-domain, in-context reinforcement learning. The authors create a new large, cross-domain dataset featuring over 700 million transitions across 10 domains, and successfully train the DPT agent to achieve high performance on unseen tasks in several domains.

**Recommendation:**\
I recommend to reject. The paper is lacking in several fundamental areas, including the following: comparison with baselines, clarity of research methodology, and unconvincing results.

**Strengths:**

- The analysis in Section 4.4 is potentially interesting.
- A sizeable new cross-domain benchmark and dataset is created.
- High performance on unseen tasks in 6 out of 10 domains.

**Weaknesses:**

- Very little comparison with baselines
- No mention of the significance of the results. Not mentioning number of seeds, the standard deviation or confidence intervals for any methods, makes it impossible to judge any level of significance.
- Unclear research methodology. There is no mention of hyperparameter tuning or strict validation - test set splits.
- Reproducability is lacking. The experimental details are very lacklustre.
- Results not that strong. Only on a single domain (Meta-World) out of 10 does the performance on unseen tasks appear to be better than one of the baselines (Vintix), whilst appearing to be the same as the other baseline (REGENT). Furthermore, performance in the online and offline setting are overlapping in most domains, suggesting it doesn't actually learn in-context at all. This is corroborated by the mostly flat curves in Figure 5.
- Overclaiming in the introduction.
- Lack of background section makes it difficult to judge novelty.

**Questions:**

- Which contributions are novel, and in what way exactly?
- How was hyperparameter tuning performed? And how was contamination between validation and test sets avoided?
- Several potential baselines were mentioned in the related works section, why did you not compare to them? Why are the existing baselines only evaluated on very few of the domains?
- How did you test for significance of the results?
- Figure 3 shows mostly overlapping performance between the online and offline setting, and Figure 4 shows mostly no change in performance as context size increases. Do these two things suggest your agent is not actually learning in-context?
- Figure 4 shows concentration of the action distribution for longer contexts, but how does this actually relate to performance?


**Things to improve that did not impact decision:**
- Figure 2: What do the red lines and shaded regions indicate?
- Figure 3: How is the performance for the online setting defined?
- Line 437: Performance on the Industrial-Benchmark domain does not improve significantly with prompt size as the shaded regions are completely overlapping.
- If each domain requires its own pre-trained encoder, is it really a cross-domain setting?

---

> ### Author Response · Authors · 2025-11-24
> **Official Comment 1**
>
> We thank the reviewer for the careful consideration of our work. Below we address each weakness and question one by one.
>
> # Addressing weaknesses and questions
>
> ## W1. Q3. Insufficient baseline comparison
>
> We agree that a broader baseline comparison would further strengthen the paper, but there are several reasons for the current setup.
>
> The only overlapping domains with JAT and Gato are Meta-World and MuJoCo. In those works, all Meta-World tasks (MT50) are included in the training set, whereas we follow a more rigorous and meta-learning oriented ML45-style split with 5 tasks kept unseen for the model and also vary environment parameters in MuJoCo. Since one of our main goals is to study adaptation to unseen test tasks, a direct comparison on our test split is therefore not well aligned with their setting. However, our model can still be compared to JAT-/Gato-style baselines on the shared training tasks, and we will add these training-task comparisons to the appendix of the final manuscript.
>
> To strengthen the baseline section, we additionally trained Vintix on our full 10-domain dataset. We present the corresponding results on the test set in the table below. While we agree that a more thorough hyperparameter tuning of Vintix would be ideal, this is not feasible within the rebuttal period; even so, Vintix shows clearly weaker performance than our model. These results will be incorporated into the final manuscript.
>
> | Domain | Vintix Online | Ours Online | Vintix Offline | Ours Offline |
> | --- | --- | --- | --- | --- |
> | Bi-DexHands | -0.45 +- 0.08 | **-0.43 +- 0.08** | -0.42 +- 0.08 | **0.07 +- 0.05** |
> | CityLearn | 0.01 +- 0.02 | **0.78 +- 0.00** | 0.77 +- 0.00 | **0.78 +- 0.00** |
> | ControlGym | 0.97 +- 0.03 | **0.99 +- 0.31** | 1.00 +- 0.00 | 1.00 +- 0.00 |
> | HumEnv | 0.04 +- 0.06 | **0.09 +- 0.02** | 0.06 +- 0.02 | **0.07 +- 0.01** |
> | Industrial Benchmark | 0.42 +- 0.05 | **0.98 +- 0.00** | 0.86 +- 0.05 | **0.96 +- 0.01** |
> | Kinetix | 0.20 +- 0.05 | **0.23 +- 0.11** | 0.17 +- 0.05 | **0.23 +- 0.05** |
> | Meta-World | 0.05 +- 0.01 | **0.14 +- 0.02** | 0.09 +- 0.01 | **0.69 +- 0.03** |
> | MetaDrive | 1.02 +- 0.00 | 1.02 +- 0.21 | 1.02 +- 0.00 | 1.02 +- 0.00 |
> | MuJoCo | 0.98 +- 0.01 | **1.00 +- 0.00** | 1.00 +- 0.00 | 1.00 +- 0.00 |
> | SinerGym | 0.04 +- 0.06 | **0.86 +- 0.07** | 0.08 +- 0.02 | **0.92 +- 0.02** |
>
> Finally, the training setup of REGENT is quite different from ours, and adapting REGENT to our setup would require substantial additional implementation and tuning that is not realistic within rebuttal time. We agree that such an extended comparison would be interesting and leave it as future work.
>
> ## W2. Q4. Significance of the results
>
> We would like to clarify that Figure 2 and Figure 5 already report performance with confidence intervals, and the number of seeds is specified in the captions. We used the recent widely accepted statistical instruments provided by Agrawal [1] (NeurIPS 2021).
>
> All results in Appendix G are also presented with confidence intervals. We agree that Figure 3 would benefit from showing uncertainty, but adding CIs directly to that plot would make it visually cluttered given the number of curves. To address this while keeping the figure readable, we will add a table that mirrors Figure 3 and reports the corresponding values with confidence intervals on the test set.
>
> We have carefully rechecked our manuscript, and to the best of our knowledge, all other quantitative results already include CIs. However, if you believe there are specific plots or tables where CIs are missing, we would be happy to add them in the final version.
>
> ## W3. Q2. Experimental setup and Reproducability
>
> We thank the reviewer for highlighting that our hyperparameter-tuning protocol was under-specified. In practice, we performed a small grid search over learning rate, weight decay, and the number of warmup steps for the cosine scheduler.
>
> For tuning, we carved out a validation split from the training data by selecting several training tasks per domain and using their trajectories exclusively for validation in those runs. Each candidate configuration was trained for a reduced number of epochs, and we selected the best setting based on performance on this validation split. The final test sets (tasks and trajectories) were never used for model or hyperparameter selection, so there is no contamination between validation and test. We will add the details described above to the revised manuscript.
>
> To further support reproducibility, we are preparing the training and inference code for release. We will share an anonymous GitHub link here within two days, as we need a short amount of time to clean the code for anonymization.
>
> [1] Rishabh Agarwal, Max Schwarzer, Pablo Samuel Castro, Aaron Courville, and Marc G Bellemare. Deep reinforcement learning at the edge of the statistical precipice.

---

> ### Author Response · Authors · 2025-11-24
> **Official Comment 2**
>
> ## W4. Q5. Results clarification
> We thank the reviewer for the careful reading and the opportunity to clarify the results. In the paper, we compare to Vintix on the four domains where it was originally trained. On these, our model outperforms Vintix on Meta-World and BiDexHands, while Mujoco and Industrial Benchmark are already at a very high performance level for Vintix and our model essentially matches it there.
>
> As described above, when we re-train Vintix on all 10 domains under our setup, our model also achieves higher scores on SinerGym and CityLearn. But our main goal is not to claim per-domain SOTA everywhere, but to show that a DPT-based large action model is competitive across many heterogeneous domains while supporting adaptation to new tasks.
>
> Regarding in-context learning, the “online” score in our plots is measured at the point where performance has saturated after trial-and-error interaction. The overlap between online and offline curves therefore indicates that the model can adapt online (from zero expert trajectories provided) up to the level achieved when expert demonstrations are directly provided in context, not that it fails to learn.
>
> Finally, Figure 5 shows offline scaling: we vary the number of expert transitions in a fixed context without any online interaction. Flat curves in this figure simply mean that the model already reaches its asymptotic performance with relatively few expert transitions, rather than an absence of in-context learning.
>
> ## W6. Lack of background
> We thank the reviewer for this comment and agree that a dedicated background section would make it easier to assess our novelty.  We are currently preparing this section and will add it to the manuscript in a few days.
>
> ## Q1. Novelty clarification.
> We appreciate the reviewer’s question and clarify our novelty more explicitly below:
> 1. **Scaling up DPT as a basis for Large Action Models.**
>     The field of Large Action Models is promising but currently challenging. The dominant approach is to use straightforward expert-distillation, which in our view lacks of adaptation abilities.  Prior ICRL work with DPT has mostly been benchmarked on a small number of toy or single-domain tasks; our work is a concrete step toward making such models work in a genuinely multi-domain setting.
>
> 2. **Collecting and releasing a diverse cross-domain dataset.**
>     The ICRL community currently has only a limited number of publicly available large-scale datasets. We collect and standardize a substantially larger cross-domain dataset (over 700M transitions across 10 domains, 209 training tasks and 46 held-out tasks), which we will release. We view this as an important practical contribution, as data availability is often the main bottleneck for progress in this area.
>
> 3. **Improving performance of existing action models on unseen tasks.**
>     Compared to existing action models such as Vintix and REGENT, our scaled DPT model achieves better performance on unseen tasks in several domains. We do not claim dramatic gains everywhere, but we do obtain improvements on the challenging problem of new-task adaptation, which we see as a meaningful step forward for Large Action Models. We are not aware of any results in offline in-context reinforcement learning demonstrating such a performance.
>
> ## Q6. Influence of action distribution on performance
> We thank the reviewer for this question and agree that the connection should be made explicit. We ran an additional evaluation where we varied the context length and measured the model’s performance under the same setup as in Figure 4.
> The results (https://postimg.cc/dk9vH8fr) indicate that performance systematically improves as the context length increases, matching the trend of increased concentration in the action distribution.
>
> ## Additional questions:
> 1. The red lines indicate the maximum achieved performance, and the shaded regions show the confidence intervals for the evaluations.
> 2. In the online setting, performance is defined as the mean score over episodes after the model reaches saturation and stops improving. We will state this explicitly in the text.
> 3. Thank you for pointing this out; we will revise the text to avoid overclaiming.
> 4. The question of how to build a universal encoder for different-sized observations and actions remains an open challenge. In our case, the model has a shared transformer backbone that contains information about each domain, which is why we view it as cross-domain. An analogy can be made with CV models from a few years ago: a ResNet backbone is used, while the final layers are substituted to solve specific tasks. The full model still contains knowledge about many different images, but specific connectors are needed for different types of tasks. Similarly, current RL models interacting with different domains require different connectors; we see this as a limitation of the current stage of the field that we hope will be addressed in future work.

---

> > ### Comment · Reviewer_3WLW · 2025-11-25
> >
> > I thank the authors for their thoughtful answers. I believe the paper has potential, but requires significant revisions (some of which have already been clarified during rebuttal). Therefore, I recommend that it is resubmitted at a future venue so that it can go through the full reviewing process again. As such, I will keep my score.
> >
> > Perhaps the authors could consider submitting it to a Benchmark & Dataset track, since the largest contribution seems to be the cross-domain dataset.

---

> > > ### Author Response · Authors · 2025-11-25
> > >
> > > Thank you for the prompt reply.
> > >
> > > Could you please clarify what significant revisions are needed in you opinion? We believe that we addressed most of them if not all. This can help us to make the work better (even if not for the current iteration), right now, we are a bit lost in what are exactly the problems you feel are required to be addressed.

---

> > > > ### Comment · Reviewer_3WLW · 2025-11-25
> > > >
> > > > I am referring to the weaknesses as brought up by me and the other reviewers. In particular, the need for improved clarity of the contributions, details of experimental methodology, and the additional comparison with baselines. These have been addressed to some extent in the rebuttal, but I believe these are revisions to fundamental, core parts of the paper, that require careful review, which means I recommend the paper to be revised and resubmitted at a future venue.

---

> > > > > ### Author Response · Authors · 2025-11-27
> > > > > **Official comment 3**
> > > > >
> > > > > Here is the link to our anonymised code: https://anonymous.4open.science/r/iclr2026-C5B5/README.md

---

> ### Author Response · Authors · 2025-12-01
> **Official comment 4**
>
> Dear reviewer,
>
> Based on your suggestion, we added a background section in **Appendix I in the end of the manuscript**. We plan to expand the main text with this material in the camera-ready version, where additional space will be available.

---

### Official Review · Reviewer_rRdx · 2025-11-01

**Soundness:** 3
**Presentation:** 3
**Contribution:** 2
**Rating:** 4
**Confidence:** 5

**Summary:**

This paper extends pretrained Decision Transformer models to diverse multi-domain environments and integrates a flow-matching objective for action generation. The authors evaluate on 209 tasks across 10 domains and report that their approach (DPT with flow matching) outperforms baseline methods.

**Strengths:**

- The paper is clearly written and easy to follow.
 - The topic of leveraging past interaction data for generalizable policy learning is important and relevant to the community.

**Weaknesses:**

- The primary contribution appears to be the incorporation of a flow-matching objective into a Decision Transformer framework. However, flow-matching and related diffusion-style generation objectives have been extensively explored in prior work, which makes the contribution feel incremental.
 - The paper does not clearly specify the types of observations used in each domain. Since the tasks span diverse settings, it is important to clarify whether inputs are proprioceptive states, images, or mixed modalities. If the experiments rely solely on low-dimensional proprioceptive inputs, the significance and applicability to high-dimensional tasks may be limited.
 - Although the motivation emphasizes multi-domain generalization, the architecture groups tasks and encodes them using group-specific encoders. This raises the question of how much knowledge is actually shared across domains. A comparison with models trained on single-domain data would help clarify whether multi-domain training provides measurable benefit.

**Questions:**

- What observation modalities are used in each domain? Are the inputs entirely proprioceptive, or do any domains include image-based observations?
 - How does the performance of the multi-domain model compare to models trained separately for each domain? Is there measurable improvement from cross-domain data sharing?
 - Could the authors provide experiments or discussion on applying the model to high-dimensional visual inputs?

---

> ### Author Response · Authors · 2025-11-24
> **Official Comment**
>
> We appreciate reviewer for careful reading and constructive feedback.
>
> # Addressing weaknesses and questions
>
> ## W1. Incremental contribution
>
> We agree that flow-matching and related diffusion-style objectives are well studied, and we do not intend to present “flow-matching inside a Decision Transformer” as our main novelty.
>
> Our primary contribution is scaling the Decision Pretrained Transformer framework to a complex cross-domain setting by training a single large action model across many heterogeneous continuous-control domains, whereas most prior In-Context Reinforcement Learning works evaluate on only a small number of closely related tasks. We believe that ICRL is a promising framework for building Large Action Models because it can adapt to new rewards and dynamics, providing richer behavior than the simple expert-distillation pipelines that are currently widely used.
>
> A second contribution of our work is the large multi-domain dataset we collected and standardized. It contains over 700M transitions (a 3.2x increase) across 209 training tasks spans 10 domains, with 46 additional tasks (compared to 15 previously) reserved for evaluation. The field currently has only a limited number of publicly available large-scale datasets, so we view releasing this dataset as a valuable contribution.
>
> Considering this, the use of a flow-matching objective should be understood as a practical choice to handle continuous actions stably in the DPT framework, rather than as the main conceptual contribution.
>
> ## W2. Q1. Q3. Clarifying the types of observations
>
> Thank you for pointing out that the observation types were not clearly specified. All experiments in the current version use low-dimensional proprioceptive observations in every domain.
> In this work, we focused on proprioceptive observations because they require substantially fewer computational resources, which allowed us to scale the model further. Additionally, domains with image-based observations require more time for preprocessing, and including images would significantly increase the size of the dataset. For these reasons, we decided to leave image-based environments for future work.
> Conceptually, extending our framework to image-based observations is not difficult. Image observations can be incorporated by plugging in a pre-trained visual encoder (e.g., CLIP-style) or a trainable hand-designed encoder and feeding its embeddings into the same DPT-based action model without changing the core algorithm. We agree that experiments with high-dimensional visual inputs would be a valuable addition, but collecting the necessary image data and retraining large models with visual encoders  is not feasible during the rebuttal period, so we leave this direction to future work.
>
> ## W3. Q2. Does multi-domain help generalization
>
> To address this concern, we trained separate single-domain models for several representative domains under exactly the same conditions as the 10-domain model (identical architecture size, hyperparameters, and number of seen tokens).
>
> The results in the table below show that the multi-domain model either outperforms or matches the single-domain models on both online and offline metrics, with particularly clear gains on Meta-World and Industrial Benchmark, and no systematic degradation elsewhere. This suggests that, despite using group-specific encoders, the shared backbone does exploit cross-domain data and yields measurable benefits over training separate models.
>
> | Domain | Online Test (Single) | Online Test (Full) | Offline Test (Single) | Offline Test (Full) | Online Train (Single) | Online Train (Full) | Offline Train (Single) | Offline Train (Full) |
> | --- | :---: | :---: | :---: | :---: | :---: | :---: | :---: | :---: |
> | Industrial Benchmark | 0.950 ± 0.004 | ********0.980 ± 0.007****** | 0.950 ± 0.007 | ********0.960 ± 0.009****** | 0.940 ± 0.009 | ********0.990 ± 0.006****** | 1.010 ± 0.001 | ********1.020 ± 0.001****** |
> | Meta-World | -0.000 ± 0.001 | ********0.140 ± 0.023****** | 0.460 ± 0.029 | ********0.690 ± 0.036****** | 0.650 ± 0.015 | ********0.850 ± 0.080****** | 0.990 ± 0.001 | ********1.000 ± 0.000****** |
> | SinerGym | 0.900 ± 0.002 | ********0.910 ± 0.002****** | ********0.970 ± 0.010****** | 0.960 ± 0.017 | 0.960 ± 0.031 | ********1.000 ± 0.026****** | ********1.010 ± 0.004****** | 0.990 ± 0.017 |

---

> ### Comment · Area_Chair_vBW3 · 2025-11-26
>
> The evaluation of this submission has a large variance so we need to gather more information from both the authors and the reviewers. Please take a look at the author's response and see how it affects your evaluation.
>
> Best,
> AC

---

### Author Response · Authors · 2025-12-02
**Summary of Our Rebuttal and Discussions**

Dear Reviewers and ACs,

we are grateful for the time you spent reviewing our paper and for your constructive suggestions on how to improve our work.

Reviewers highlighted the following strengths of our work:

1. High performance on both unseen and seen tasks.
2. The new ~700M-transition dataset (10 domains, 209 training and 46 held-out tasks) was repeatedly noted as a valuable community resource.
3. A single DPT-based Large Action Model trained on a large cross-domain dataset was viewed as relevant and technically non-trivial.
4. Reviewers found the paper generally clear and well-structured.

Reviewers also raised several concerns, which we addressed during the rebuttal with additional experiments and clarifications:

1. **Does multi-domain training help?**

    To test cross-domain sharing, we trained separate single-domain models under identical budgets and compared them to the 10-domain model. The multi-domain model matches or outperforms the single-domain ones on both online and offline metrics, with clear gains on Meta-World and Industrial Benchmark and no systematic degradation, indicating a real benefit from cross-domain data.

2. **Role of the next observation o′o′.**

    Following the original DPT paper, we initially removed o′o′ for efficiency. In rebuttal, we added an ablation in which we trained single-domain models with and without o′o′ under identical settings. The results show no consistent advantage from including o′o′. We explicitly avoid claiming that removing o′o′ is universally harmless, but it does not appear critical in our setup.

3. **Flow-matching vs Gaussian heads.**

    We ran an ablation comparing Gaussian heads (GH) and flow-matching heads (FM). On unseen test tasks, FM is at least as good and often substantially better (e.g., clear gains on SinerGym, HumEnv, Meta-World). Since our goal is generalization to unseen tasks, these results support using FM heads.

4. **Novelty concerns.**

    We clarified that our main contributions are: (i) scaling DPT-based ICRL to a genuinely multi-domain continuous-control setting; (ii) constructing and releasing a substantially larger cross-domain dataset than existing public ICRL data; and (iii) improving performance on unseen tasks compared to Vintix and REGENT in several domains. Flow matching is presented as a practical design choice for continuous actions, not the core novelty, and we clarified algorithmic differences between DPT, Vintix, and REGENT.

5. **Background section.**

    We added a dedicated background section on RL, ICRL, and flow matching in Appendix I, with the intention to move part of it into the main text in a camera-ready version.


All additional experiments and discussions will be incorporated into the final version.

Best,

Authors

---

### Meta-Review · Area_Chair_6KVu · 2026-01-07

**Summary:**

This paper presents a scalable approach to In-Context Reinforcement Learning (ICRL) by extending the Decision Pre-Trained Transformer (DPT) with a Flow Matching policy head. A significant contribution of this work is the curation and standardization of a large-scale cross-domain dataset (over 700M transitions across 10 domains), which facilitates the training of generalist agents capable of adapting to unseen tasks.

The initial reviews were mixed. Reviewers rRdx and KAK3 recognized the value of the large-scale dataset and the clear writing, while Reviewers 3WLW and FmEz raised concerns regarding the novelty, the justification for architectural choices (specifically removing the next observation $o'$), baseline comparisons, and the necessity of Flow Matching.

However, the authors provided an exceptionally thorough rebuttal that addressed the core technical concerns with new empirical evidence:
1.  **Multi-domain vs. Single-domain:** They demonstrated that the shared backbone yields performance gains or parity compared to single-domain models, justifying the multi-domain training approach.
2.  **Ablation of Flow Matching:** New experiments confirmed that the Flow Matching head generalizes better to unseen tasks compared to standard Gaussian heads, validating the architectural contribution.
3.  **Role of Next Observation ($o'$):** An ablation showed that removing $o'$ does not statistically degrade performance in this setting, addressing the theoretical concerns raised by FmEz.
4.  **Baselines:** The authors added comparisons against Vintix trained on the full dataset, demonstrating the superiority of their method.

While Reviewer 3WLW suggests the work might be better suited for a Benchmark/Dataset track, I believe the combination of a high-utility community resource (the dataset), a successfully scaled architecture (Large Action Model), and strong empirical analysis fits well within the scope of the main conference. The paper tackles the difficult problem of generalist agents with a robust solution. Therefore, I recommend acceptance.

**Reviewer Concerns:**

**Addressed Concerns:**
*   **Multi-domain Efficacy (rRdx):** The authors provided a table comparing single-domain vs. multi-domain training, showing clear benefits in transfer and no degradation, effectively settling this concern.
*   **Significance of Flow Matching (KAK3, 3WLW):** The authors conducted an ablation study comparing Flow Matching heads against Gaussian heads. The results showed that Flow Matching consistently performed better or equal on unseen tasks, justifying its inclusion.
*   **Removal of Next Observation $o'$ (FmEz):** The authors ran a controlled experiment with and without $o'$. The results showed negligible performance differences, empirically justifying the efficiency trade-off of removing it.
*   **Baseline Comparisons (3WLW):** The authors retrained Vintix on their 10-domain dataset and provided a direct comparison, where their method outperformed the baseline on key metrics.
*   **Lack of Background (FmEz):** The authors added a background section in Appendix I to formalize the concepts of RL, ICRL, and Flow Matching.

**Outstanding Concerns:**
*   **Venue Fit (3WLW):** Reviewer 3WLW acknowledged the dataset's value but felt the paper was primarily a dataset contribution and suggested it should undergo a full revision for a dataset-specific venue.

**Reviewer Scores:**

*   **Reviewer rRdx (4->6):** Given the successful multi-domain ablation and clarifications, I believe this reviewer would likely raise their score to a **6 (Weak Accept)**.
*   **Reviewer FmEz (4->6):** The reviewer's main technical gripe was the removal of $o'$. The ablation proved this was not detrimental. With this and the added background section, I believe this reviewer would move to a **6 (Weak Accept)**.
*   **Reviewer KAK3 (6->6/8):** The reviewer asked for specific ablations (FM vs Gaussian), which were provided and supported the paper's claims. I believe this reviewer would either maintain the 6 or move to an **8 (Accept)** given the robust response.
*   **Reviewer 3WLW (2->4):** Originally a 2. This reviewer is arguing for a resubmission to a different track. It is unlikely his/her score would change significantly, perhaps to a **4**, as the objection is fundamental to the paper's categorization rather than just its technical correctness.

---

### Decision · Program_Chairs · 2026-01-26

Accept (Poster)